# Wintertime Extreme Warming Events in the High Arctic: Characteristics, Drivers, Trends, and the Role of Atmospheric Rivers

Weiming Ma[1], Hailong Wang[1], Gang Chen[2], Yun Qian[1], Ian Baxter[3,4], Yiling Huo[1], Mark W. Seefeldt[5]

[1]Atmospheric, Climate, and Earth Sciences Division, Pacific Northwest National Laboratory, Richland, WA, USA
[2]Department of Atmospheric and Oceanic Sciences, University of California Los Angeles, Los Angeles, CA, USA
[3]Department of Geography, University of California, Santa Barbara, Santa Barbara, CA, USA
[4]Earth Research Institute, University of California, Santa Barbara, Santa Barbara, CA, USA
[5]National Snow Ice and Data Center, University of Colorado Boulder, Boulder, CO, USA

*Correspondence to*: Weiming Ma (weiming.ma@pnnl.gov), Hailong Wang (Hailong.Wang@pnnl.gov)

**Abstract.** An extreme warming event near the North Pole, with 2-meter temperature rising above 0℃, was observed in late December 2015. This specific event has been attributed to cyclones and their associated moisture intrusions. However, little is known about the characteristics and drivers of similar events in the historical record. Here, using data from ERA5, we study these winter extreme warming events with 2-meter temperature over a grid point above 0℃ over the high Arctic (poleward of 80°N) that occurred during 1980-2021. In ERA5, such extreme wintertime warming events can only be found over the Atlantic sector. They occur rarely over many grid points, with a total absence during some winters. Furthermore, even when occurring, they tend to be short-lived, with the majority of the events lasting for less than a day. By examining their surface energy budget, we found that these events transition with increasing latitude from a regime dominated by turbulent heat flux into the one dominated by downward longwave radiation. Positive sea level pressure anomalies which resemble blocking over the northern Eurasia are identified as a key ingredient in driving these events, as they can effectively deflect the eastward propagating cyclones poleward, leading to intense moisture and heat intrusions into the high Arctic. Using an atmospheric river (AR) detection algorithm, the roles of ARs in contributing to the occurrence of these extreme warming events defined at the grid-point scale are explicitly quantified. The importance of ARs in inducing these events increases with latitude. Poleward of about 83°N, 100% of these events occurred under AR conditions, corroborating that ARs were essential in contributing to the occurrence of these events. Over the past four decades, both the frequency, duration, and magnitude of these events have been increasing significantly. As the Arctic continues to warm, these events are likely to increase in both frequency, duration and magnitude, with great implications for the local sea ice, hydrological cycle and ecosystem.

## 1 Introduction

In recent decades, the Arctic has experienced dramatic changes, with its surface warming at a rate substantially faster than the rest of the world (Serreze and Barry, 2011; Previdi et al., 2021; Rantanen et al., 2022). Such amplified Arctic warming is especially pronounced in winter (Zhang et al., 2021b). Several key mechanisms have been proposed to explain the amplified

warming in the Arctic. These proposed mechanisms include local feedbacks, such as the ice-albedo feedback (Kumar et al., 2010; Screen and Simmonds, 2010; Dai et al., 2019), water vapor and cloud feedback (Vavrus, 2004; Beer and Eisenman, 2022), and lapse rate feedback (Pithan and Mauritsen, 2014; Stuecker et al., 2018), and non-local mechanisms that consist of both ocean and atmospheric heat transport (Hwang et al., 2011; Graversen and Burtu, 2016; Singh et al., 2017; Graversen and Langen, 2019). The atmospheric component can be further decomposed into a sensible heat and a latent heat (moisture) component, with the latent heat component playing a more important role (Graversen and Burtu, 2016). Concurrent with this warming trend is the rapid reduction in both sea ice extent and thickness (Serreze and Stroeve, 2015; Stroeve and Notz, 2018). Although it is still highly debated, Arctic warming and sea ice loss can reduce the meridional temperature gradient and have thus been hypothesized to modulate extreme weather events over mid-latitudes through their influence on large-scale circulation there (Cohen et al., 2014; Screen et al., 2018; Zou et al., 2021; Ma et al., 2021). In addition to these strong trends in the mean climate, the Arctic warming may also lead to more frequent occurrence of local synoptic weather extremes, ranging from rapid sea ice loss (Park et al., 2015; Gimeno et al., 2019; Wang et al., 2020; Zhang et al., 2023), rain-on-snow events (Serreze et al., 2021; Dou et al., 2021), extreme Arctic cyclones (Rinke et al., 2017; Parker et al., 2022; Crawford et al., 2022) to heatwaves (Woods and Caballero, 2016; Graham et al., 2017; Dobricic et al., 2020). The increased frequency and intensification of these weather extremes are expected to exert profound impacts on the Arctic ecosystem and local communities (Amstrup et al., 2010; Post et al., 2013; Ford et al., 2021).

It has been found that, besides contributing to the mean warming over the Arctic, atmospheric moisture transport can exert an important influence on many aspects of the Arctic climate. For example, enhanced atmospheric moisture transport into the Arctic in spring can lead to anomalous downward longwave radiation and thus precondition the sea ice for a more rapid melt in the subsequent months (Kapsch et al., 2013; Schröder et al., 2014; Mortin et al., 2016). Spring atmospheric moisture convergence can thus serve as an important predictor for summer minimum sea ice extent. In addition to its effects on the sea ice, recent increases in the Arctic river discharges have also been found to be primarily driven by an increase in poleward atmospheric moisture transport (Zhang et al., 2013). As sea ice continues to decline and atmospheric moisture transport continues to increase, the interannual variability of Arctic precipitation is expected to be increasingly controlled by atmospheric moisture transport (Bintanja et al., 2020).

Studies have shown that the bulk of atmospheric moisture transported into the Arctic is accomplished by episodes of extreme moisture transport events, termed Arctic moisture intrusions or atmospheric rivers (ARs) (Nash et al., 2018; Zhang et al., 2023). ARs, filaments of intense moisture transport in the atmosphere, have traditionally been identified as mid-latitude phenomena. Early studies found that, despite occupying only about 10% of the mid-latitude circumference, ARs are responsible for more than 90% of the poleward moisture transport there (Zhu and Newell, 1998). ARs have been studied extensively for mid-latitude regions due to their important contribution to the regional hydrological cycle (Leung and Qian, 2009; Viale et al., 2018; Lavers and Villarini, 2015; Waliser and Guan, 2017; Lamjiri et al., 2017; Pan and Lu, 2020). In recent years, it gets increasingly recognized that ARs also exert considerable impacts on the Arctic climate (Hegyi and Taylor, 2018; Nash et al., 2018; Zhang et al., 2023). When a large amount of moisture carried by ARs intrudes into the Arctic in a short

period of time, the rapid moistening of the lower atmosphere and the resulting cloudy condition and enhanced downward longwave radiation can lead to a rapid rise in surface temperature and substantial sea ice loss. Therefore, ARs are potentially an important driver of extreme events over the Arctic region.

In late December 2015, the high Arctic near the pole experienced an episode of extreme warming, with surface temperature exceeding 0℃ (Moore, 2016; Graham et al., 2017; Binder et al., 2017). Subsequent examination reveals the short-lived nature of this event, with buoys close to the North Pole recording temperatures above 0℃ for less than an hour (Fig. 1 in Graham et al. (2017) and Fig. 2 in Moore (2016)). This event was driven by an AR-like moisture plume carried into the high Arctic by a cyclone (Moore, 2016). While this event has been studied in detail, our knowledges about similar events over the high Arctic during winter is far from being complete. To fill this knowledge gap, in this study, using high spatiotemporal resolution data from the European Centre for Medium-Range Weather Forecasts Reanalysis, version 5 (ERA5; Hersbach et al., 2020), we seek to address the following questions: (1) What are the characteristics of extreme warming events in ERA5, in terms of duration, frequency and temporal clustering? (2) How does the surface energy budget evolve during the development of these events?  And what is the dominant energy budget term associated with these events? (3) What are the favorable large-scale circulation patterns driving these events? (4) What roles do ARs play in the initiation of these events in particular?

This paper is structured as follows. Section 2 describes the data and the AR detection algorithm used. In section 3, we first show the characteristics of Arctic extreme warming events, in terms of spatial distribution, duration and occurrence frequency. We then investigate the driving mechanisms of these events from the surface energy budget and large-scale circulation perspectives. Attentions are given to those events with very short or very long duration. As will be shown, the large-scale circulation associated with these events exhibits typical patterns that favor the intrusion of moisture into the Arctic. We thus further quantify the role of ARs in contributing to the occurrence of or intensifying these events. Lastly, the trends of Arctic extreme warming events are also explored. In section 4, we conclude by giving a brief summary of the major findings and a discussion on some limitations of this study.

## 2 Methods

### 2.1 Data

To examine the spatiotemporal characteristics of these extreme warming events in detail, we employ hourly data from ERA5, with a spatial resolution of $0.25° \times 0.25°$. Previous studies suggest that, compared to in-situ observations, ERA5 exhibits a warm bias in wintertime Arctic surface air temperature (Graham et al., 2019). However, recent studies pointed out that the warm bias in ERA5 may be state-dependent, with a positive bias found under radiatively clear condition and a negative or negligible bias under opaquely cloudy condition (Batrak and Müller, 2019; Herrmannsdörfer et al., 2023). Further research is needed to evaluate the performance of ERA5 under extreme warming conditions over the Arctic.

## 2.2 Definition of extreme high Arctic warming events

In this study, we define Arctic warm events in two ways: 1) as grid points with 2-meter air temperature (T2m) >= 0°C, which is the main focus of the study, and 2) as contiguous regions with T2m >= °C, which complements the results from the grid-point scale perspective and will be described in more detail in Section 3.3. Case studies based on in-situ buoy observations have been conducted by previous studies to examine the characteristics and drivers of winter extreme warming events over the high Arctic (Moore, 2016; Graham et al., 2017), which characterized extreme warming events using point observations based on meteorological buoys. Since these extreme warming events are usually driven by large-scale circulation (Woods and Caballero, 2016; Moore, 2016; Kim et al., 2017; Messori et al., 2018), when extreme temperature is detected over a grid point, it is common that such extreme temperature can also be found over its neighbouring grid points. Under such conditions, one approach is to define extreme warming events by using contiguous regions with temperature exceeding a predefined threshold. This approach allows us to investigate the spatial extent of the events and their variability, which is an important characteristic for heat extreme. Furthermore, defining extreme warming events this way also makes the analysis of the large-scale driver convenient. However, to facilitate a more direct comparison with previous case studies shown in Moore (2016) and Graham et al. (2017), here we focus on the first approach which defines extreme warming events at the grid-point scale. As will be shown later in the manuscript, this approach is also amenable for analysing the spatial heterogeneity of the event characteristics, such as event duration and event surface energy budget. Specifically, in this study, extreme warming events are defined as those events with T2m over a grid point reaching or exceeding 0°C over the high Arctic that covers the regions poleward of 80°N. We focus on the winter season (December to February or DJF). The event starts when the T2m first reaches or exceeds 0°C, and the event is considered to end at the time when the T2m first drops below 0°C. The time between the start of the event and termination is considered as the duration of the event. It is possible that T2m can rise to or above 0°C within a few hours following the termination of a previous extreme warming event over the same grid point. When such situation occurs, it is likely that both extreme warming events are driven by the same weather system. To test how sensitive the results are to these situations, we impose a 24-hour, 48-hour or 120-hour interval requirement between the termination of an event and the start of a following event over the same grid point. If the following event occurs within 24, 48 or 120 hours after the termination of the previous event, it is excluded from the analyses. We found that only the surface energy budget of the long duration events (defined in Section 3.2) prior to the onset is sensitive to whether a time interval requirement is imposed. More specifically, imposing a 120-hour interval requirement makes the anomalies of turbulent fluxes, T2m and integrated water vapor decrease from around zero (Fig. 6g) to a negative value from around 5-day lag to 2-day lag (not shown). This suggests that previous long-lasting events are probably double counted as part of the next long-lasting events prior to their onset when no time interval is imposed. Other than that, all the results presented in this study are very similar regardless of whether this additional constraint is imposed or not (not shown). For simplicity, the results being presented in this study are obtained without imposing this constraint.

## 2.3 Calculation of anomaly fields

Following Wang et al. (2020), all anomalies presented in this study are obtained by first removing the seasonal cycle. To obtain the seasonal cycle in the first place, the hourly time series is averaged across the years from 1979 to 2021 to get a raw seasonal cycle, and then a 31-day running mean is subsequently applied to obtain a smoothed seasonal cycle for de-seasonalizing. The Arctic has experienced amplified warming in the past four decades. This amplified warming is associated with significant positive trends in the downward longwave radiation (DLW), integrated water vapor transport (IVT) and integrated water vapor

(IWV) over nearly the entire high Arctic and significant positive (or negative) trends in the latent heat flux (LHF) and sensible heat flux (SHF) over the sea ice covered (or partially sea ice covered) region of the high Arctic Atlantic sector (not shown). To exclude the effects of these decadal trends, the anomalies of T2m, DLW, IVT, LHF, SHF, and IWV are further detrended after removing the seasonal cycle. Similarly, for any given hour, to obtain the linear trend, a 31-day running mean centered at the hour is first applied to the de-seasonalized data. The smoothed data is then used to derive the linear trend used for the

detrending.

## 2.4 AR detection Algorithm

We use the IVT-based AR detection algorithm developed by Guan and Waliser (2019). This algorithm is an updated version of the original AR detection algorithm, documented in Guan and Waliser (2015), which is one of the earliest and most popular automated AR detection algorithms in the AR community. In addition, it is recommended by the Atmospheric River Tracking

Method Intercomparison Project (ARTMIP) for AR research over the high latitudes (Rutz et al., 2019). Notable common criteria employed by both algorithms are as follows: (1) a seasonally and regionally dependent 85th percentile of the IVT magnitude or 100 kg m$^{-1}$ s$^{-1}$, whichever is larger, is used as the threshold to identify contiguous regions of enhanced IVT ("object"); (2) to ensure coherence, at least half of the grids within the identified object need to have a IVT direction within 45° of the object mean IVT direction; (3) the object mean poleward IVT exceeds 50 kg m$^{-1}$ s$^{-1}$; lastly (4) the detected object is

longer than 2000 km and with a length-to-width ratio exceeding two. Compared to the original algorithm, the updated algorithm includes several major refinements: (1) iterative thresholds are used to increase the chance of an "object" to be detected as AR. In this study, five iterative thresholds are used: 85th, 87.5th, 90th, 92.5th and 95th; (2) improvements are also made on the identification of the AR axis that helps to better characterize the AR length and orientation; (3) the function of tracking of individual ARs across space and time is enabled. By the time when ARs reach the Arctic, they are usually near the end of their

life cycle. Following Mattingly et al. (2018), the length requirement for a detected AR is thus relaxed from 2000 km to 1500 km.

The zonal and meridional components of the IVT vector are calculated by vertically integrating the moisture flux at 1000, 850, 700 and 500 mb following:

$$zonal\ IVT = \frac{1}{g} \int_{1000}^{500} uqdp$$

$$meridional\ IVT = \frac{1}{g}\int_{1000}^{500} vqdp$$

where g is the gravitational acceleration, u is the zonal and v is the meridional wind component, and q is specific humidity. Because using hourly data with the original spatial resolution as input to the AR detection algorithm is computationally too expensive, only data at 00, 06, 12, and 18 UTC are used for AR detection. Furthermore, we bi-linearly regrid the data to $1° \times 1°$ before calculating IVT. The generated AR statistics are then mapped back to the original resolution ($0.25° \times 0.25°$) using the nearest neighbour method.

## 3 Results

### 3.1 Characteristics of the high Arctic extreme warming events

Over most of the high Arctic, winter mean T2m is generally below $-20$ °C, with the exception found over a small region near $80°$N of the Atlantic sector (Fig. 1a). The mean T2m there can reach as high as $-13°C$. Given such severe cold conditions over the wintertime high Arctic, temperature above $0°C$ indeed represents an extreme condition over there. By examining the maximum hourly T2m of all the winter months (December to February) from 1979 to 2021, we found that only regions over the Atlantic sector ever experienced T2m above $0°C$ (Fig. 1b), which is consistent with the fact that the Atlantic sector serves as the major pathway for moisture and heat transport into the Arctic (Dufour et al., 2016; Yang and Magnusdottir, 2017; Papritz et al., 2022). The Pacific sector, as another important pathway for moisture and heat transport into the Arctic (Dufour et al., 2016; Gimeno et al., 2019), has never been warm enough to break $0°C$ over 1979-2021, although the winter maximum hourly T2m reached as high as about $-1°C$. Therefore, we focus mainly on the Atlantic sector from now on.

Consistent with the buoy in-situ observations (Moore, 2016; Graham et al., 2017), these extreme warming events with T2m over a grid point staying continuously above 0°C tend to be short-lived, with about 70% and 90% of the events lasting shorter than half a day and one day, respectively (Fig 2c). On average, these events lasted for 11.55 hours. The most long-lasting events are found over the regions close to $80°$N between $0°$-$30°$E, with a mean duration of about 16 hours (Fig. 2a). Over this region, those extreme long-lasting events, which are defined as events with duration longer than the local 95[th] percentile duration, can last longer than two days (Fig. 2b). Moving away from this region, the mean duration drops gradually. Over regions eastward of about $60°$E and poleward of about $80°$N, the mean duration is shorter than five hours. Over those regions, even the extreme long-lasting events tend to last less than 10 hours, further confirming the short-lived nature of these events over the winter high Arctic observed on buoys (Moore, 2016; Graham et al., 2017).

In addition to being short-lived, these events also occurred very rarely over many of the grid points of the winter high Arctic (Fig. 3). Regions with mean seasonal occurrence frequency greater than one can only be found over a confined region near $80°$N between $0°$-$30°$E (Fig. 3a). Moving away from this region, the occurrence frequency decreases dramatically and becomes less than one over most of the regions. When only those winters with at least one event occurred are considered, the mean occurrence frequency over most regions only increase slightly to 1-2 events per season (Fig. 3b). Note that while the occurrence

frequency of these extreme warming events can be considered rare over many of the grid points, over the entire high Arctic, they occur quite frequently. There was at least one extreme warming event somewhere in the studied region in 40 out of the 42 studied winters (Fig. S1). This appears to disagree with the findings in Moore (2016) and Binder et al. (2017). Specifically, Moore (2016) found that wintertime Arctic extreme warming events with T2m >= 0°C occur only once or twice each decade. Three events were identified from 1958 to 2015 in their study. Binder et al. (2017) found 12 out of the 36 winters with at least one extreme warming event over the wintertime high Arctic from 1979 to 2014. These seeming discrepancies between the results in our studies and those found in Moore (2016) and Binder et al. (2017) stem mostly from the differences in the datasets used, the definition of the high Arctic and the definition of the winter season. In particular, Moore (2016) defines Arctic extreme warming events as those events occurring poleward of 85°N during December in Japanese 55-year Reanalysis (JRA-55) (Kobayashi et al., 2015). As shown later, there are only 9 days with extreme warming events identified during the studied period if we change the analysis regions to poleward of 85°N, which brings our result much closer to that in Moore (2016). The slightly more events identified in our study can be attributed to the differences in the definition of winter season (i.e., December-February versus December) and datasets used (i.e., ERA5 versus JRA55). Following Binder et al. (2017), when the high Arctic is defined as regions poleward of 82°N, we find 16 out of 36 winters with extreme warming events from 1979 to 2014 (not shown), which also makes our quantitative result more comparable to that in Binder et al. (2017). The remaining difference is likely due to the different spatiotemporal resolution of datasets used between studies (i.e., ERA5 versus ERA-Interim).

Besides being rare, by examining the time interval between termination of an event when a grid point's T2m first falls below 0°C and the start of the subsequent event when a grid point's T2m first exceeds 0°C for those winters with at least two events occurring over a grid point, we found that these events tend to exhibit some degree of temporal clustering (Fig. 4). About 80% of the events reoccurred in less than five days (Fig. 4a). Over those regions where the mean seasonal occurrence frequency is less than one, the mean time interval in between events is usually less than two days, suggesting that these events tend to occur back-to-back (Fig. 4b). As being shown in Section 3.3, the clustering of these extreme warming events may be driven by the same persistent large-scale circulation that steers successive weather systems into the affected region. These results suggest that the preconditioning by the previous events likely plays an important role for the initiation of the subsequent events.

## 3.2 Surface energy budget associated with the high Arctic extreme warming events

It is found that enhanced DLW plays an important role for Arctic warming during winter (Messori et al., 2018; Park et al., 2015; Woods and Caballero, 2016; Murto et al., 2023). To investigate whether DLW also plays a role in the extreme warming events defined in this study, we examine the anomalies of the surface energy budget terms when the T2m >= 0°C. The total number of hours with T2m >= 0°C over the study period reaches its maximum (more than 1000 hours) over the region near 80°N and between 0°-30°E (Fig. S2). Moving away from this region, the number drops rapidly towards the boundary with regions that have never experienced T2m >= 0°C during winters of the past four decades. As shown in Fig. 5, when T2m >= 0°C, the atmosphere above is anomalously moist, with IWV anomalies exceeding 4 kg m$^{-2}$ nearly everywhere (Fig. 5b).

Regional IWV anomalies can reach up to about 9 kg m$^{-2}$. With the winter IWV climatology being less than 3.5 kg m$^{-2}$ nearly everywhere over the Atlantic sector in the high Arctic (not shown), such moistening of the atmosphere represents 100-300% increases in the IWV and is most likely achieved by intense moisture intrusions from the lower latitudes. Consistent with the moistening of the atmosphere, DLW anomalies also increase substantially everywhere (Fig. 5c). The enhancement of the DLW anomalies increases with latitude and can reach up to about 130 W m$^{-2}$ poleward of 85$^{\circ}$N. The spatial pattern of the DLW anomalies corresponds well with that of the IWV anomalies, suggesting the importance of enhanced IWV in inducing the anomalous DLW. Over regions close to 80$^{\circ}$N, the magnitude of DLW anomalies is relatively weaker. However, the downward turbulent heat flux (THF) anomalies, especially SHF, are substantially intensified (Figs. 5d, e). Climatologically, the THF over these regions is upward, with magnitudes reaching more than 100 W m$^{-2}$, due likely to the partially open ocean underneath (Fig. S3). During extreme warming events, the strong advection of both moisture and heat in the lower atmosphere (Figs. 5b, f) likely results in the reversal of the vertical temperature gradient, leading to a strong suppression of the upward THF over these partially sea ice covered regions. In contrast to the spatial pattern of DLW anomalies, the SHF anomalies weaken with latitude. Compared to DLW and SHF, downward LHF anomalies are substantially weaker (Fig. 5e), with the magnitude decreasing with latitude and even a reversed sign over the most north-western regions. The anomalous upward LHF over this region is likely caused by the rapid cold and dry advection shortly after the start of extreme warming events when a grid point's T2m first exceeds 0°C, resulting in a reversal of the moisture gradient and overall weak temperature advection over the region (Fig. 5f). As shown later, such a rapid transition to cold and dry advection could be caused by the passage of a cold front. The patterns of these surface energy budget terms suggest that the winter extreme warming events over the high Arctic can be categorized into two types: (1) the DLW dominance type, which usually occurs poleward of about 83$^{\circ}$N, and (2) the SHF dominance type, which occurs over the lower latitude regions near 80$^{\circ}$N. These results are consistent with Murto et al. (2023) who examines the wintertime high Arctic extreme surface energy budget anomalies and finds that DLW plays an increasingly more important role as events move further into the Arctic.

Next, we examine the temporal evolution of the anomalous surface energy budget for the extreme warming events (Fig. 6). Even nine days before the start of the extreme warming events when a grid point's T2m first exceeds 0°C, T2m is already 3-4 °C higher than normal (Fig. 6a). This is likely because these extreme warming events occur more often during warm winters, but it is also possible that this warm anomaly is preconditioned by a previous moderately or extremely warm event. The background temperature over the Atlantic sector, under which these events occur, is thus anomalously warm. Indeed, the occurrence frequency of the events correlates significantly with the winter mean T2m over the Atlantic sector of the Arctic (or the entire Arctic), with a correlation coefficient of about 0.64 (0.61). Consistent with the warm anomalies, both IWV and DLW show positive anomalies. These results suggest that while the occurrence of these extreme warming events contributes to making the background state anomalously warm and moist, a warm and moist winter in turn favours the occurrence of extreme warming events. About four days prior to the start of the events, T2m, all surface energy budget terms, warm advection and IWV start to climb and peak at around the start of the events. Both the SHF and DLW play comparable roles in contributing to the formation of these events, with SHF having a slightly larger magnitude. Following the start of the events, while the other

terms drop more gradually, warm advection ceases immediately and shifts to weak cold advection afterwards, implying the passage of cold fronts.

To understand what determines the duration of the events at grid-point scale, we further divide the extreme warming events into short duration events (i.e., grid points experiencing the warming for a short duration), defined by duration $<=$ the 5$^{th}$ percentile of the duration of all events (1 hour), and long duration events (i.e., grid points experiencing the warming for a long duration), defined by duration $>=$ the 95$^{th}$ percentile of the duration of all events (40 hours). The long duration events only occur equatorward of 85$^{°}$N while the short duration events can be found both poleward and equatorward of 85$^{°}$N. As shown

above, extreme warming events can be divided into the DLW dominated ones that occurred over the fully sea ice covered regions poleward of ~83°N and SHF dominated ones that occurred over the partially sea ice covered regions equatorward of ~83°N (Fig. S3). We thus further divide the short duration events into those occurring poleward of 85$^{°}$N and equatorward of 83$^{°}$N to maximize the differences between these two types of events. Since the composites for all the events are dominated by the events with relatively short duration over lower latitude regions, the temporal evolution of T2m, surface energy budget

terms, temperature advection and IWV for the short duration events equatorward of 83$^{°}$N are very similar to those for all the events (Figs. 6c, d vs Figs. 6a, b). For the short duration events poleward of 85$^{°}$N, the anomalies of T2m, DLW, SHF and IWV start to increase about six days prior to the start of the events when a grid point's T2m first exceeds 0°C, suggesting that more persistent weather patterns are required to initiate the extreme warming events poleward of 85$^{°}$N (Figs. 6e, f). Indeed, these events are dominated by the DLW anomalies, with their magnitude being twice those of the SHF anomalies. Following the

start of the events, all quantities drop sharply. The anomalies of SHF and LHF even reverse sign and the warm advection prior to the start of the events shifts to strong cold advection. Contrary to the short duration events poleward of 85$^{°}$N, the long duration events are dominated by SHF anomalies, with their magnitude, in this case, being twice those of the DLW anomalies (Fig. 6g). The anomalies did not increase until about two days prior to the start of the events. Unlike the long duration events, all anomaly terms remain elevated for a prolonged period after the start of the events, and then gradually level off (Figs. 6g,

280  h)

To further investigate the mechanisms that determine the duration of these events, we create anomaly composites for temperature advection, IVT and SLP centred at the grid point where extreme warming events occur. As the same large-scale circulation pattern can cause extreme warming events to occur over more than one grid point, it thus can be counted more than once within the same composite or across different composites. However, as the composites are centred at the grid point where

the extreme warming event is identified, the position of the same large-scale circulation relative to the grid point would thus differ among the composites centred at different grid points. As we show below, the relative position of the SLP anomalies is what determines the duration of the extreme warming events. The double counting of the same circulation pattern for the composites shown in Figs. 7, 8, and 9 thus has minimal impact on the conclusion. As shown in Fig. 7, the start of the short duration events equatorward of 83$^{°}$N is associated with a positive SLP anomaly to their southeast and a negative SLP anomaly

to their west. This circulation pattern effectively channels moisture and heat into the regions where the extreme warming events occur (Woods and Caballero, 2016; Messori et al., 2018; Wang et al., 2020). Even six days prior to the start of the events when

a grid point's T2m first exceeds 0°C, the positive SLP anomaly already appears. A weak warm advection can also be found south of the event regions. As time evolves, the positive SLP anomaly deepens, and a negative SLP anomaly starts to develop over the west of the events. Concurrent with these changes in the SLP, moisture transport and warm advection intensify over the event regions. Less than one day prior to the start of the events, a cold advection anomaly develops west of the events and moves over the event regions immediately following the start of the events, leading to the drop in T2m. By day six after the start of the events, the SLP dipole pattern mostly vanishes. For the short duration events poleward of 85°N, the SLP anomaly dipole already starts to develop six days prior to the start of the events (Fig. 8). As time evolves, the dipole pattern intensifies, resulting in a strong SLP gradient over the event regions. Consistent with the presence of the strong SLP gradient, moisture and heat advection enhance greatly compared to the short duration events equatorward of 83°N. Less than one day prior to the start of the events, a strong cold advection is well developed over the west of the events. It immediately moves over the event regions after the start of the events, leading to sharp drops in the T2m and other surface energy budget terms. Like the short duration events equatorward of 83°N, the positive SLP anomaly southeast of the event regions already starts to develop for the long duration events six days before the start of the events (Fig. 9). However, at the meantime, a negative SLP anomaly can also be found southwest of the event regions. As the events evolve, the positive SLP anomaly intensifies while the negative SLP anomaly extends northward. This configuration in the dipole pattern leads to sustained moisture and heat advection into the event regions. Unlike the short duration events, the cold advection west of the long-duration event regions never develops, allowing the warm anomalies to persist. These results suggest that the position of the negative SLP anomaly relative to the event regions plays a key role in determining the duration of the events. When the negative SLP anomaly is located at the west of event regions, cold advection develops and moves over the event regions immediately after the start of the events, causing the T2m to fall below 0°C. However, when the negative SLP anomaly locates at the southwest of the event regions, cold advection never develops, resulting in sustained moisture and heat advection into the event regions, leading to prolonged warm anomalies.

### 3.3 Large-scale circulation associated with concurrent warming events

The analyses presented so far are all based on extreme warming events occurring at grid-point scale. We next focus on large-scale circulations responsible for driving concurrent warming events over a large area of the Atlantic sector. To do that, we first calculate the total area with T2m >= 0°C over the high Arctic at each hourly snapshot. We then identify all the periods with T2m >= 0°C continuously over at least one grid point. These periods are defined as a concurrent warming event. The onset of these concurrent warming events is then defined as the time when the area with T2m >= 0°C first exceeds zero (one grid point), and the event ends when the area with T2m >= 0°C first falls back to zero. It is possible that the termination of one event is followed shortly by the onset of a subsequent event. Under such a situation, these two concurrent warming events are likely influenced by the same large-scale circulation pattern. We thus impose a constraint that the time interval between the onset of one event and the termination of the subsequent event needs to be longer than five days. Otherwise, the subsequent event is discarded in our analysis of large-scale circulations. Lastly, to focus on the most intense events, only those events with

a peak area larger than $5 \times 10^9$ m$^{-2}$ are retained for analyses. There are a total of 96 events that satisfy these criteria. Further analyses show that the timing of the peak area corresponds well with the timing of the maximum T2m anomaly averaged over the Atlantic sector of the high Arctic (-15°W – 60°E and poleward of 80°N, roughly corresponding to the region ever experienced T2m >= 0°C shown in Fig. 1b). For example, 92 (82) out of the 96 identified concurrent warming events have their peak area occurred within 24 (12) hours of the timing of the maximum T2m anomaly. As shown in Fig. 10 (1st column), even four days prior to the peak of the events (day 0), which is defined as the time when the total area of extreme warming events reaches maximum, a positive SLP anomaly and a negative SLP anomaly start to appear over the northwest Eurasia and west Greenland, respectively. As time evolves, the dipole pattern intensifies, and the negative SLP anomaly also moves poleward. The anomalous dipole reaches maximum magnitude during the peak of the events and channels large amount of moisture into the Arctic. Four days after the peak of the events, the dipole mostly dissipates. These results further corroborate the importance of the anomalous SLP dipole in driving the Arctic weather extremes found in previous studies (Woods and Caballero, 2016; Messori et al., 2018; Wang et al., 2020; Zheng et al., 2022).

To gain a more detailed understanding on the spatiotemporal evolution of the large-scale circulation, we apply a K-means clustering method to the spatiotemporal evolution of all 96 events from six days prior to and after the peak of the events. We varied the numbers of clusters ranging from two to four. Three clusters are identified, which give a good balance between the numbers of events in each cluster and sufficient representation of the large-scale circulation patterns. The first cluster features a strong dipole pattern in the SLP anomalies, with a positive SLP anomaly over northwest Eurasia and a negative SLP anomaly over Greenland (2nd column in Fig. 10). As time evolves, the negative SLP anomaly intensifies and propagates into the Arctic and then dissipates over the Laptev Sea, while the positive SLP anomaly remains relatively stationary. The second cluster exhibits a strong and persistent positive SLP anomaly over northern Eurasia, while the negative SLP anomaly is very weak (3rd column in Fig. 10). As time evolves, the negative SLP anomaly moves poleward and dissipates rapidly over the Beaufort Sea. Contrary to the second cluster, the positive SLP anomaly in the third cluster is very weak and short-lived (4th column in Fig. 10). This cluster is dominated by a negative SLP anomaly over Greenland. Unlike the other two clusters, the negative SLP anomaly predominantly exhibits a westward movement. These results suggest the importance of blocking-like structures for steering cyclones into the Arctic and are consistent with previous studies on the roles of blocking in transporting moisture and heat into the Arctic (Papritz, 2020; Papritz and Dunn-Sigouin, 2020; Murto et al., 2022; Papritz et al., 2022).

Different spatial patterns of the large-scale circulation can result in different impacts (Fig. 11). Compared to the T2m anomaly composite at the peak time of all events (Fig. 11a), the first cluster, which is dominated by a strong dipole pattern of SLP anomalies, shows an overall stronger warming over the Atlantic sector (Fig. 11b). However, the area with significant warming is slightly smaller than that based on all the events. The warming anomalies over the Atlantic sector based on the second cluster, which is dominated by the positive SLP anomaly, are comparable to the composite of all the events, but it exhibits the largest spatial extent compared to those based on all the events and the other two clusters (Fig. 11c). Lastly, the warm anomaly based on the third cluster, which is dominated by the negative SLP anomaly, exhibits the weakest warming over the Atlantic sector, and the spatial extent of the warming anomaly also is confined over the Atlantic sector only (Fig. 11d). Therefore, the

presence of the positive SLP anomalies which resemble blocking are important in determining both the magnitude and spatial extent of the warm anomalies. These results further imply that the persistence of the blocking-like structures over the northwest/northern Eurasia can lead to sustained moisture and heat advection into the Arctic. This, in turn, can precondition the ambient in such a way that once a weather system, such as a cyclone, gets steered into the Arctic, it readily triggers the occurrence of the concurrent warming events.

Previous studies have shown that such a dipole pattern in the SLP anomalies is ideal for moisture intrusions or ARs moving into the Arctic (Park et al., 2015; Woods and Caballero, 2016; Messori et al., 2018; Wang et al., 2020; Papritz and Dunn-Sigouin, 2020; Papritz et al., 2022). Indeed, we found that, during the peak time of these concurrent warming events, the AR occurrence frequency, defined as the fraction of time when a grid point is under AR conditions, increases substantially (Figs. 11e-h). The AR frequency can even exceed 30%. With the winter climatological AR frequency ranging from about 0.5-2.5% over the region, this represents an over 10-fold increase in AR frequency. Notably, cluster one, which corresponds to the strong dipole pattern in SLP, is most effective in guiding ARs into the Arctic, while cluster three, which corresponds to the negative pressure dominated pattern in SLP, is least effective in guiding ARs into the Arctic.

### 3.4 Roles of ARs in contributing to the occurrence of extreme warming events

The above analyses suggest that a strong moisture and heat transport by ARs likely plays an important role in contributing to the occurrence of the extreme warming events at the grid-point scale. To better quantify this role of ARs, we first examine the surface energy budget during AR days (Fig. 12). Here, AR days experienced over any grid point are defined as those days with at least one 6-hourly time at 00, 06, 12 or 18 UTC under AR conditions. During AR days, the surface is anomalously warm, with T2m anomalies exceeding 10℃ nearly everywhere (Fig. 12a). Concurrently, the atmosphere is anomalously moist, with IWV anomalies above 3 kg m$^{-2}$ over most of the area (Fig. 12b). ARs also lead to ubiquitous warm advection (Fig. 12f). Both the sensible heat and latent heat transported by ARs into the Arctic lead to enhanced DLW anomalies, with a magnitude exceeding 60 W m$^{-2}$ nearly everywhere (Fig. 12c). ARs also lead to downward anomalies in both the SHF and LHF, especially over the regions only partially covered by sea ice near 80°N (Figs. 12d, e). These results confirm that ARs indeed have a strong warming effect over the high Arctic during winter (Woods and Caballero, 2016; Zhang et al., 2023).

To show the tight connection more explicitly between ARs and the extreme warming events defined at the grid-point scale, Fig. 13a shows the fraction of the extreme warming events which occurs during AR days. Equatorward of about 83°N where the background temperature is relatively warm, ARs are not the necessary conditions for initiating extreme warming events. There is still a nonnegligible fraction of events that are driven by other weather disturbances. However, the role of ARs becomes increasingly important as the extreme warming events occur over more poleward regions. Poleward of about 83°N, the fraction of extreme warming events that occur during AR days reaches 100%. Further examining the temporal evolution of IVT for the extreme warming events reveals that IVT usually peaks two to three hours prior to the start of the events (Fig. S4). The results here thus suggest that, for a large fraction of the regions where extreme warming events can occur, the presence of ARs and their impact on and interaction with the local environment (Papritz et al., 2023) likely exert a strong control on the

occurrence of these events. Climatologically, the fraction of time with T2m above zero is very close to zero nearly everywhere, except over a small region near 80$^\circ$N and between 0$^\circ$-30$^\circ$E where the fraction can exceed 6% (Fig. 13b). However, if only AR days are considered, the fraction of time with T2m above zero increases substantially (Fig. 13c). By defining the ratio of the fraction of time with T2m above zero during AR days to that of all days as the risk ratio, we can see that ARs increase the risk of extreme warming events dramatically, ranging from about 10 times more likely over lower latitude regions to about 50 times more likely over higher latitude regions (Fig. 13d). ARs thus serve as a key ingredient for the occurrence of the extreme warming events over the high Arctic during winter.

An in-situ observed extreme warming event with T2m >= 0°C happened near the end of 2015 and over regions close to the pole (Moore, 2016). If we focus on the regions poleward of 85$^\circ$N, there are only nine days when T2m >= 0°C was found over at least one grid point of the regions from 1979 to 2021. ERA5 successfully simulates the occurrence of the extreme warming event during 12/29/2015-12/30/2015. For all these nine days, ARs can be found intercepting the 85$^\circ$N latitude over 15$^\circ$W-60$^\circ$E for at least one 6-hourly time step of each day, which are defined as AR deep intrusion days, suggesting that all these events are associated with ARs. Compared to the AR deep intrusion days without extreme warming events occurring poleward of 85$^\circ$N, those deep intrusion days with extreme warming events found poleward of 85$^\circ$N exhibit a much more intense filament of IVT (Figs. 14a vs 14b). The IVT filament also penetrates deeper into the high Arctic. In line with the stronger IVT, the SLP dipole also intensifies, with the negative SLP center locating more poleward. As has been shown in previous studies (Messori et al., 2018; Murto et al., 2022), the more poleward located negative SLP center could be associated with the locally generated Arctic cyclones. During deep intrusion days, the daily IVT averaged over regions poleward of 85$^\circ$N and between 15$^\circ$W-60$^\circ$E increases substantially from the climatological daily mean of ~25 to ~78 kg m$^{-1}$ s$^{-1}$ (Fig. 14c). Out of the ten (five) highest daily IVT averaged over the defined region, eight (five) of them are associated with extreme warming events occurring poleward of 85$^\circ$N, further confirming the extreme nature of these extreme warming events.

### 3.5 Trends of extreme warming events

In the past four decades, winter mean T2m poleward of 80$^\circ$N has been increasing significantly at a rate of 0.8 ℃ decade$^{-1}$ (Fig. 15a). Consistent with the overall climate warming, both the winter maximum hourly T2m and the mean T2m for grid points above 0℃ increase significantly at 0.4 ℃ decade$^{-1}$ and 0.09 ℃ decade$^{-1}$, respectively. The slower increase of these extreme T2m events is likely due to the presence of underlying sea ice that imposes a constraint on their warming rates. In line with Graham et al. (2017), the background warming also makes the occurrence of the extreme warming events more likely. The event occurrence frequency has been increasing at a rate of 2150 events per season per decade for the extreme warming events defined at the grid-point scale (Fig. 15b). Consistent with the increasing trend in the occurrence frequency, both the number of days and the number of hours with at least one extreme warming event found over the high Arctic exhibit significant upward trends with magnitude of 6.8 days per season per decade and 114 hours per season per decade (Fig. S1). At the same time, they also become more persistent, with the mean duration increased by 1.5 hours per decade. The duration of most long-lasting events each year has increased at an even faster rate of 17.6 hours per decade. Consistent with these increasing trends in the

425 characteristics of the extreme warming events defined at the grid-point scale, the frequency, spatial extent, and duration of the concurrent warming events defined in Section 3.3 also exhibit significant positive trends in the past four decades (Figs. 15c, d).

Given the significant increase in both the event frequency and duration, it is natural to ask whether the increases are solely driven by the background warming or whether changes in AR frequency also play a role. Over the Atlantic sector of the high Arctic, ARs show positive trends over most of the regions in the past four decades (Fig. S5a). However, significant trends are only found over a small region near the pole. Following Ma et al. (2020), we further decompose the trends into a dynamical component, driven by changes in atmospheric circulation, and a thermodynamic component, driven by changes in the moisture field. The decomposition reveals a counterbalancing effect between the two components (Text S1 and Fig. S5). The moistening of the Arctic atmosphere has resulted in a substantial increase in AR frequency, especially over the regions equatorward of 85°N (Figs. S5c and S6a). However, the weakening of winds leads to a reduction in the AR frequency (Fig. S5b and S6b). These two components combined result in insignificant positive trends in the AR frequency over most of the regions. Based on these results, the roles played by changes in AR frequency are likely minor in driving the increase in extreme warming events. Nevertheless, even without any changes in AR frequency, ARs are more likely to induce extreme warming events under a warmer background temperature.

## 4 Conclusions and Discussions

Using hourly data from ERA5, we perform detailed analyses on the characteristics and drivers of the extreme warming events defined as a grid point with T2m >= 0 °C over the winter high Arctic. Based on ERA5, these events occur predominantly over the Atlantic sector. Except over a small region near 80°N between 0°-30°E, such extreme warming events occur, on average, less frequent than once in each winter. Consistent with in-situ observations (Moore, 2016; Graham et al., 2017), they tend to be short-lived, with a mean duration less than half day. Furthermore, these extreme warming events exhibit some degree of temporal clustering. The temporal clustering identified here could be caused by the clustering of weather systems, as has been found over mid-latitudes for cyclones and ARs (Pinto et al., 2013; Priestley et al., 2017; Fish et al., 2019, 2022). However, further research is still needed to identify whether similar clustering for cyclones and ARs occurs over the high Arctic. By examining their surface energy budget, the extreme warming events can be categorized into two different types: SHF dominance type, which occurs over regions equatorward about 83°N, and the DLW dominance type, which occurs over regions poleward of about 83°N. Notably, long-duration events (i.e., grid points with T2m >= 0°C for at least 40 hours), which occur over regions near 80°N, are mainly associated with persistent downward SHF anomalies. Composite analysis suggests that the position of the grid point experiencing extreme warming events relative to the negative SLP anomaly seems to play a key role in determining the event duration. Short duration events (i.e., grid points with T2m >= 0°C for 1 hour) are usually associated with a negative SLP anomaly located to their west. This spatial pattern leads to rapid cold advection after the T2m first exceeding 0°C (i.e., start of the events) and causes T2m to drop below 0°C. When the negative SLP anomaly is located

southwest of the grid point of the extreme warming event, creating sustained warm advection to the grid point even after the start of the event, it thus prolongs the event.

The large-scale circulation responsible for the occurrence of warming events with T2m >= 0°C over large areas of the Atlantic sector consists of a dipole pattern in the SLP anomalies, with a positive SLP anomaly over the northwest Eurasia and a negative SLP anomaly over Greenland. This dipole pattern can effectively channel heat and moisture into the high Arctic, resulting in a large-scale warming. K-means clustering applied to the spatiotemporal evolution of these concurrent warming events, defined as a contiguous region with T2m >= 0°C concurrently, further reveals that they mainly consist of three different types of SLP spatial patterns: dipole dominance type, anticyclone dominance type, and the cyclone dominance type. By steering cyclones into the high Arctic, the positive SLP anomaly which resembles blocking plays an important role in determining both the strength and spatial extent of the concurrent warming events. These large-scale circulations create an ideal environment for moisture intrusions into the Arctic. Using the Guan and Waliser (2019) AR detection algorithm, we show that ARs play a critical role in contributing to the occurrence of the winter high Arctic extreme warming events defined at the grid-point scale. Over most of the regions ever experienced extreme warming events, 100% of these events occur under AR conditions. The chance of having an extreme warming event can even become 50 times higher under AR conditions over some regions than otherwise. ARs are thus potent contributor to the occurrence of heat extreme over the high Arctic.

In the past four decades, the wintertime mean T2m over the high Arctic has been increasing significantly at a rate of 0.8 °C per decade. Concurrent with this rapid warming in the background temperature is the significant increase in both the frequency, intensity and the duration of extreme warming events defined at the grid-point scale and concurrent warming events. In addition, the spatial extent of the current warming events also exhibits a significant upward trend. In contrary to the significant background warming, despite their positive sign, trends in wintertime AR frequency are not yet significant due to the counterbalancing effect of changes in circulation and the moisture field. The increasing trends in the frequency and duration of wintertime extreme warming events are thus likely driven by the increasingly warming background T2m while the direct contribution from ARs is likely minor. Nevertheless, with continuously amplified warming over the wintertime Arctic and the projected increases in AR activities (Zhang et al., 2021a), the future wintertime high Arctic is expected to witness stronger, more frequent and long-lasting extreme warming events.

The current study does have several limitations. The high spatiotemporal resolution of ERA5 data provides an unprecedented opportunity to investigate the high Arctic wintertime extreme warming events. However, it is known that reanalyses are not real observations. They are produced by numerical models and constrained by limited observations through data assimilation. Biases relative to the actual observations are thus expected to exist in reanalysis products (Huang et al., 2017; Graham et al., 2019; Ma et al., 2024). For example, due to the misrepresentation of sea ice thickness and the absence of snow layer on top of sea ice in the numerical models, reanalysis products, including ERA5, suffer a warm bias over the wintertime ice-cover Arctic under radiatively clear condition (Batrak and Müller, 2019). In addition, given that in-situ observations over the Arctic, which are used to constrain reanalyses, are sparse, the representations of Arctic climate in reanalyses can be further degraded. This limitation calls for more field campaigns to observe the Arctic atmosphere. In-situ observations for the wintertime extreme

warming events and Arctic moisture intrusions or ARs are especially valuable in evaluating the representations of these events in reanalyses. Besides the potential uncertainty associated with the ERA5 dataset used here, the AR detection algorithm is another potential source of uncertainties for the results presented in section 3.4. It has been shown that there is a large spread in the detected AR statistics among major global AR detection algorithms participated in ARTMIP (Rutz et al., 2019; Lora et al., 2020). The AR detection algorithm (Guan and Waliser, 2015; 2019) used in this study is one of the very few global AR detection algorithms that can detect noticeable occurrences of ARs over the Arctic. This algorithm is thus recommended by the ARTMIP community for studying high-latitude ARs. For future research focusing on Arctic ARs, intercomparison studies are especially needed to better understand the Arctic AR uncertainties due to AR detection algorithms and/or datasets. The results presented in this study can serve as a good starting point for addressing the limitations discussed above.

Given the critical roles played by the SLP dipole and ARs in determining the occurrence and characteristics of the wintertime extreme warming events, it is important to understand their variability at different timescales and identify large-scale climate modes that are responsible for such variability. An improved understanding on the variability of the SLP dipole and ARs would likely lead to a better understanding and prediction of the Arctic climate across timescales. The results in this study also suggest that a correct representation of the SLP dipole and ARs is key to simulating the occurrence of extreme weather events over the high Arctic at synoptic timescale. As we rely on climate models for future Arctic climate projection, further research is needed to evaluate how well climate models can faithfully represent the SLP dipole and ARs that affect the high Arctic. In this study, we focus on the T2m extreme warming events. It is expected that warming events as such would have a considerable impact on the underlying sea ice. Links between March persistent atmospheric circulation and September minimum sea ice extent (Kapsch et al., 2019) and between warm winter and subsequent thinner spring sea ice over the Arctic (Stroeve et al., 2018) have been established by previous studies. As have been shown in this study, these extreme warming events tend to cluster in time. It would be interesting to further investigate their cumulative effects on the longer-term sea ice growth and the subsequent sea ice melt in the following summer. If a link between the occurrence frequency of such warming events and the subsequent summer sea ice minimum can be established, an improved prediction of the SLP dipole and ARs mentioned above would likely further extend the prediction lead time of summer sea ice minimum.

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

**Data Availability**

ERA5 data can be found at https://www.ecmwf.int/en/forecasts/dataset/ecmwf-reanalysis-v5

**Author Contributions**

710   W.M. conceived the study. W.M. designed the study with contributions from H.W. W.M. performed the analyses and wrote the initial draft of the paper. All authors contributed to interpreting the results, editing, and revising the manuscript.

**Competing Interests**

At least one of the co-authors is an editor of Atmospheric Chemistry and Physics.

**Acknowledgements**

This research was supported by the U.S. Department of Energy (DOE), Office of Science, Office of Biological and Environmental Research, Regional and Global Model Analysis program area, as part of the HiLAT-RASM project. This research used resources of the National Energy Research Scientific Computing Center (NERSC), a U.S. DOE Office of Science

User Facility operated under Contract No. DE-AC02-05CH11231. The Pacific Northwest National Laboratory (PNNL) is operated for DOE by Battelle Memorial Institute under contract DE-AC05-76RLO1830. G.C. is supported by the U.S. NSF grant AGS-2232581 and NASA grant 80NSSC21K1522.

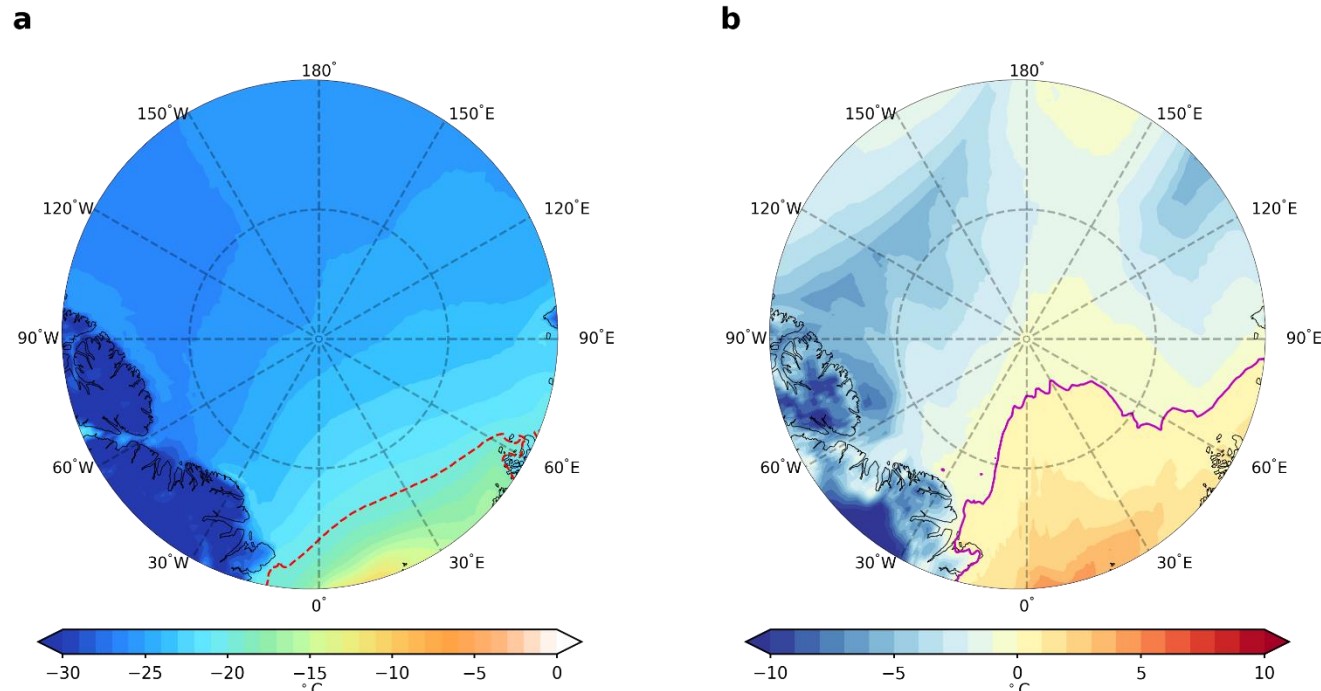

**Figure 1.** (a) Time mean 2-meter air temperature (T2m) and (b) maximum hourly T2m over all winter from 1979 to 2021 in ERA5 reanalysis.
The red dashed line in (a) denotes -20℃. The purple line in (b) denotes **0**℃.

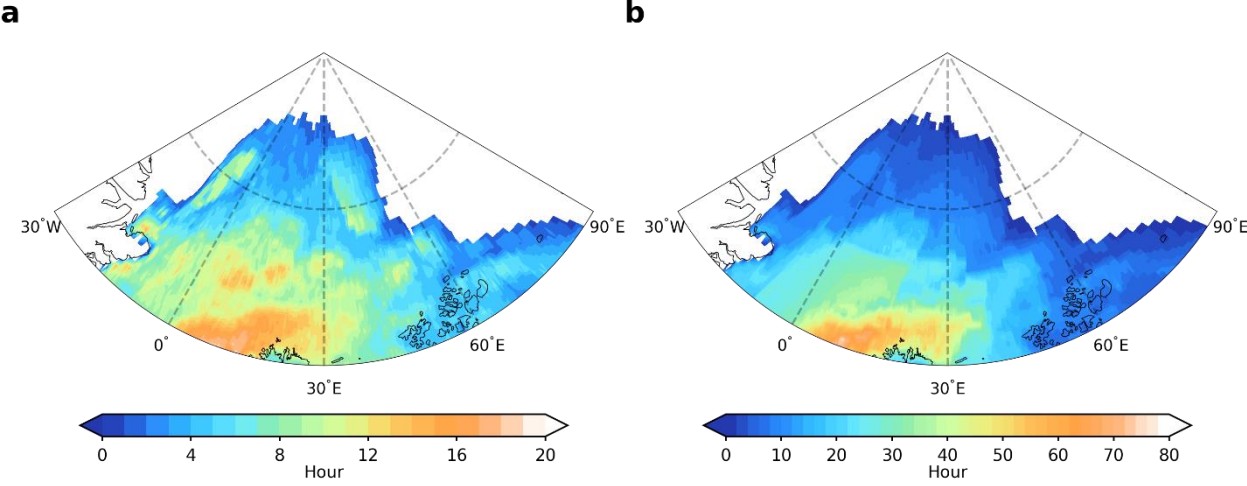

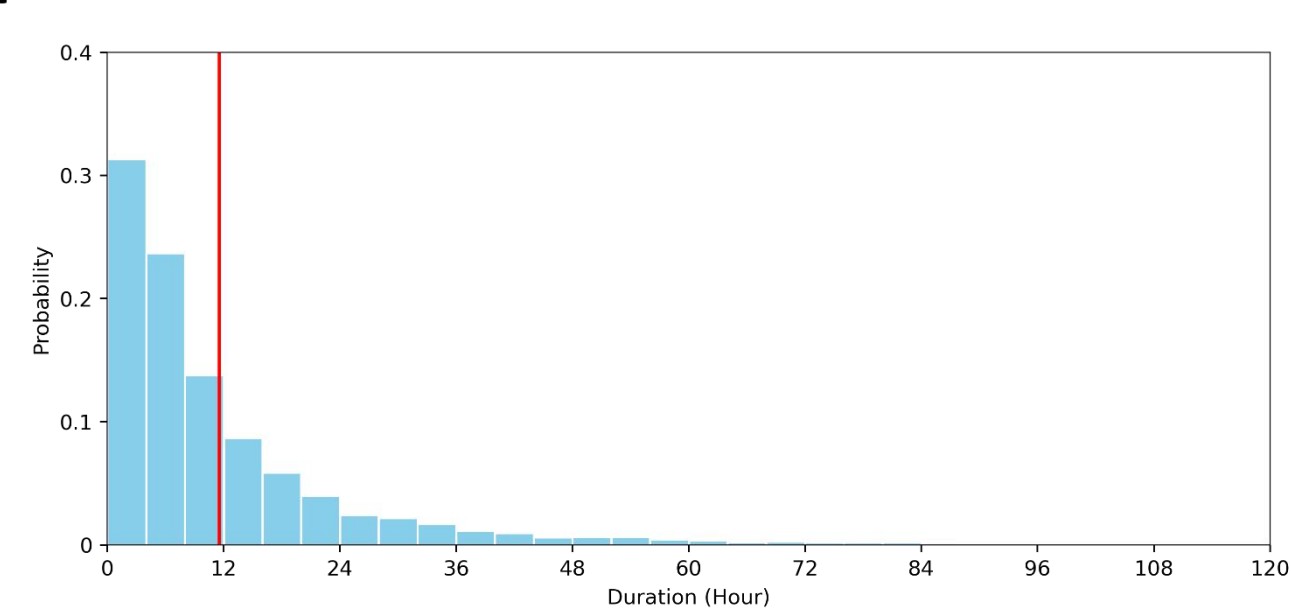

**Figure 2**. Spatial distribution of the (a) mean duration and (b) extreme duration of the high Arctic extreme warming events defined as a grid point with T2m >= 0 °C. Extreme duration is defined as the local 95th percentile of the duration distribution. (c) Probability density function of the duration distribution for all events happened during winter from 1979 to 2021. The red vertical line in (c) marks the mean of the distribution (11.55 hours).

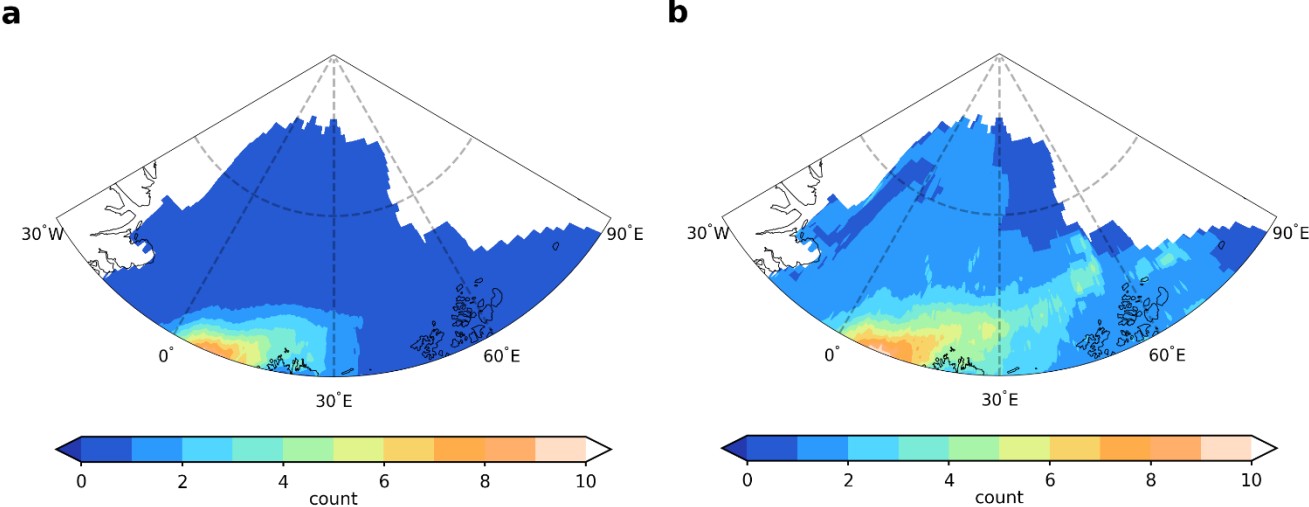

**Figure 3**. Spatial distribution of (a) the seasonal mean event count for all winters and for (b) only those winters with at least one extreme warming event from 1979-2021.

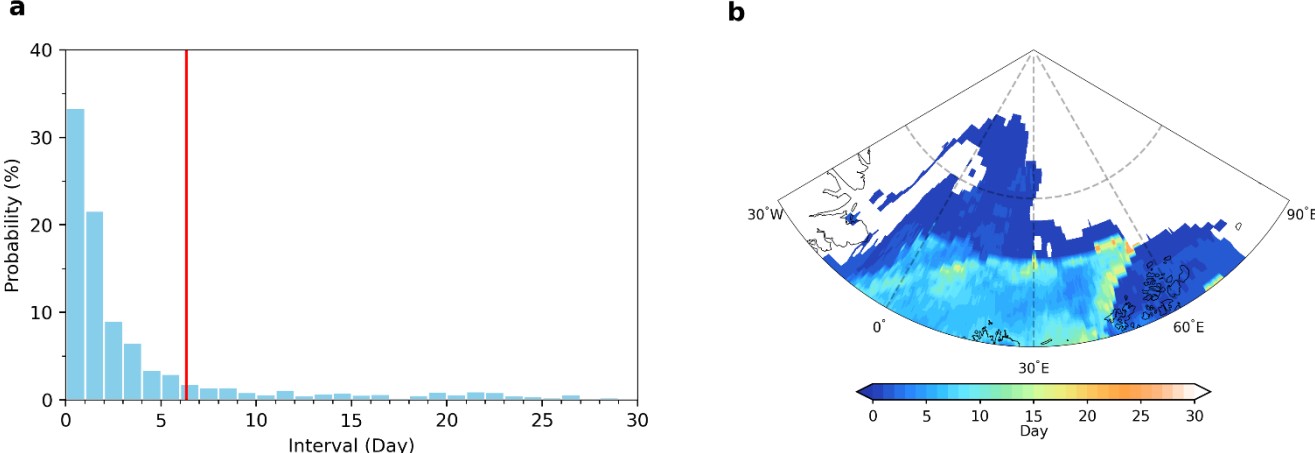

**Figure 4**. (a) Probability density function of the time interval between the termination of an extreme warming event (i.e., time when a grid point's T2m first falls below 0℃) and the start of the following extreme warming event (i.e., time when a grid point's T2m first exceeds 0℃) over the same grid point. (b) Spatial distribution for the mean time interval. Grid points never experienced an extreme warming event or less than two extreme warming events within a single winter (thus with a mean time interval of zero) have been masked out in (b) and excluded from (a). The red vertical line in (a) marks the mean of the distribution (6.32 days).

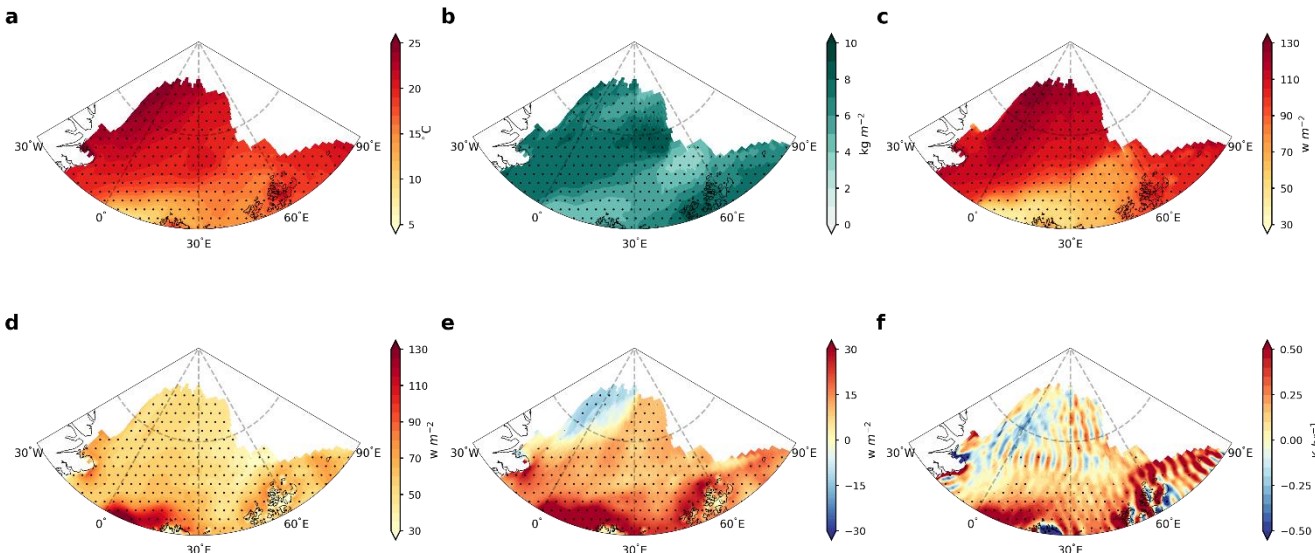

**Figure 5**. Spatial distribution of the mean anomalies of (a) T2m, (b) column-integrated water vapor (IWV), (c) downward longwave radiation (DLW), (d) sensible heat flux (SHF), (e) latent heat flux (LHF) and (f) horizontal temperature advection averaged over all hours with T2m >= 0 °C over a grid point. Positive values in (d) and (e) indicate fluxes directed from the atmosphere toward the surface. Stippled areas indicate that anomalies are significant at the 0.05 level based on the Student's t-test.

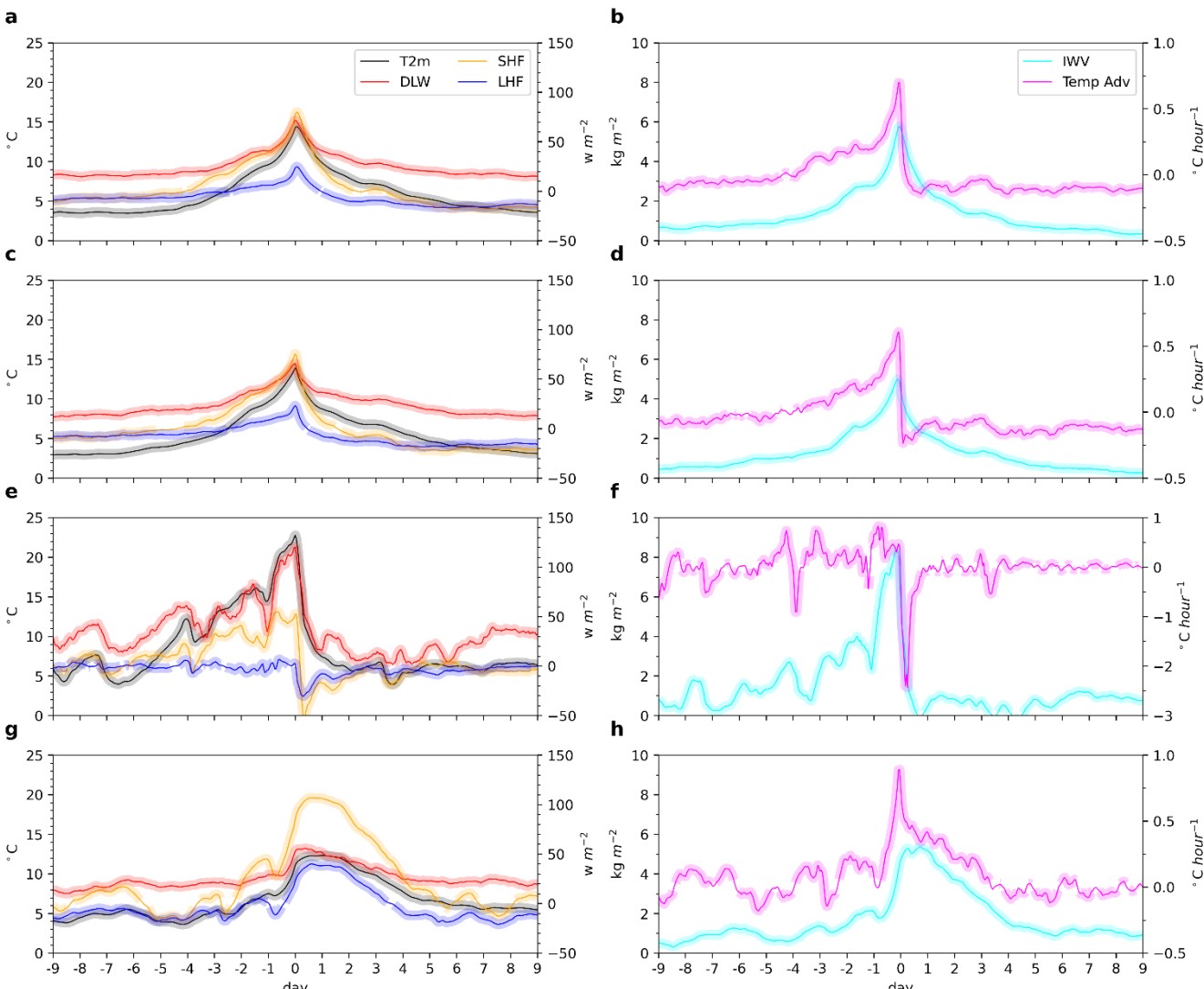

**Figure 6**. Temporal evolution of the anomalies of T2m, DLW, SHF, LHF, IWV and temperature advection for all the extreme warming events defined as any grid points with T2m >= 0 °C (a, b), short duration events (i.e., grid points with T2m >= 0℃ for 1 hour) equatorward of 83°N (c, d), short duration events poleward of 85°N (e, f), and long duration events (i.e., grid points with T2m >= 0°C for at least 40 hours; g, h). Note that long duration events occur only over regions equatorward of 83°N. These curves are constructed by averaging the temporal evolution of various anomaly terms across all extreme warming events within the respective groups. There are 191555, 18586, 1642, and 10097 events included in the groups of all events, short duration events equatorward of 83°N, short duration events poleward of 85°N and long duration events, respectively. Day 0 corresponds to the start of an extreme warming event when a grid point's T2m first exceeds 0℃. The shading indicates that the anomalies are significant at the 0.05 level based on the Student's t-test.

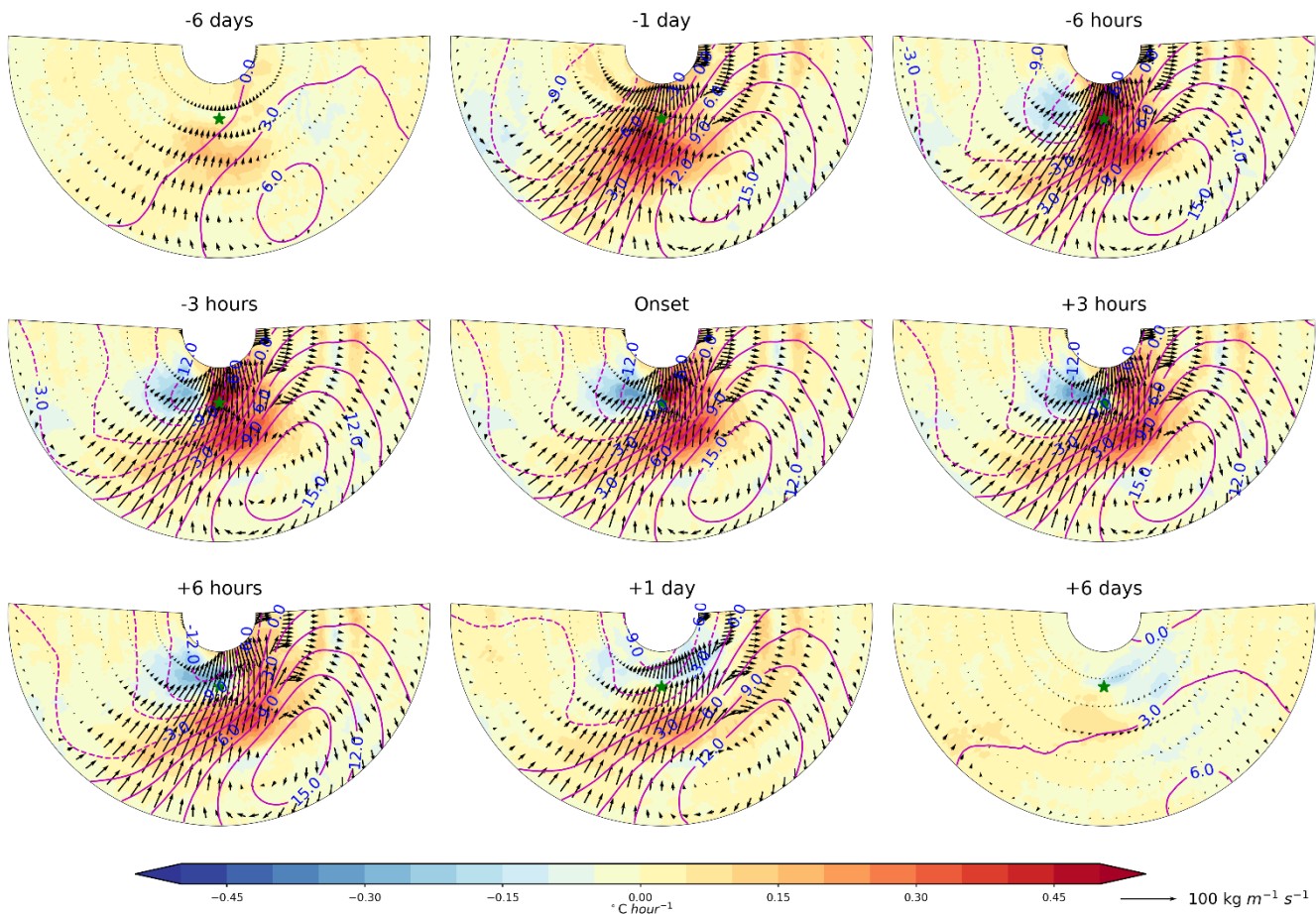

**Figure 7**. Composites centred at the event grid point for the temporal evolution of integrated water vapor transport (IVT) anomalies (vectors), sea level pressure (SLP) anomalies (lines) and temperature advection anomalies (shading) before, during and after the start of the short duration extreme warming events (i.e., grid points with T2m >= 0°C for 1 hour) equatorward of 83°N. The green star in each panel indicates the grid point where the extreme warming events took place. Regions 5° poleward, 20° equatorward, 100° westward/eastward of the event grid point are included in the composites.

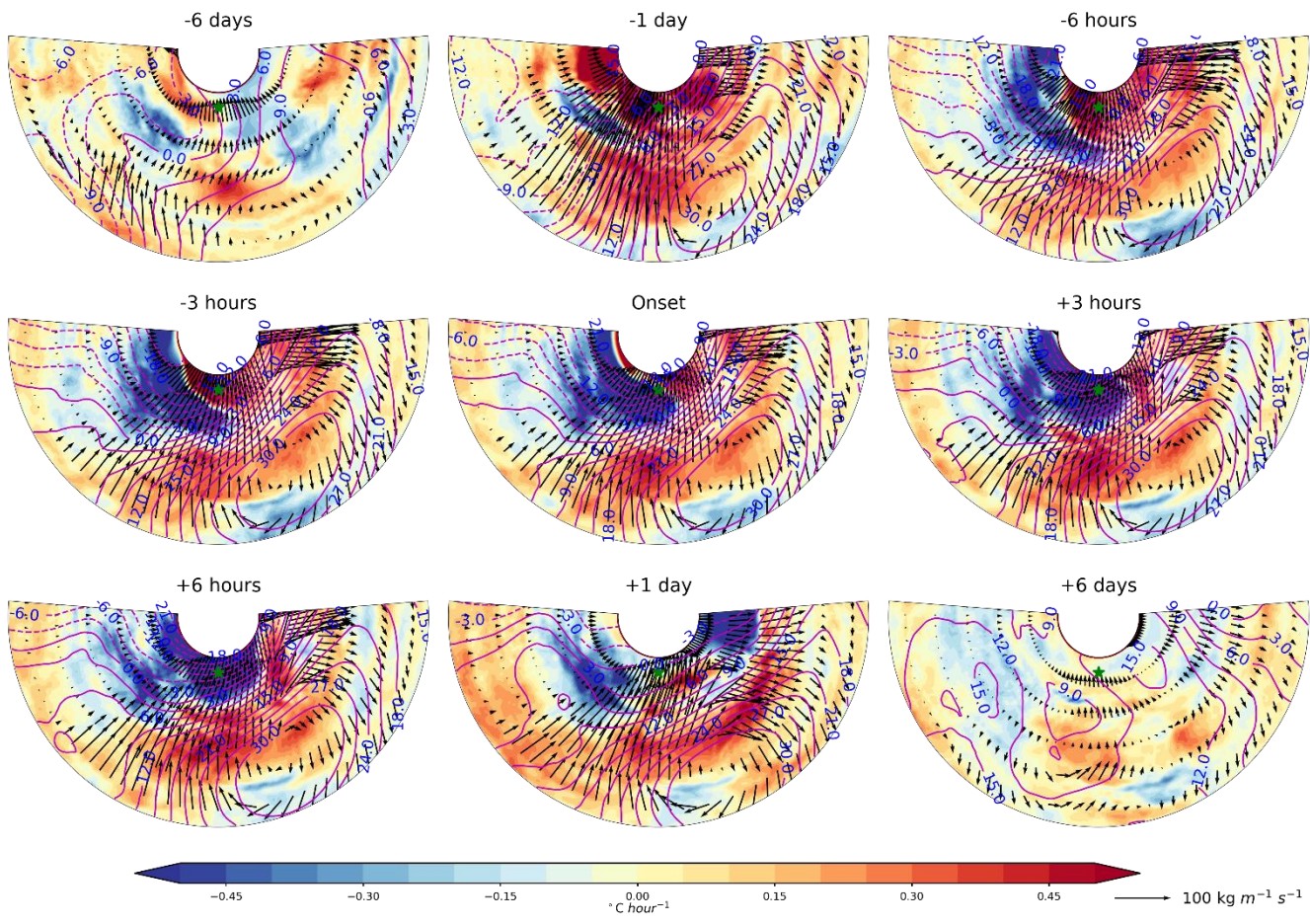

**Figure 8**. Same as Fig. 7, but for the short duration events (i.e., grid points with T2m >= 0°C for 1 hour) poleward of 85°N. Regions 2° poleward, 20° equatorward, 100° westward/eastward of the event grid point are included in the composites.

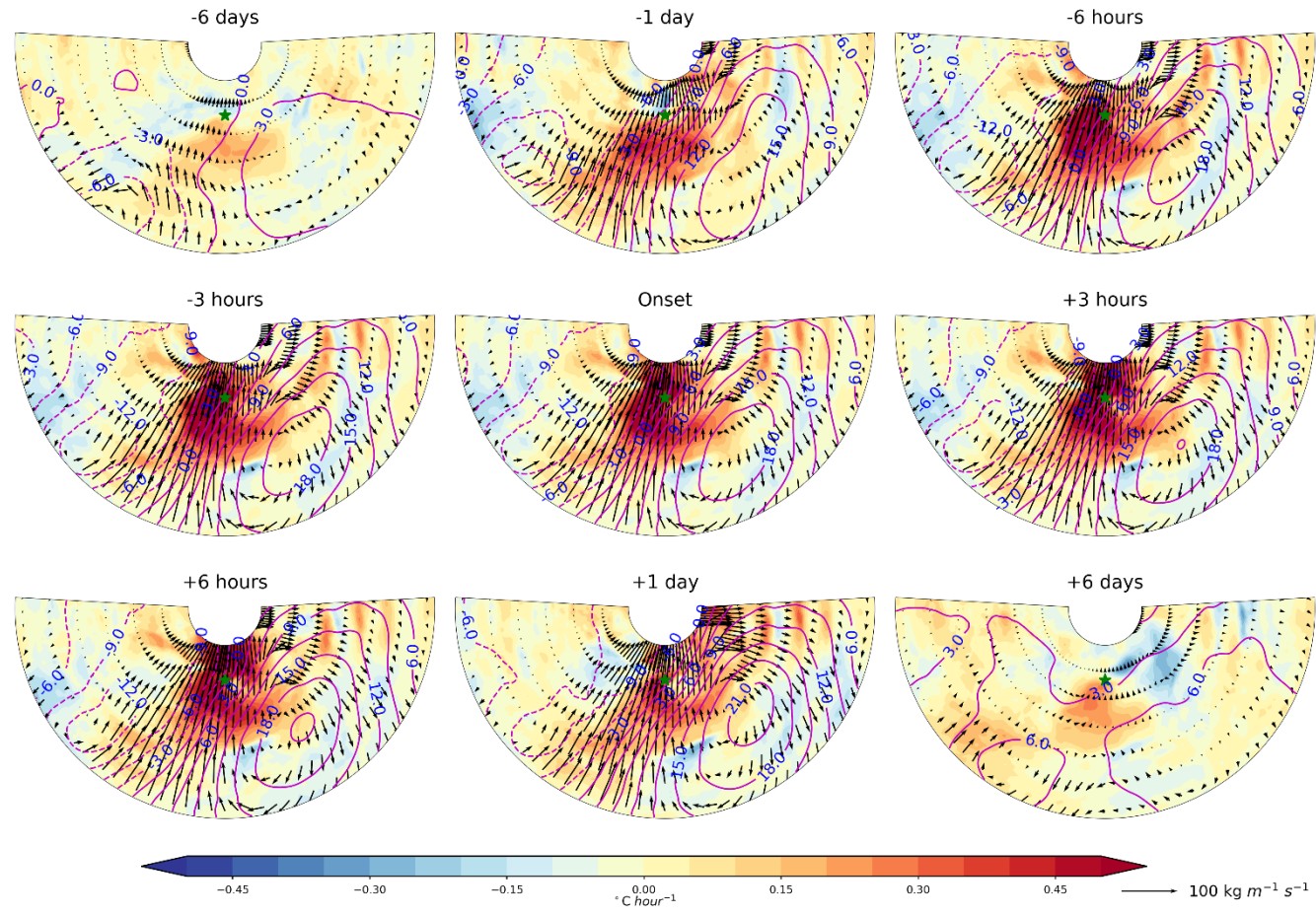

**Figure 9**. Same as Fig. 7, but for the long duration events (i.e., grid points with T2m >= 0°C for at least 40 hours).

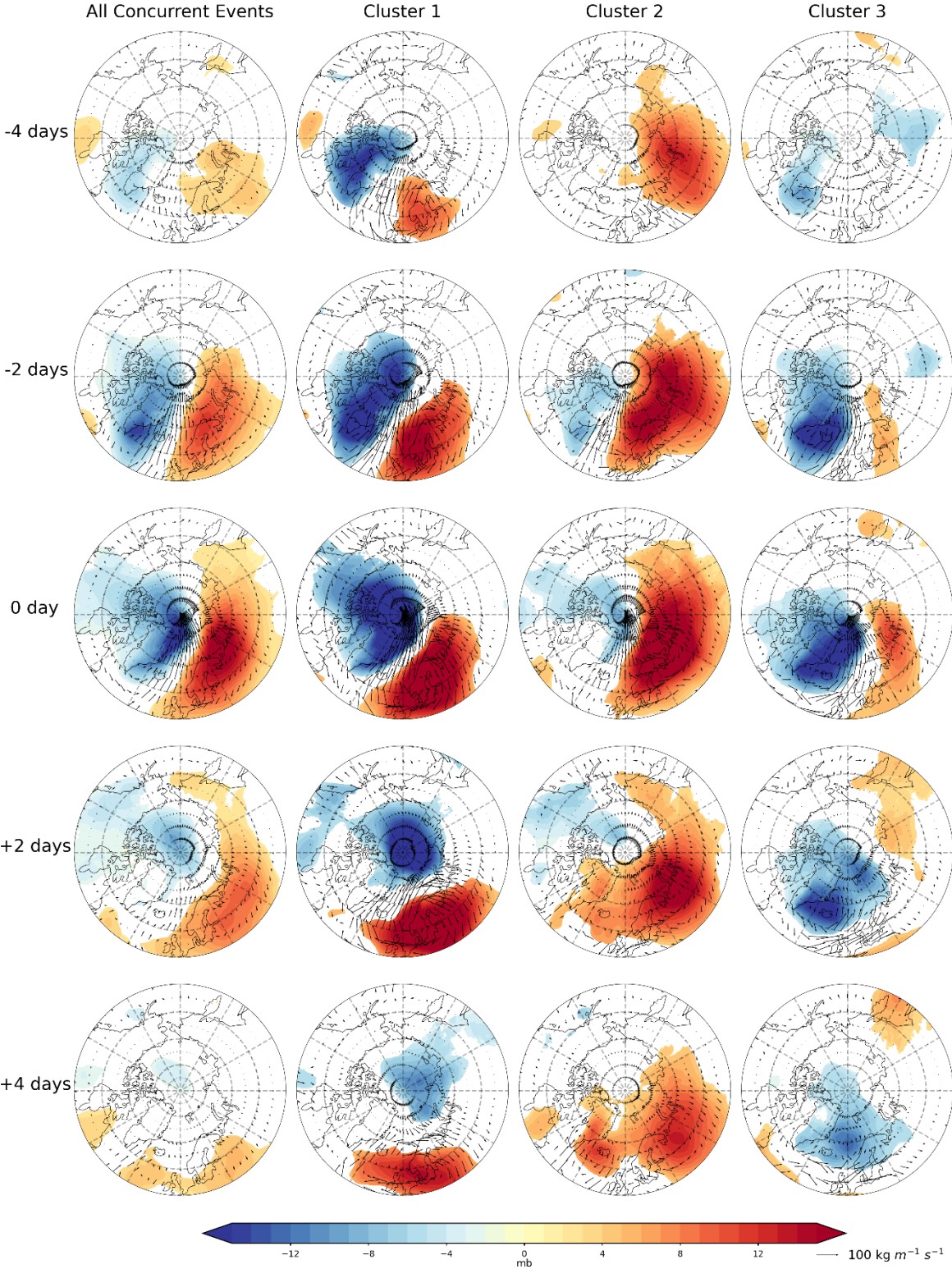

**Figure 10**. Temporal evolution of the large-scale circulation associated with the concurrent warming events. The shaded contours show the SLP anomalies, and the vectors represent the IVT anomalies. The 1st column describes the composites for all the concurrent warming events. The 2nd, 3rd, and 4th columns show the composites for the 1st, 2nd, and 3rd cluster, respectively, obtained from K-means clustering. There are 96 events identified in total. The number of events in each cluster is 22, 39, and 35 for the 1st, 2nd, and 3rd cluster, respectively. Only anomalies that are significant at the 0.05 level based on the Student's t-test are shown. See the definition of concurrent warming events in the text. Day

0 indicates the time with the largest area where temperature exceeds 0℃.

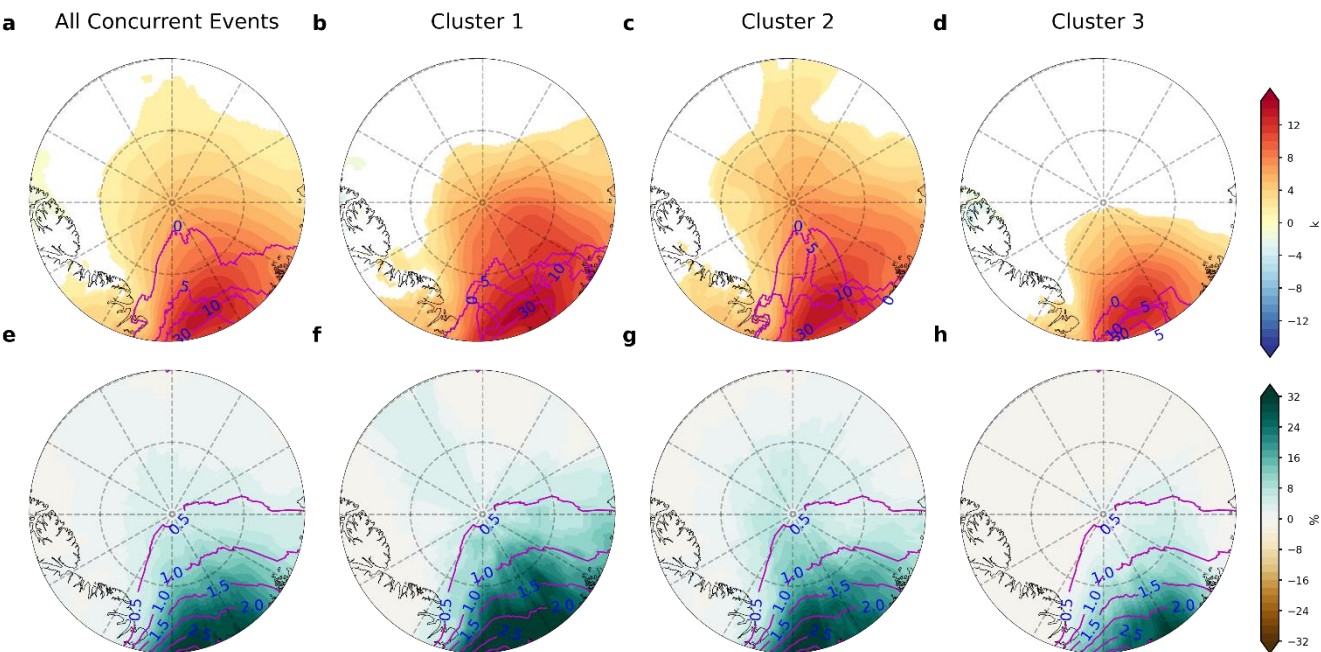

**Figure 11**. Same as Fig. 10, but for the T2m anomalies and AR frequency during the peak of the concurrent warming events. The peak of the concurrent warming events is defined as the time when the areas with temperature above 0℃ reach maximum. The purple line contours in (a)-(d) depict the fraction of time when the T2m over a grid point reaches or exceeds 0℃ during the peak of the concurrent warming events and (e) – (h) the climatology of winter AR frequency.

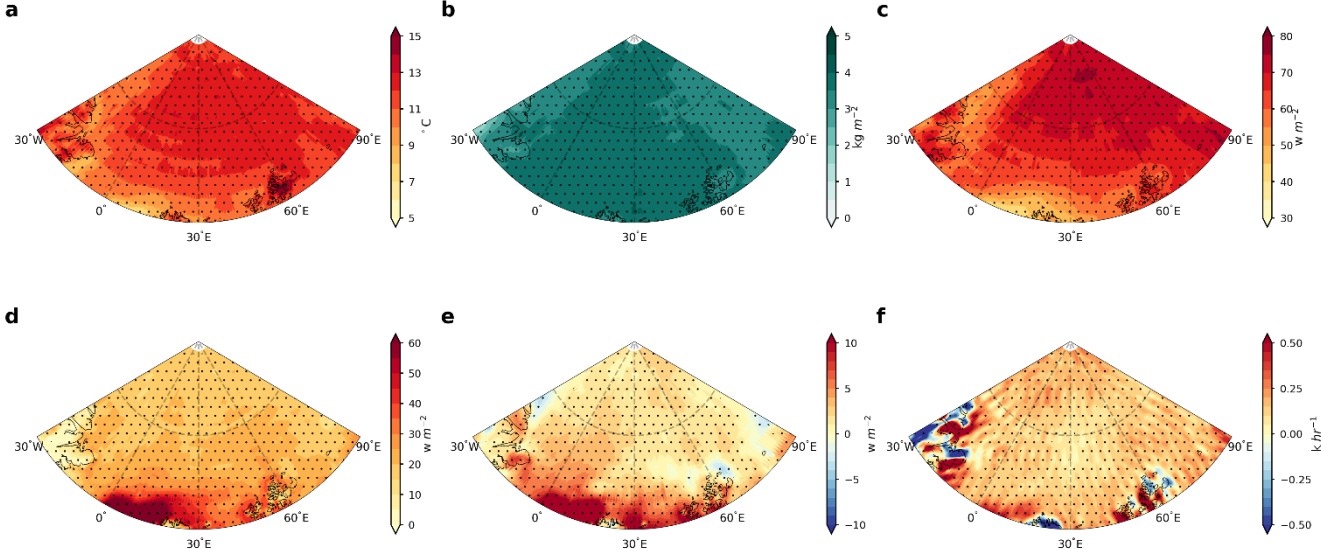

**Figure 12**. Spatial distribution of the anomalies of (a) T2m, (b) IWV, (c) DLW, (d) SHF, (e) LHF and (f) temperature advection during AR days. Stippled areas indicate anomalies are significant at the 0.05 level based on the Student's t-test.

**Figure 13**. Spatial distribution of (a) the fraction of extreme warming events defined as a grid point with T2m >= 0 °C that occurs during AR days, (b) the fraction of time for all winter hourly snapshots from 1979-2021 with T2m above 0℃, (c) the fraction of time for all AR day hourly snapshots from 1979-2021 with T2m above 0℃, and (d) the risk ratio, which is calculated by dividing (c) by (b).

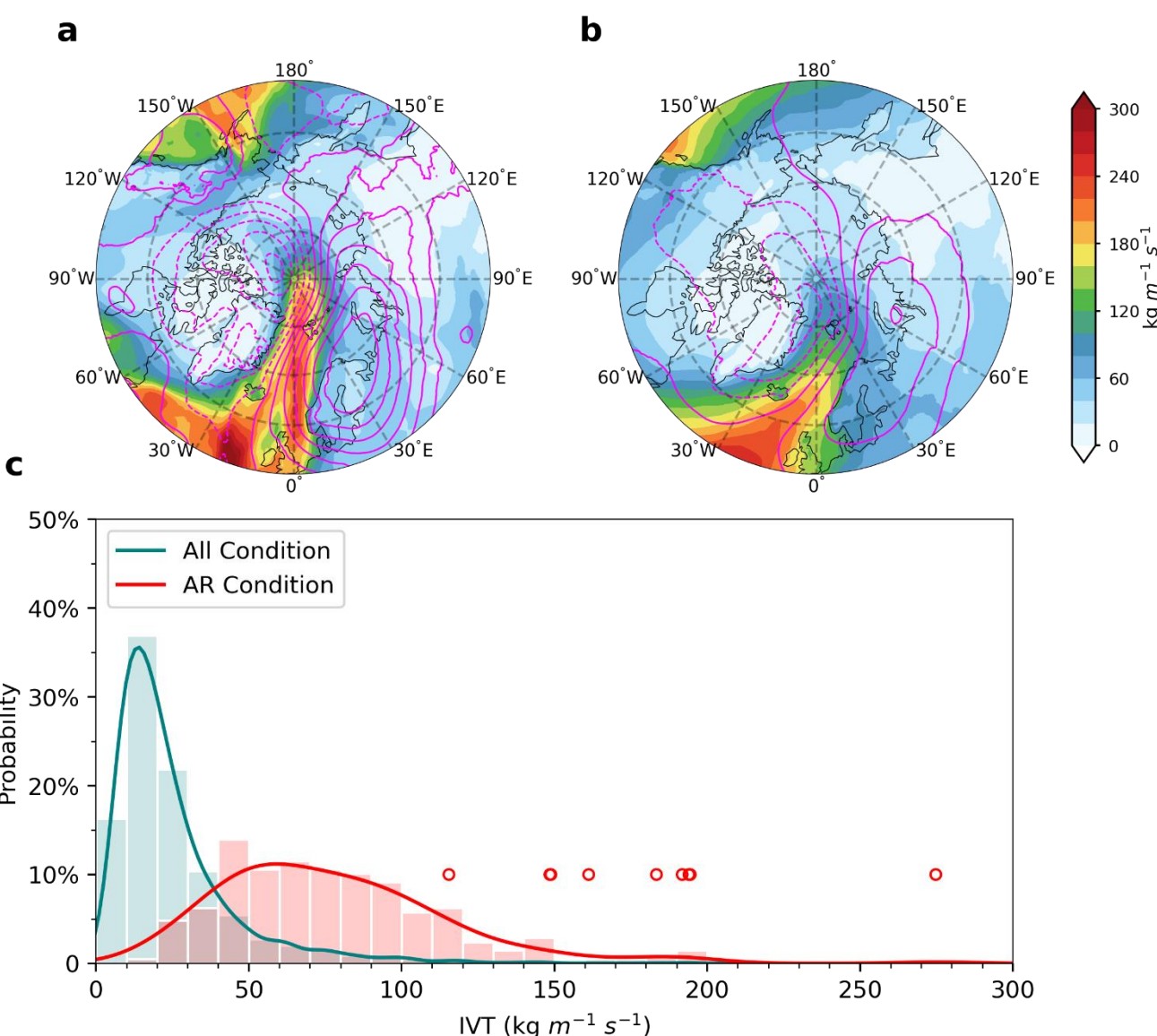

**Figure 14**. Composites of IVT (shaded contours) and SLP anomalies (line contours, solid lines denote positive values while dashed lines denote negative values) for (a) the nine days with extreme warming events occurring poleward of 85°N, (b) all the days with ARs intercepting 85°N between 15°W-60°E, but no extreme warming events found poleward of 85°N. (c) Probability density function of the daily IVT averaged over regions poleward of 85°N and between 15°W-60°E. The green bars/line are for all the winter days from 1979-2021, red bars/line are for

the days with ARs intercepting 85°N between 15°W-60°E. The open circles are for the nine days with extreme warming events occurring poleward of 85°N.

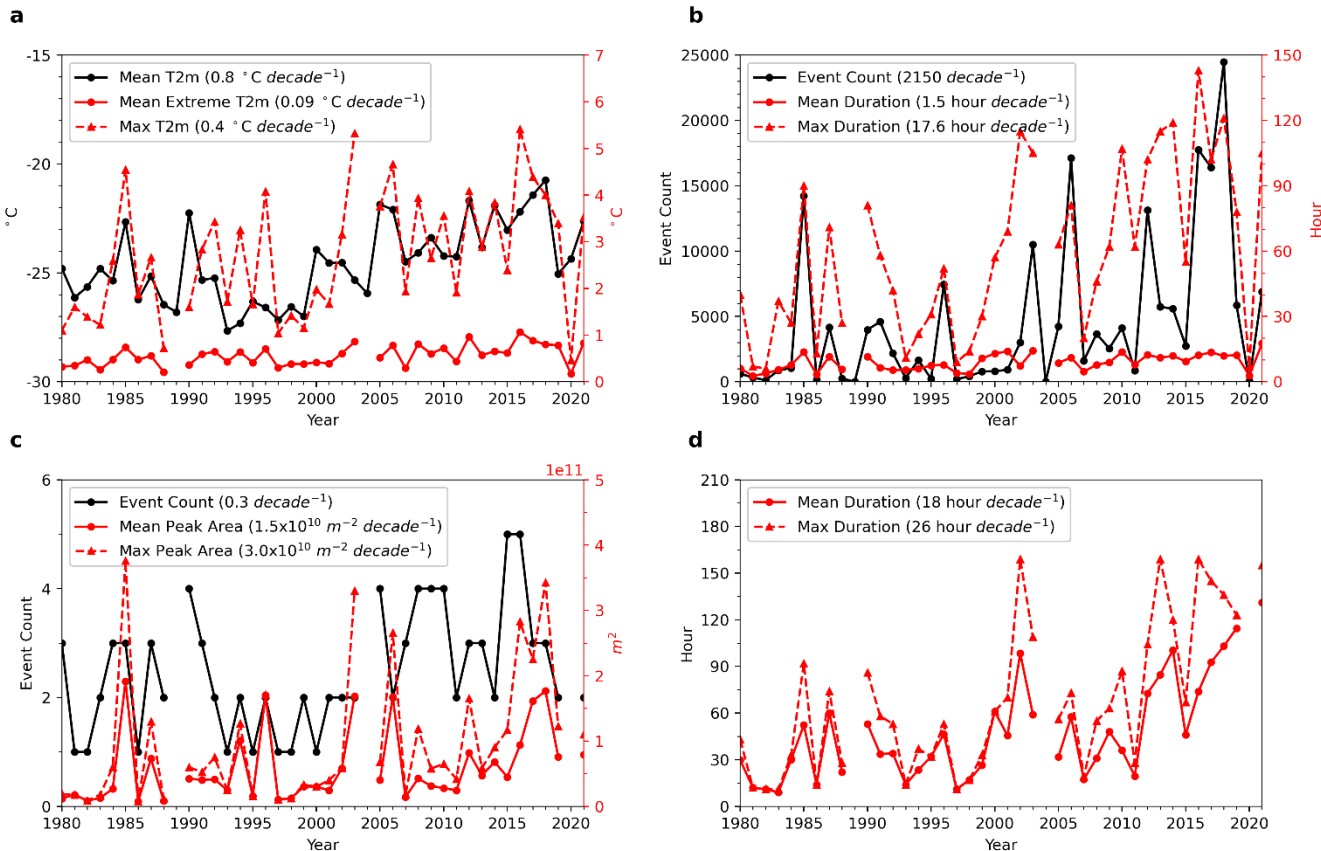

**Figure 15**. Trends in (a) the area-weighted spatial mean T2m over the entire high Arctic (black solid line with circles), mean T2m only for grid points above 0℃ (red solid line with circles) and the seasonal maximum hourly T2m (red dashed line with triangles) over the high Arctic. (b) is the same as (a), but for the trends in extreme warming event count (black solid line with circles), mean event duration (red solid line with circles) and seasonal maximum event duration (red dashed line with triangles). (c) and (d) show trends in the characteristics of the 96 concurrent warming events identified in Section 3.3. Trends in (c) the event count (black solid line with circles), mean peak area (solid
red line with circles) and seasonal maximum peak area (red dashed line with triangles) of the concurrent warming events. (d) shows the mean duration (red solid line with circles) and seasonal maximum duration (red dashed line with triangles) of the concurrent warming events.

For those seasons with only one concurrent warming event, the mean peak area would be the same as the maximum peak area in (c) and the mean duration would be the same as the maximum duration in (d). All trends are significant at the 0.05 level based on the Student's t-test.
