# Peer review of "Wintertime Extreme Warming Events in the High Arctic: Characteristics, Drivers, Trends, and the Role of Atmospheric Rivers"

_EGUsphere, 2023_

## Referee Comment (RC1)

Review of "Wintertime Extreme Warming Events in the High Arctic: Characteristics, Drivers, Trends, and the Role of Atmospheric Rivers" by W. Ma et al.

General comments:

This study investigates the characteristics of extreme wintertime Arctic warm events, using hourly ERA5 data for the period 1979-2021. In most of the study, Arctic warm events are defined as grid points poleward of 80°N where the two-meter air temperature (T2m) exceeds 0°C. Adjacent grid points with T2m > 0°C at the same time are considered as separate warm events. They find that the events are rare and only occur over the Atlantic sector of the Arctic, with a mean duration of less than half a day. Warm events are associated with positive anomalies in integrated water vapor, downward longwave radiation, sensible heat flux and, in most areas, latent heat flux. They are located in a region of intense sea level pressure (SLP) gradients, with a negative SLP anomaly to the west and a positive anomaly to the southeast, and they typically coincide with atmospheric rivers. In an additional analysis of so-called "concurrent warm events", where events are defined as coherent objects and not as single grid points, they use a K-means clustering method and find three different large-scale circulation clusters, with the first characterized by a strong SLP dipole, the second mainly by a strong surface anticyclone and the third mainly by a strong surface cyclone. Finally, they show positive trends in the number of events (again defined as single grid points with T2m > 0°C), their magnitude and duration over the past 40 years.

Overall, the manuscript is interesting, and the study may be well suited for publication if a couple of scientific comments are addressed, and some formulations are further improved. Some specific suggestions are given below.

Specific comments:

1) My major comment concerns your definition of Arctic warming events. You call it a warming event when T2m exceeds 0°C at a grid point. If T2m is > 0°C at several neighbouring grid points at the same time, you refer to them as several separate warming events. I find this quite confusing, as it is most likely just one event with a larger spatial extent. You then write that the duration of the events is often less than 1h (your "short-duration events"), but I assume that they are typically much longer-lived when they can move over several grid points. And according to your description of Fig. 5, the temperature drops very quickly after an Arctic warm event, but this is just the temperature at a specific grid point. The composites in Fig. 6-8 indicate that the warm temperatures are simply being advected to other grid points. I assume there is a lot of double counting in the different figures (for instance, the fields in the composites are considered several times if a warm event spans several grid points). I think you have to be very careful with the interpretation of your results and with the wording. Rather than "warming events", I would typically write something like "grid points with T2m > 0°C" to avoid confusion, and mention where necessary that the fields are considered several times when a warm episode covers more than one grid point
I find the definition of the "concurrent warming events" much more convincing. If I understood it correctly, there you considered adjacent grid points with T2m > 0°C as one "warming event" and also somehow traced the events in time. How is the tracing working exactly, can the warm events move over several grid points? It would be interesting if you could provide some information about their occurrence frequency in the past 40 years, their duration, their spatial extent and other characteristics. And maybe you do not need to impose a threshold on the area and the duration of these events, as you do so far, but could investigate the characteristics of all of them.

2) Abstract and section 4: Please specify that you define an "Arctic warming event" as a grid point with T2m > 0°C.

3) Lines 15-16: "… with a seasonal occurrence frequency of less than one over most of the regions." Could you be more specific about what you mean with "less than one"? Instead of a decimal number per season, which might be a bit difficult to grasp, it would be more informative if you could write the absolute number of events with T2m > 0°C in the considered time period. As you define the events per grid point, you could provide a mean number over all grid points (or two separate numbers for more southern and more northern latitudes).

4) Lines 69-70: "Subsequent examination reveals … with the duration of staying above 0°C for less than an hour." Which study are you referring to? According to Fig. 1 in Binder et al. 2017 (Geophys. Res. Lett.), maximum temperatures reached values above 0°C during three episodes that each lasted about 1 day, i.e., much longer than 1 hour.

5) "This event is driven by an AR-like moisture plume carried into the high Arctic by a cyclone." Please indicate which study you are referring to in this sentence.

6) Please also cite the paper by Messori et al. 2018, J. Climate (doi: 10.1175/JCLI-D-17-0386.1), which also investigates drivers of wintertime Arctic warm events.

7) Lines 90-92: From the sentence it is not clear based on which dataset you define the extreme warming events. Is it based on buoy observations, as suggested by the title of the subsection and the first sentence, or based on ERA5, which you mention later in the same subsection? If – as I assume – you use ERA5 data to define the events, then I find the title and the first sentence of the subsection rather confusing. Also, in that case I would first describe the ERA5 data in a paragraph and mention the temporal and spatial resolution, the season, the time period you investigate, etc., and only then describe how you identify the extreme warming events. On the other hand, if you use buoy observations, please describe this dataset in more detail. How do you obtain gridded data from buoy observations (Fig. 2)? Furthermore, in that case I would make a separate subsection for the description of the ERA5 data.

8) Line 101: "Results based on previous studies …" Please add references. And maybe this sentence fits better in the introduction than the method section.

9) Lines 113-115: How do the trends in DLR, IVT, LHF, SHF and IWV look like, are they all positive? How does the detrending work?

10) Line 158: "Over regions eastward of about 60°N and poleward of about 85°N, the mean duration is shorter than 5 hours." How do you arrive at this statement? There is no data in this region in Figs. 2a,b, right?

11) Lines 165-171: Maybe you can add that it is very likely that two warming events that occur right after each other are driven by the same weather system (as you already pointed out in section 2.1).

12) Fig. 4: What do the fields show exactly? Is it the anomaly of the mean value of IWV, DLR, etc., averaged over all warm events at a specific grid point? And maybe you can add in the caption that positive values in d) and e) indicate fluxes directed from the atmosphere toward the surface.

13) Line 185: "due likely to the partially open ocean" Would it be possible to overlay a contour of the mean position of the sea ice edge during warm events and in the climatology?

14) Line 190-191: "likely caused by the rapid cold and dry advection shortly after the onset of warming events" Do you know why there is cold and dry advection in this region after the onset of the warming events?

15) Fig. 5: Which one is the curve for T2m, is it the black line labelled "TS" in the legend? If so, please adapt it in the legend. And how are the curves constructed? Do you show the anomaly of the mean

T2m, DLW, etc. over all grid points with T2m > 0°C? If so, please be more specific in the figure caption. Also, as mentioned in the first comment, I find it a bit confusing that you write in the caption "for all the warming events", I would rather write something like, "at grid points where T2m > 0°C", because most likely neighbouring grid points with T2m > 0°C at the same time are not separate warming events.

16) Line 202-203: "These results suggest that a warm and moist winter favours the occurrence of warming events." It could also be the other way around: the warm events contribute to making the winter anomalously warm and moist. Maybe the anomalously high temperatures nine days before the onset of the extreme warming event are related to a previous warm event in some region of the Arctic. For this, a timeline of the number of warm events per season could be interesting.

17) Line 204-205: "Both the SHF and DLW play comparable roles in driving these events, …" I find it confusing that you write that SHF and DLW "drive" the Arctic warm events. I assume that it is mainly the large-scale flow configuration that drives these events, i.e., the dipole in the SLP anomalies that you see in your composites (Fig. 6 and 9) and the upper-level flow structure (as has also been described in previous studies, e.g., Binder et al. 2017, Messori et al. 2018). Also, in the conclusion (line 369), you write that long-duration events are mainly driven by persistent downward SHF anomalies. Maybe you can just write "associated with" instead of "driven by".

18) Figs. 6-8, picking up on the first comment: Since neighbouring grid points with T2m > 0°C at the same time are considered as separate events, is it right that they are included several times in your composites? Could it then just be that your "short-duration events" are the grid points located closest to the cold front of the cyclone and the "long-duration events" are grid points located further to the east in the cyclone's warm sector, but they actually occur at the same time and belong to the same warm episode? This indicates that your definition of "warm events" might be problematic.

19) Line 228: "a high anomaly to their southeast and a low anomaly to their west." I guess you mean anomalies in SLP, please write this explicitly. And I would write positive/negative (rather than high/low) SLP anomalies (also in the rest of the manuscript).

20) Fig. 7: It is difficult to distinguish the IVT anomaly vectors and the SLP anomaly contours. Maybe you can change the colour of either of them? And it appears that there should be more SLP anomaly contours above +21 hPa, but the plotting stops at this threshold, which looks a bit strange.

21) Line 240-242: "… the high anomaly southeast of the event regions already starts to develop for the long duration events (Fig 8)." Please specify which time step you are referring to.

22) Line 317-318: "… for a large fraction of the regions where warming events can occur, ARs are the only weather system capable of triggering the occurrence of the warming events." Fig. 12a simply tells you that warm events typically co-occur with ARs, but I don't agree with your statement that they are the only weather systems that can trigger warm events. In contrast, in most cases it is probably the interplay between various weather systems that is important for triggering the event (like, for instance, the surface cyclones to the west and the anticyclones to the east, which channel the poleward heat and moisture transport, as well as geopotential height anomalies at upper levels, etc.). The AR can only reach the Arctic because of this interplay between various weather systems.

23) Fig. 13c is not mentioned in the text.

24) Fig. 14: Is mean T2m averaged over the entire Arctic poleward of 80°N or only the Atlantic sector that you consider in the rest of your study?

25) Line 342-343 and Fig. 14b: "The event occurrence frequency has been increasing at a rate of 2150 events per season per decade." Here it would again be helpful if you could specify that you mean the number of grid points with T2M > 0°C. This number of course strongly depends on the grid

spacing. I would find it more meaningful if you showed a timeline of the number of days per season (or the number of hours) where the temperature exceeded 0°C in some region of the domain, or a timeline of the number of your so-called "concurrent warming events".

Technical corrections:

26) Line 14: "over the high Arctic (poleward of 80°N) occurred during 1980-2021"
Typo: "that occurred"

27) Line 16: "regions" – maybe better: "region"

28) Lines 41-44: "ranging from … and heatwaves" should be "ranging from … to heatwaves"

29) Line 59: "phenomenon" should be "phenomena"

30) Line 68: "an episode of extreme warming event" should either be rephrased to "an episode of extreme warming" or "an extreme warming event"

31) Fig. 3: "The red vertical line in (c) …" You probably mean (a). And I think you can remove the "6.32" in the top left corner of Fig. 3a (or move it to the right to the position of the red line).

32) Throughout the manuscript: I would place the references in chronological order.

33) Line 301: "likely plays roles" – maybe better: "likely plays an important role"

34) Line 339 and caption Fig. 14: "for those above 0°C" should be "for grid points above 0°C"

35) Line 370: "experiencing warming event" should be "experiencing a warming event"

36) Line 371: "to its west" should be "to their west"

37) Line 373: "located at southwest of the grid point with warming event" should be "located southwest of the grid point of the warming event"

38) Line 379: I would write "anticyclone dominance type" and "cyclone dominance type" instead of "high/low dominance type"

39) Line 398: "are thus can be expected": delete either "are thus" or "can be"

40) Line 403: "AR detection algorithm" should be "the AR detection algorithm"

---

## Referee Comment (RC2)

**Review notes for the manuscript to ACP: "Wintertime Extreme Warming Events in the High Arctic: Characteristics, Drivers, Trends, and the Role of Atmospheric Rivers" by Ma et al.**

**General comments**

This paper presents characteristics of extreme wintertime Arctic extreme events for 1979 – 2021 utilizing hourly ERA5 data. Extreme warm events are defined per grid-point when the hourly near-surface temperature (T2m) reaches or exceeds a threshold of 0 °C in the high Arctic (north of 80°N). The strict spatial and threshold criteria restrict the events only to be found in the Atlantic sector. The authors find that these events are rare and short-lived, usually lasting less than one day. The highest frequency and longest duration of these events are found to be confined to a region close to 80°N and within 0-30°E. These events are further associated with positive anomalies in sensible heat fluxes (largest contribution closest to the lower latitude boundary) and positive anomalies in both IWV and DLW, with largest contributions especially further towards the Arctic interior. Large-scale circulation anomalies with a well-known dipole pattern in the SLP anomalies are found to be favourable for warm and moist air advection into the affected grid-points. A 100% match is found between warming events at northerly latitudes and the co-occurrence of atmospheric rivers (ARs). The grid-point wise defined warming events are then elaborated to regions of several grid-points with temperatures at or above 0 °C, i.e., "concurrent warming events", a method taking into account the spatial extent of such events. Three different large-scale circulation patterns are found to be associated with these regionally defined warming events: a strong dipole in SLP, a blocking-like surface anticyclone or a strong negative SLP anomaly over Greenland. The authors finalize the paper by a trend analysis of grid-point wise defined warming events, where a positive trend in both duration, magnitude and frequency of the events are found. This paper has well-done figures and gives a nice overview of extreme warming events (per grid-point) in the high Arctic, and discusses their drivers with respect to the changes in e.g., SEB anomalies and their relation to large-scale circulation and AR frequency. Most of the results presented here agree and follow results from previous studies. I still have a few concerns regarding the event definition and interpretation of the results presented here. Thus, I suggest minor revision before any possible acceptation of the paper. See comments below for clarifications.

**Specific comments**

- The authors aim to make a climatological record of warm extreme events and associate these to large-scale circulation patterns and ARs. But this is something that has already been done by previous studies, however, with different dataset (ERA-Interim) and event definitions. For example, Graham et al. (2017) (https://doi.org/10.1002/2017GL073395), a study that is also cited in this paper, do not only use in-situ observations, but also look at the historical record winter warming events in the Arctic using different temperature thresholds. They also find a positive trend in duration and occurrence of warming events. Also, your Fig. 1b is similar to their Fig. 2a, showing maximum 2m temperatures, however they use 6-hourly ERA-Interim data compared to yours hourly ERA5 data. Another study that is completely left out of this paper is Messori et al. (2018) (https://doi.org/10.1175/JCLI-D-17-0386.1), where the drivers behind warm extreme events in the high Arctic are also examined. Please refer to these two studies in your paper for a comparison to similar studies. What is novel in your study compared to theirs? What is the reason for an event definition with a fixed threshold in your study?

- *Warming event definition*:
  - o I also find it a bit concerning, as raised by the first reviewer regarding the event definition, that warming events, as the authors name these extremes, are defined based on a *fixed temperature threshold* (2-m temperature $>= 0°C$) and for *each grid-point* separately. What if grid-points close to each other actually belong to the same event that is affected by the same synoptic weather system? Grid-point defined warming events, using a Eulerian approach, don't say that much about the real nature of the synoptic weather event, as this event, let's call it an AR guided by the large-scale circulation, will move and affect grid-points further away in the direction of the AR, leaving the previously affected grid-point to cool. As the authors point out, there can be temperature fluctuations in hourly data (so that the temperature shortly drops below 0°C), but still be part of the same synoptic event (which has a different impact on the surface temperature depending on the time scale and location). The extent of these intrusions can then also

affect adjacent grid-points at the same time step, however, in your study, these two would be counted as two separate events? The authors explain on L98 the method "interval requirement", and state that your results are not affected by the temperature fluctuations. Could you please re-explain how this was done? Could you think of using a longer time span of 4-6 days (representing the synoptic scale) for your requirement? Thus, having less than two days between two warming events (as stated in L168-170) would most likely just be the same event. How about the close-by grids? How often do the authors find that several grid-points experience temperatures above zero degrees at the same time? One way to avoid these issues would be to maybe increase the time from hourly to daily and check for days where the temperature exceeds a threshold in the grid for at least once or up to a certain percentage within the time? The authors could also consider looking at pre-defined regions instead of per grid-point. In Messori et al. (2018), warm extreme events were defined over the polar cap (as in this study) as daily T2m anomalies computed against a transient climatology (long-term trends and seasonality removed) and area-weighted over the study domain. Furthermore, anomalies were smoothed to remove fluctuations and only events at least one week apart were chosen in order to avoid double counting of the same event. Despite this, when considering large-scale drivers, in a study from Murto et al. (2022; also referred to in the current paper), they could associate up to 3 of the warm extreme events from Messori et al. (2018) to one or two consecutive blocking events, suggesting that these warm events were actually one event but affected by a similar large-scale setting. Did the authors of this current paper try to define events using temperature anomalies? The event definition as it is now is for me a bit problematic, and caution must be taken when interpreting the results of your study. Please also point out in the abstract that events are defined per grid point.

o Another issue that arises when defining events by a Eulerian perspective is to refer to the "lifecycle" of the event, i.e., with an *onset and decay*. As written above, ARs tend to move and thereby at a future timestep will affect nearby grid-points. When talking about "onset" and "decay" of an event, it refers to some phenomenon that usually moves and which can be tracked. Thus, the same AR can continue to the next point, but it does not mean that the event has "decayed" when the temperature drops back below zero degrees at another grid-point. I would maybe consider using another word usage here instead of "onset" and "decay" when referring to the warming events at a point. Maybe use "time when a grid-point's T2m exceeds zero degrees"? Close-in-time warming events at one (or nearby) grid-point and the risk of double counting events is then further accumulated in lead/lag composite plots, as in Figs. 6-8. For example, as stated in L230, anomalies 6-days prior to the onset of a warming event could just be a result of a warming event that was present at that lag-time. A timestep at the onset of one event could be same time as the decay of another event. In Fig. 6 it is also clear that the warm air is advected to other regions, which shows that heat is transported further into the Arctic, leading to a temperature drop (and as you name it, decay of the event) at a grid-point when the source of the heat is moved (Woods and Caballero (2016) found, for example, that it takes five days for a moist intrusion to cross the Arctic interior, which is within the timeframe of the event duration here).

o *Spatial restriction*: Why did the authors decide to restrict to the polar cap (north of 80N), when events are defined from absolute temperatures? Studies found that warm extremes are mainly located in the Atlantic sector (Graham et al. 2017) and associated with strong moist-air intrusion from the Atlantic that penetrate into the high Arctic (Messori et al. 2018). With a more relaxed temperature threshold, as done in Graham et al. 2017, Pacific warming events can also occur. I would like to see some discussion to why Pacific warming events are not included in this study. For example, sea ice extends further south at the Pacific side and thus airmasses from southerly latitudes have a longer path to cool before reaching into the polar cap compared to Atlantic pathways. A northerly ice edge at the Atlantic side, on the other side, allows the air to collect moisture and heat for a longer time and distance before losing them while traveling northwards. Storm tracks might also be more active over the Atlantic side. In Graham et al. (2017), a southerly latitude band was also chosen for the Pacific side compared to Atlantic to include both sectors with warming sources. Studies also show that ARs from the Pacific side, such as in 2007, can

have an important impact on the temperatures in the Arctic. Southerly winds promoted ice-export and the warm and moist air transport over the Beaufort Sea enhanced surface temperatures, and led to anomalous SEB fluxes and the onset of sea ice melt (e.g., Graversen et al. 2011 https://doi.org/10.1007/s00382-010-0809-z and Stroeve et al. 2008 https://doi.org/10.1029/2008EO020001). Please add a bit more discussion around the reason for the chosen study area (distance to sea ice, importance of both transport pathways...) in the introduction and around L147.

- *Concurrent warming events:* (Sect. 3.3). For me, this method seems much better, as, to my understanding, these events are defined by finding areas in the Atlantic sector where several grid-points at one timestep satisfy the criterion for warming events. What is the region where these events occur (maybe show on a map figure)? Are these areas coherent, i.e., that the grid points are adjacent to each other? What are the main characteristics for these events (number of points included in one area, duration, trends, seasonal evolution, spatial frequency…)? The lifecycle perspective of these events is again utilized here (onset and decay), which rises similar questions as stated above. How is the decay defined (when the temperature for all or at least one grid point within the area drops below zero)? Again, as stated in L258, close-by areas can also be affected by the same weather event, and here the authors decide to use a 5-day temporal limit, which is good. How is the temporal criterion applied here (on warming events (grid-points) within a region or for separate regions)? The definition of the peak of these events is somewhat unclear: on L263 it is referred to as the hourly time when the area of the grid-points with $T2m \geq 0°C$ is the largest, but at L281 the authors refer to the peak time of the T2m anomaly when referring to Fig.10. Does the time at maximum area of grids satisfying the criterion always correspond with maximum T2m anomalies? Please clarify. If this is not the case, maybe an intensity measure where both the spatial extent (area) and the average temperature anomaly (magnitude) could be used to define the peak.

- How much does the *surface type* (ocean, sea ice, leads in sea ice) affect the spatial location of the warming events, and the anomalies in e.g., SEB? Did the authors consider to divide the warming events into ocean and sea ice (with different SIC's) – this could improve the quality of the paper and the interpretations of the results. L185 states that the climatological SHF is upward in winter, which is true over warm ocean or openings in the sea ice. But over sea ice, I think that the climatological values of turbulent fluxes are almost negligible, if not slightly positive. The surface type influence can nicely be seen in S2, with a sharp division between negative and near neutral values at the sea ice edge. Thus, L187 "suppression of the upward THF" is true if the surface is relatively warm, but ARs could also enhance downward THF if the surface is relatively cold (e.g., over sea ice). Discussion about the impact of different surface types on the results shown here would be nice to see and adding the sea ice edge on the climatological plots would be helpful.

- *Wording* and terminology for the warming events: The authors are not always consistent with the wording of their events: warming events or extreme warming events for the grid-point wise defined events, and concurrent warming events or large-scale events (L378) for a larger region. Please be consistent with the terminology, and always refer to which events (region or grid-point) the analysis shown in a figure or discussed in the text is referring to. For example, add it to the abstract and conclusions, as well as in figure captions (such as in Fig. 5 for the duration categories). In the abstract it is, for example, not clearly stated that the match between ARs and warming events are done on the concurrent events.

- The word "driving" or "driver" is used several times in the paper. I would rather write "associated with" or "related to" instead of driving. For example, in L300, I would rather write that the SLP pattern are "guiding" the ARs or "making a pathway".

- Figure 2c and Fig. 3a: remove the value shown in the upper left corner but keep it in the captions. Also, I assume the caption for Fig. 3 for the red line should refer to panel (a), not c. I would also suggest to change the color for zero in Fig. 3b to another color to be clearer (blue according to the colorbar means no days or actually more than 0?).

- I suggest the authors to add more references, at least when discussing the results of this paper and when comparing directly to other observations or studies (e.g., in L153).

- At L158: I assume the authors mean that the duration drops gradually away from the region at 80N, 0-30W to less than 5h north of 85N OR east of 60E (there are no data in the corner north of 85N and east of 60E, right?)

- Figure 4: Would be interesting to see how many grid-points satisfy the temperature criterion over all seasons within the study period, to know if the average e.g., DLW is a result of only a few or several events. The average occurrence of these events (S1) gives some idea about this, but maybe a relative frequency plot would better demonstrate this (so that 100% would refer to the max occurrence of events per grid point over all periods). I also find the Figure S1a to be relevant for the main paper, as main characteristics for events include spatial distribution.

- I find it interesting to see that you find a relationship between the duration of the events, their locations and associated SEB anomalies. Have you performed any correlation analysis to make your statement stronger? I assume that we would then find higher correlation between longer-lasting events (longer duration) and SHF anomalies, whereas shorter-lived events would be correlated with DLW, especially at higher latitudes. Intuitively, I would think longer lasting events are those that penetrate further into the Arctic, but this confusing comes only because of your different way of defining events and their duration (at a fixed point; your longer-lasting events are closer to the warm and moist air source). Maybe worth reminding the reader again what your definition is to avoid possible confusion.

- When talking about anomalies in SLP or SEB components, for example, we are mainly interested in the sign of the anomaly. Therefore, I would suggest writing "positive/negative" instead of "high/low" when writing about anomalies.

- Figure 5: what is the absolute number of events within each sub-category based on duration? A map showing the locations of these grid-points would be nice to see. Further point out in Fig. 5 caption that the durations are defined per grid-point. Could the stronger anomalies shown in Fig. 7 compared to the other ones be a result of less events included in the composite?

- Figs 6-8: these figures are nice but also a bit messy with so many black lines. I would suggest changing the color for the SLP anomaly contours to purple (as in Fig. 13a).

- How do the authors think that the spatial patterns of the events (or occurrence and duration) might directly be affected by cyclones or blocking? Adjacent grid-points could be captured by the same cyclone but affected by either the cold advection or the warm sector (L206). Have you looked at cyclone tracks or used blocking detection algorithms to associate your warming events with them (instead of using ARs and SLP anomalies to represent the large-scale setting)? The authors also write "blocking" (L380) in the conclusions despite not using a blocking detection algorithm (?). I would suggest rewriting to "blocking-like structure" and maybe refer to other studies (e.g., Woods and Caballero 2016, Messori et al. 2018 and Murto et al. 2022).

- In sect. 3.4, why do the authors return back to the grid-point defined warming events to relate them to ARs? Have you looked at how your results would change when using the concurrent warming events?

- L325: are these nine days of T2m above zero degrees found here in your study? Please mark their locations on a map, e.g., in Fig. 13a. Have you studied the origin of the ARs or utilized the tracking algorithm of the AR shapes, if it was provided with the AR algorithm?

- Figure 10: Are the T2m anomalies shown for all grid-points independent of if the grids are part of a concurrent warming event (or only that the time is at the peak of the event)? Maybe some density lines would be helpful to show where these concurrent warming events are spatially located and how these anomalies extend wrt the originally defined events.

- One additional concern is the statement at L318: "ARs are the *only* weather system capable of triggering the occurrence of the warming events". ARs are definitely important! That grid-points are co-occurring in time and space with ARs (even though 100% overlap), does not, however, directly imply that these warm anomalies can only result from an AR. In a recently published paper (Papritz et al. 2023 https://doi.org/10.1175/JCLI-D-22-0883.1), the relative contributions to the warm potential temperature anomalies extending in the whole tropospheric column (associated with extreme positive SEB anomalies over wintertime Arctic sea ice) were investigated. They found (using backward trajectories) that only airmasses ending up in the middle troposphere had an AR-like evolution, whereas airmasses making up the positive anomalies closest to the surface actually had an Arctic origin! These airmasses were either warmed diabatically while crossing over warmer oceans or when airmasses descend from higher altitudes, but all within the Arctic. It was these two airmasses together that could give rise to these anomalous positive vertically extending potential temperature anomalies. Local processes are thus also important, so I would suggest rewriting this strong sentence and add "likely" or "strong impact" instead of stating ARs are the only driver.

- Are the trends discussed in Sect. 3.5 based on grid-point defined warming events? Referring here to an absolute number of warming events (2150) does not tell the reader so much, as your events (and the number) are dependent on the grid-size, the temporal resolution of the data etc. Maybe more informative would be to show seasonal trends of days with atleast one warming event per grid point? Or rather use the concurrent warming events here and calculate their trends per decade. I also think that the event definition (a fixed temperature threshold) and with a rapidly warming Arctic, warmer temperatures become more common by default (compared to the first decade of your study period).

- Figure 13c is not referred to in the main text

- I am also lacking a final concluding statement. The authors nicely summarize the findings of the paper in the final section, and list some limitations. What potential of further studies would your study contribute? One way to tie the final section to the rest of the paper is to answer the questions raised in the introduction in the conclusions.

- Please add some discussion about ERA5 warm bias to the limitations of this study (relate to the representation of snow and sea ice in ERA5, see e.g., Batrak and Müller 2019 https://doi.org/10.1038/s41467-019-11975-3).

**Technical corrections**

I agree on the typing errors pointed out by the first reviewer. Below some minor correction suggestions in addition to them:

- Add a reference to panel (f) for the temperature advection in the figure (Fig. 11) caption
- I would suggest to add the latitude threshold in the caption for the long-lasting events in Fig. 5g, h
- Be consistent with the terminology used in the text and in the figures. In Figs. 5a and 14a, "TS" is used instead of "T2m" (I assume), it is always referred to as T2m in the main text.

---

## Author Response (AR1)

**egusphere-2023-2018: Response to Review Comments**

We thank both reviewers for their insightful comments and constructive suggestions that helped improve both the quality and clarity of this work. Below please see our point-by-point responses to all review comments and revisions made to the manuscript. The original comments are in blue.

**Reviewer #1**

**General comments:**

This study investigates the characteristics of extreme wintertime Arctic warm events, using hourly ERA5 data for the period 1979-2021. In most of the study, Arctic warm events are defined as grid points poleward of 80°N where the two-meter air temperature (T2m) exceeds 0°C. Adjacent grid points with T2m > 0°C at the same time are considered as separate warm events. They find that the events are rare and only occur over the Atlantic sector of the Arctic, with a mean duration of less than half a day. Warm events are associated with positive anomalies in integrated water vapor, downward longwave radiation, sensible heat flux and, in most areas, latent heat flux. They are located in a region of intense sea level pressure (SLP) gradients, with a negative SLP anomaly to the west and a positive anomaly to the southeast, and they typically coincide with atmospheric rivers. In an additional analysis of so-called "concurrent warm events", where events are defined as coherent objects and not as single grid points, they use a K-means clustering method and find three different large-scale circulation clusters, with the first characterized by a strong SLP dipole, the second mainly by a strong surface anticyclone and the third mainly by a strong surface cyclone. Finally, they show positive trends in the number of events (again defined as single grid points with T2m > 0°C), their magnitude and duration over the past 40 years.

Overall, the manuscript is interesting, and the study may be well suited for publication if a couple of scientific comments are addressed, and some formulations are further improved. Some specific suggestions are given below.

We thank the reviewer for taking the time to review our manuscript and providing thoughtful and constructive comments. In particular, the comments on the definition of the warming events and the suggestions on showing the trends in the characteristics of the concurrent warming events helped improve the clarity and quality of our manuscript. Our responses to the specific comments are shown below, with their original comments in blue.

**Specific comments:**

1) My major comment concerns your definition of Arctic warming events. You call it a warming event when T2m exceeds 0°C at a grid point. If T2m is > 0°C at several neighbouring grid points at the same time, you refer to them as several separate warming events. I find this quite confusing, as it is most likely just one event with a larger spatial extent. You then write that the

duration of the events is often less than 1h (your "short-duration events"), but I assume that they are typically much longer-lived when they can move over several grid points. And according to your description of Fig. 5, the temperature drops very quickly after an Arctic warm event, but this is just the temperature at a specific grid point. The composites in Fig. 6-8 indicate that the warm temperatures are simply being advected to other grid points. I assume there is a lot of double counting in the different figures (for instance, the fields in the composites are considered several times if a warm event spans several grid points). I think you have to be very careful with the interpretation of your results and with the wording. Rather than "warming events", I would typically write something like "grid points with T2m > 0°C" to avoid confusion, and mention where necessary that the fields are considered several times when a warm episode covers more than one grid point.

I find the definition of the "concurrent warming events" much more convincing. If I understood it correctly, there you considered adjacent grid points with T2m > 0°C as one "warming event" and also somehow traced the events in time. How is the tracing working exactly, can the warm events move over several grid points? It would be interesting if you could provide some information about their occurrence frequency in the past 40 years, their duration, their spatial extent and other characteristics. And maybe you do not need to impose a threshold on the area and the duration of these events, as you do so far, but could investigate the characteristics of all of them.

Our study is inspired by Moore (2016) and Graham et al. (2017), both of which used meteorological buoy observations to characterize the late December 2015 North Pole warming event. In both studies, data from these meteorological buoys, located at different locations (grid points) around the North Pole, were used to characterize the same evolving large-scale warming event (defined as concurrent warming event in our study). As mentioned in the original manuscript (line 90), defining Arctic warming events at the grid-point scale can facilitate a direct comparison with the findings in Moore (2016) and Graham et al. (2017). Furthermore, defining these warming events at the grid-point scale can also help us gain new insight into what determines the duration of these events at different locations. For example, we find that the location of the grid points experiencing warming events relative to that of the sea level pressure (SLP) dipole is key in determining the event duration. We also find that long duration events are driven by anomalous downward sensible heat flux (SHF) while anomalous downward longwave (LW) radiation seems to play a more important role in driving short duration events. These insights are hard to obtain if we define warm events from a Lagrangian perspective as a contiguous region with T2m >= 0°C. As events defined in this way would inevitably encompass many grid points at any given time and the events also evolve constantly through space and time.

We do agree with the reviewer that when several neighboring grid points are experiencing T2m >= 0°C simultaneously, we can view them as one event with a larger spatial extent. This perspective is totally valid. In fact, to take this perspective into account, we define the so-called "concurrent warming events" later in the manuscript. In this study, we examine these warming events from both the grid-point scale perspective and the large-scale perspective suggested by the reviewer. However, we focus more on the grid-point scale perspective, following Moore (2016) and Graham et al. (2017), who examined the late December 2015 high Arctic warming event similarly.

It's fairly common to define extreme events at grid-point scale in the literature. For example, when extreme precipitation over midlatitudes is driven by extratropical cyclones or atmospheric rivers, it can occur in many neighboring grid points. This type of extreme precipitation can be treated as one event with a large spatial extent and characterized with contiguous regions exceeding a threshold by tracking them through space and time. However, it is also common to study extreme precipitation at grid-point scale (e.g., Pfahl et al., 2017). Analogously, we believe it is reasonable to define extreme warming events at the grid-point scale as well. Nonetheless, to more explicitly explain why we define warming events at the grid-point scale and to acknowledge the large-scale perspective, we have now added the following text to the Methods section:

"*Case studies based on in-situ buoy observations have been conducted by previous studies to examine the characteristics and drivers of winter extreme warming events over the high Arctic (Moore, 2016; Graham et al., 2017), which characterized extreme warming events using point observations based on meteorological buoys. Since these extreme warming events are usually driven by large-scale circulation (Woods and Caballero, 2016; Moore, 2016; Kim et al., 2017; Messori et al., 2018), when extreme temperature is detected over a grid point, it is common that such extreme temperature can also be found over its neighbouring grid points. Under such conditions, one approach is to define extreme warming events by using contiguous regions with temperature exceeding a predefined threshold. However, to facilitate a more direct comparison with previous case studies shown in Moore (2016) and Graham et al. (2017), here we choose another approach to define extreme warming events at the grid-point scale. Specifically, in this study, extreme warming events are defined as those events with 2-meter air temperature (T2m) over a grid point reaching or exceeding* $0°C$ *over the high Arctic that covers the regions poleward of* $80°N$*. Although the focus of this study is on warming events defined at grid-point scale, we also investigate extreme warming events identified by contiguous regions with temperature exceeding a threshold in section 3.3, which complements the results from the grid-point scale perspective.*"

Of course, as pointed out by the reviewer, defining extreme warming events at the grid-point scale can result in double counting of the same large-scale circulation pattern when making composites. In fact, double counting inevitably occurs when creating the composites shown in Figs. 6, 7, and 8 of the original manuscript (Figs. 7, 8, and 9 in the revised manuscript). However, even if the same large-scale pattern occurs multiple times within the same composite or across composites, the relative position of the SLP dipole within the same composite and across composites is what matters. As the purpose of these composites shown in Figs. 6, 7, and 8 is to show that the duration of the warming events is determined by the position of the SLP dipole relative to the location of grid points with the extreme warming events, the double counting here would have minimal impact on our conclusion.

To further show that the impact of the double counting on our conclusion is indeed minimal, we reproduced Figs. 6, 7, and 8 by randomly picking one grid-point scale event at each of those hours with at least one event and compositing all randomly picked events. More specifically, at each hour, we first identify all the grid points with T2m first exceeding 0°C. The number of the grid points with T2m first exceeding 0°C gives us the number of events at that hour. We then record the duration and position (latitude) of all the identified events of that hour and classify them into different categories (e.g., long duration events, short duration events poleward of

85°N, and short duration events equatorward of 83°N). For the categories with at least one event, we then randomly pick one event from each of those categories. The above steps are repeated for each hour during the study period. Lastly, the randomly picked events of each category are used to construct the composites. This approach can ensure that only one event is selected, even when neighboring grid points are experiencing warming events simultaneously. The same large-scale circulation pattern is thus counted only once. As shown in Figs. R1, R2, and R3, the results based on the composite approach described above closely resemble those presented in Figs. 6, 7, and 8. The only difference is that Figs. R1, R2, and R3 tend to be nosier, especially for the composites of the short duration events poleward of 85°N, owing to the smaller sample size. These results thus further confirm that double counting have minimal impact on our conclusion.

To point out this caveat, the following discussion has been added to the manuscript:

"*As the same large-scale circulation pattern can cause extreme warming events to occur over more than one grid point, it thus can be counted more than once within the same composite or across different composites. However, as the composites are centred at the grid point where the extreme warming event is identified, the position of the same large-scale circulation relative to the grid point would thus differ among the composites centred at different grid points. As we show below, the relative position of the SLP anomalies is what determines the duration of the extreme warming events. The double counting of the same circulation pattern for the composites shown in Figs. 7, 8, and 9 thus has minimal impact on the conclusion.*"

[Figure]

**Figure R1.** The same as Fig. 6 in the original manuscript (Fig. 7 in the revised manuscript), but based on the event sample obtained by randomly picking one event for each of those hours when at least one event is present. See the response above for a more detailed description of the composite approach.

[Figure]

**Figure R2.** The same as Fig. 7 in the original manuscript (Fig. 8 in the revised manuscript), but based on the event sample obtained by randomly picking one event for each of those hours when at least one event is present. See the response above for a more detailed description of the composite approach.

[Figure]

**Figure R3.** The same as Fig. 8 in the original manuscript (Fig. 9 in the revised manuscript), but based on the event sample obtained by randomly picking one event for each of those hours when at least one event is present. See the response above for a more detailed description of the composite approach.

For the "concurrent warming events", we consider all grid points that experience warming events simultaneously as part of a single event. Since the region where T2m can exceed 0°C is usually small (see Fig. 1b), when many grid points over a region with small spatial extent experience T2m above 0°C at the same time, it is very likely that they are driven by the same large-scale circulation pattern. Visual examination of these concurrent warming events does confirm that they are spatially coherent. We thus did not track the events, nor did we impose the constraint that grid points with T2m above 0°C at the same time have to be all connected to each other. However, we do impose that the peak area with T2m >= 0°C for a single concurrent warming event must be greater than $5 \times 10^9$ m$^2$. This is to ensure that the identified concurrent warming events are driven by large-scale circulation instead of being caused by small-scale local fluctuations of T2m. Although the focus of our study is on the statistics obtained based on events defined at the grid-point scale, we agree with the reviewer that it would also be interesting to characterize the 96 concurrent warming events identified in Section 3.3 of our study and examine their trends in frequency, duration and spatial extent. We have updated Fig. 14 in the original manuscript (Fig. 15 in the revised manuscript) to include trends in the frequency, spatial extent and duration of the concurrent warming events identified in our study. As shown in Figs. R4c and d, the frequency, spatial extent, and the duration of the concurrent warming events all exhibit significant upward trends in the past four decades. The following discussion has been added to Section 3.5 in the revised manuscript:

*"Consistent with these increasing trends in the characteristics of the extreme warming events defined at the grid-point scale, the frequency, spatial extent, and duration of the concurrent warming events defined in Section 3.3 also exhibit significant positive trends in the past four decades (Figs. 15c, d)."*

[Figure]

**Figure R4.** Trends in (a) the area-weighted spatial mean T2m over the entire high Arctic (black solid line with circles), mean T2m only for grid points above 0℃ (red solid line with circles) and the seasonal maximum hourly T2m (red dashed line with triangles) over the high Arctic. (b) is the same as (a), but for the trends in event count (black solid line with circles), mean event duration (red solid line with circles ) and seasonal maximum event duration (red dashed line with triangles). (c) and (d) show trends in the characteristics of the 96 concurrent warming events identified in Section 3.3. Trends in (c) the event count (black solid line with circles), mean peak area (solid red line with circles) and seasonal maximum peak area (red dashed line with triangles) of the concurrent warming events. (d) shows the mean duration (red solid line with circles) and seasonal maximum duration (red dashed line with triangles) of the concurrent warming events. For those seasons with only one concurrent warming event, the mean peak area would be the same as the maximum peak area in (c) and the mean duration would be the same as the maximum duration in (d). All trends are significant at the 0.05 level based on the Student's t-test. (Same as Fig. 15 in the revised manuscript).

Reference:
Pfahl, S., O'Gorman, P. & Fischer, E. Understanding the regional pattern of projected future changes in extreme precipitation. *Nature Clim Change* 7, 423–427 (2017). https://doi.org/10.1038/nclimate3287

2) Abstract and section 4: Please specify that you define an "Arctic warming event" as a grid point with T2m > 0°C.

We have specified that the Arctic warming events are defined as a grid point with T2m >= 0 °C.

3) Lines 15-16: "… with a seasonal occurrence frequency of less than one over most of the regions." Could you be more specific about what you mean with "less than one"? Instead of a decimal number per season, which might be a bit difficult to grasp, it would be more informative if you could write the absolute number of events with T2m > 0°C in the considered time period. As you define the events per grid point, you could provide a mean number over all grid points (or two separate numbers for more southern and more northern latitudes).

Since the extreme warming events are defined at grid-point scale, we simply count the total number of events that happened at each grid point during the 42 winters (1980-2021), and then divide this total count by 42. When the mean seasonal occurrence frequency over a grid point is "less than one", this implies that none is detected during some of the 42 winters at this grid point. If we average the occurrence frequency over all grid points where T2m ever reached or exceeded 0°C poleward 80°N, we get a value of 0.45. This also suggests that the mean seasonal occurrence frequency is less than one. According to the spatial distribution of occurrence (Fig. S1a in the original manuscript), it is reasonable to say that "they occur rarely, with a seasonal occurrence frequency less than one over most of the regions". To improve the clarity of this sentence, we have made the following revision to the sentence in line 15-16 of the original manuscript:

"*They occur rarely, with a total absence during some winters over most of the region.*"

In addition, we have made a new figure (Fig. R5) that shows the spatial distribution of the total occurrence frequency of the extreme warming events over the entire study period and included it as Fig. S1 in the revised supplementary document. Accordingly, the following discussion has been added to the Section 3.2 of the revised manuscript:

"*The total number of hours with T2m >= 0°C over the study period reaches its maximum (more than 1000 hours) over the region near 80°N and between 0°-30°E (Fig. S1). Moving away from this region, the number drops rapidly towards the boundary with regions that have never experienced T2m >= 0°C during winters of the past four decades.*"

[Figure]

**Figure R5**. Spatial distribution of the total number of hours with T2m >= 0°C for all winters during 1979-2021. The magenta contour outlines the regions that have ever experienced T2m >= 0°C. (Same as Fig. S1 in the revised supplementary).

4) Lines 69-70: "Subsequent examination reveals … with the duration of staying above 0°C for less than an hour." Which study are you referring to? According to Fig. 1 in Binder et al. 2017 (Geophys. Res. Lett.), maximum temperatures reached values above 0°C during three episodes that each lasted about 1 day, i.e., much longer than 1 hour.

Fig. 1 in Binder et al. (2017) shows the temporal evolution of the domain maximum T2m. The domain in their study is defined as the region poleward of 82°N and between 120°W and 120°E. Therefore, it is reasonable that the warming event in Binder et al. (2017) is more persistent. However, the studies we are referring to are Graham et al. (2017) and Moore (2016), as cited in the sentence before Lines 69-70 of the original manuscript. Fig. 1 in Graham et al. (2017) and Fig. 2 in Moore (2016) show the temporal evolution of surface air temperature observed on meteorological buoys at different locations (grid points) near the pole. Both figures show the short-lived nature of these warming events at the grid-point scale. To clarify, we have added "*locally (Fig. 1 in Graham et al. 2017 and Fig. 2 in Moore, 2016)*" after the sentence in lines 69-70.

5) "This event is driven by an AR-like moisture plume carried into the high Arctic by a cyclone." Please indicate which study you are referring to in this sentence.

We are referring to Fig. 4 in Moore (2016). This reference has been added in the revised manuscript.

6) Please also cite the paper by Messori et al. 2018, J. Climate (doi: 10.1175/JCLI-D-17-0386.1), which also investigates drivers of wintertime Arctic warm events.

Thank you for mentioning this very relevant paper. We have now cited this paper properly in the revised manuscript.

7) Lines 90-92: From the sentence it is not clear based on which dataset you define the extreme warming events. Is it based on buoy observations, as suggested by the title of the subsection and the first sentence, or based on ERA5, which you mention later in the same subsection? If – as I assume – you use ERA5 data to define the events, then I find the title and the first sentence of the subsection rather confusing. Also, in that case I would first describe the ERA5 data in a paragraph and mention the temporal and spatial resolution, the season, the time period you investigate, etc., and only then describe how you identify the extreme warming events. On the other hand, if you use buoy observations, please describe this dataset in more detail. How do you obtain gridded data from buoy observations (Fig. 2)? Furthermore, in that case I would make a separate subsection for the description of the ERA5 data.

Sorry for the confusion. Our study is based solely on the ERA5 data. The buoy observations mentioned in the first sentence were used in Moore (2016) and Graham et al. (2017) that are also cited in the first sentence. We have added "by previous studies" in the first sentence to avoid confusion. The sentence is now revised to "*Case studies based on in-situ buoy observations have been conducted by previous studies to examine the characteristics and drivers of winter extreme warming events over the high Arctic (Moore, 2016; Graham et al., 2017)*".
Furthermore, to improve the flow of Section 2.1, we have now switched the order of the first and second paragraphs of the original manuscript. Section 2.1 now starts with describing the ERA5 dataset and then introduces the definition of the extreme warming events.

8) Line 101: "Results based on previous studies …" Please add references. And maybe this sentence fits better in the introduction than the method section.

Moore (2016) and Graham et al. (2017) have been added as references. As we explained above, this sentence/paragraph has been moved up for a better reasoning of and transition to the methodology of our study.

9) Lines 113-115: How do the trends in DLR, IVT, LHF, SHF and IWV look like, are they all positive? How does the detrending work?

As shown in Fig. R6 below, the trends in DLW (or DLR), IVT and IWV are all positive over the entire high Arctic, especially over the Atlantic sector. For LHF and SHF, they show positive trends over the sea ice covered region of the Atlantic sector, and negative trends over the lower latitude region of the Atlantic sector due to sea ice retreat.
For the detrending, we follow the method used in Wang et al. (2020). For each grid point, we need to remove the seasonal cycle from the time series of hourly data. To obtain the seasonal cycle in the first place, we average the hourly time series across the years from 1979 to 2021 to obtain the raw seasonal cycle first, and then apply a 31-day running mean to the raw seasonal

cycle to obtain the smoothed seasonal cycle used for deseasonalizing the data. After removing the seasonal cycle, we calculate the linear trend based on the deseasonalized data. For each given hour of the winter season, we first calculate the 31-day running mean centered at the hour. The smoothed data is then used to derive the linear trend used for detrending for the hour.

We have updated the paragraph in lines 111-116 of the original manuscript to better explain how the detrending works:

"*Following Wang et al. (2020), all anomalies presented in this study are obtained by first removing the seasonal cycle. To obtain the seasonal cycle in the first place, the hourly time series is averaged across the years from 1979 to 2021 to get a raw seasonal cycle, and then a 31-day running mean is subsequently applied to obtain a smoothed seasonal cycle for de-seasonalizing. The Arctic has experienced amplified warming in the past four decades. This amplified warming results in significant positive trends in the downward longwave radiation (DLW), integrated water vapor transport (IVT) and integrated water vapor (IWV) over nearly the entire high Arctic and significant positive (or negative) trends in the latent heat flux (LHF) and sensible heat flux (SHF) over the sea ice covered (or partially sea ice covered) region of the high Arctic Atlantic sector (not shown). To exclude the effects of these decadal trends, the anomalies of T2m, DLW, IVT, LHF, SHF, and IWV are further detrended after removing the seasonal cycle. Similarly, for any given hour, to obtain the linear trend, a 31-day running mean centered at the hour is first applied to the de-seasonalized data. The smoothed data is then used to derive the linear trend used for the detrending.*"

[Figure]

**Figure R6**. Winter trends in (a) Downward longwave radiation (DLW), (b) integrated water vapor transport (IVT), (c) integrated water vapor (IWV), (d) latent heat flux (LHF) and (e) sensible heat flux

(SHF) from 1980 to 2021. Stippled areas indicate anomalies are significant at the 0.05 level based on the Student's t-test.

Reference:
Wang, Z., Walsh, J., Szymborski, S., and Peng, M.: Rapid Arctic sea ice loss on the synoptic time scale and related atmospheric circulation anomalies, J. Clim., 33, 1597–1617, 2020.

10) Line 158: "Over regions eastward of about 60°N and poleward of about 85°N, the mean duration is shorter than 5 hours." How do you arrive at this statement? There is no data in this region in Figs. 2a,b, right?

Sorry for the confusion. There is no data over regions eastward of about 60°E and poleward of about 85°N. Instead of "poleward of about 85°N", it should be poleward of about 80°N. We have corrected the typo.

11) Lines 165-171: Maybe you can add that it is very likely that two warming events that occur right after each other are driven by the same weather system (as you already pointed out in section 2.1).
Thank you for the suggestion. The following sentence has been added to the paragraph to point out the possible cause of the event clustering:

"*As being shown in Section 3.3, the clustering of these warming events may be driven by the same persistent large-scale circulation that steers successive weather systems into the affected region.*"

12) Fig. 4: What do the fields show exactly? Is it the anomaly of the mean value of IWV, DLR, etc., averaged over all warm events at a specific grid point? And maybe you can add in the caption that positive values in d) and e) indicate fluxes directed from the atmosphere toward the surface.

Yes, they are the mean anomaly values of IWV, DLW, etc., averaged over all extreme warming events at a specific grid point. We have now updated the caption of Fig. 4 to better explain what those fields are and also to point out that positive values in d) and e) indicate fluxes directed from the atmosphere toward the surface.

"*Spatial distribution of the mean anomalies of (a) T2m, (b) column-integrated water vapor (IWV), (c) downward longwave radiation (DLW), (d) sensible heat flux (SHF), (e) latent heat flux (LHF) and (f) horizontal temperature advection averaged over all hours with T2m >= 0 °C over a grid point. Positive values in (d) and (e) indicate fluxes directed from the atmosphere toward the surface. Stippled areas indicate that anomalies are significant at the 0.05 level based on the Student's t-test.*"

13) Line 185: "due likely to the partially open ocean" Would it be possible to overlay a contour of the mean position of the sea ice edge during warm events and in the climatology?

Thank you for the suggestion. We have overlaid the climatological 50% sea ice concentration contour (black contour) and the 50% sea ice concentration contour during warming events (red contour) in Fig. S2 (shown below as Fig. R7).

[Figure]

**Figure R7**. The climatology of (a) sensible heat flux and (b) latent heat flux during winter of 1979-2021. Overlaid are the climatological 50% sea ice concentration contour (in black) and the 50% sea ice concentration contour averaged over all warming events (in red). Positive values indicate the flux is directed downward toward the surface.

14) Line 190-191: "likely caused by the rapid cold and dry advection shortly after the onset of warming events" Do you know why there is cold and dry advection in this region after the onset of the warming events?

Based on Fig. 7 in the original manuscript, this could be caused by the passage of the cold front. The following sentence has been added to provide a possible explanation:

"*As being shown later, such rapid transition to cold and dry advection could be caused by the passage of a cold front.*"

15) Fig. 5: Which one is the curve for T2m, is it the black line labelled "TS" in the legend? If so, please adapt it in the legend. And how are the curves constructed? Do you show the anomaly of the mean T2m, DLW, etc. over all grid points with T2m > 0°C? If so, please be more specific in the figure caption. Also, as mentioned in the first comment, I find it a bit confusing that you write in the caption "for all the warming events", I would rather write something like, "at grid points where T2m > 0°C", because most likely neighbouring grid points with T2m > 0°C at the same time are not separate warming events.

Yes, the black line labeled "TS" is the curve for T2m. For consistency, we have changed "TS" to "T2m" in the legend. These curves are constructed by taking an average of the temporal evolution of various anomaly terms across all warming events. Day 0 corresponds to the onset of the warming events defined at the grid-point scale. To be more specific, we have now updated the caption and explicitly defined extreme warming event in the caption:

*"Temporal evolution of the anomalies of T2m, DLW, SHF, LHF, IWV and temperature advection for all the extreme warming events defined as any grid points with T2m >= 0 °C (a, b), short duration events equatorward of 83°N (c, d), short duration events poleward of 85°N (e, f), and long duration events (g, h). Note that long duration events occur only over regions equatorward of 83°N. These curves are constructed by averaging the temporal evolution of various anomaly terms across all extreme warming events within the different defined groups. There are 191555, 18586, 1642, and 10097 events included in the groups of all events, short duration events equatorward of 83°N, short duration events poleward of 85°N and long duration events, respectively. Day 0 corresponds to the start of the extreme warming event. The shading indicates that the anomalies are significant at the 0.05 level based on the Student's t-test."*

16) Line 202-203: "These results suggest that a warm and moist winter favours the occurrence of warming events." It could also be the other way around: the warm events contribute to making the winter anomalously warm and moist. Maybe the anomalously high temperatures nine days before the onset of the extreme warming event are related to a previous warm event in some region of the Arctic. For this, a timeline of the number of warm events per season could be interesting.

We totally agree with the reviewer that while a warm and moist winter favors the occurrence of warming events, the occurrence of extreme warming events also contributes to making the winter anomalously warm and moist, and thus precondition the environment that favors the occurrence of the next event. We have now updated the paragraph and included this alternative perspective in the revised manuscript.

*"Even nine days before the start of the extreme warming events, T2m is already 3-4 ℃ higher than normal (Fig. 6a). This is likely because these extreme warming events occur more often during warm winters, but it is also possible that this warm anomaly is preconditioned by a previous warm event. The background temperature over the Atlantic sector, under which these events occur, is thus anomalously warm. Indeed, the occurrence frequency of the events correlates significantly with the winter mean T2m over the Atlantic sector of the Arctic (or the entire Arctic), with a correlation coefficient of about 0.64 (0.61). Consistent with the warm anomalies, both IWV and DLW show positive anomalies. These results suggest that while the occurrence of these extreme warming events contributes to making the background state anomalously warm and moist, a warm and moist winter in turn favours the occurrence of extreme warming events."*

As for a timeline of the number of warm events per season, if we understand correctly, such a figure has been provided later in the original manuscript in Fig. 14b (Fig. 15b in the revised manuscript), which shows the event count of each winter (black line).

17) Line 204-205: "Both the SHF and DLW play comparable roles in driving these events, …" I find it confusing that you write that SHF and DLW "drive" the Arctic warm events. I assume that it is mainly the large-scale flow configuration that drives these events, i.e., the dipole in the SLP anomalies that you see in your composites (Fig. 6 and 9) and the upper-level flow structure (as

has also been described in previous studies, e.g., Binder et al. 2017, Messori et al. 2018). Also, in the conclusion (line 369), you write that long-duration events are mainly driven by persistent downward SHF anomalies. Maybe you can just write "associated with" instead of "driven by".

We agree with the reviewer that the large-scale circulation pattern associated with the SLP dipole is what fundamentally drives the heat and moisture changes, including the anomalies in SHF, DLW and T2m. However, in order to trigger the extreme warming events, the SLP dipole must first drive the anomalous downward SHF and DLW. In this regard, it is fair to say that both the SHF and DLW are the **direct** drivers of the warming events. We also agree with the reviewer that "associated with" is an alternative terminology to describe the relationship between the surface energy budget terms and the warming events. Accordingly, we have updated the sentence "*Both the SHF and DLW play comparable roles in driving the events…*" in line 205 of the original manuscript to "*Both the SHF and DLW play comparable roles in **directly** driving the events…*" in the revised manuscript. We also changed line 369 in the original manuscript to "*Notably, long-duration events, which occur over regions near 80°N, are mainly **associated with** persistent downward SHF anomalies*".

18) Figs. 6-8, picking up on the first comment: Since neighbouring grid points with T2m > 0°C at the same time are considered as separate events, is it right that they are included several times in your composites? Could it then just be that your "short-duration events" are the grid points located closest to the cold front of the cyclone and the "long-duration events" are grid points located further to the east in the cyclone's warm sector, but they actually occur at the same time and belong to the same warm episode? This indicates that your definition of "warm events" might be problematic.

As we have explained in our response to the first major comment, extreme warming events are often defined in two different ways. One is to define them locally at the grid-point scale, which is adopted in our study. The other is to define them as a contiguous region with spatially peak T2m exceeding a predefined threshold. This second definition is very similar to the "concurrent warming events" defined in our study. These two definitions provide complementary perspectives on studying the warming events.

Based on Figs. 6 and 8 in the original manuscript, which show the composites of the short duration and long duration events equatorward of 83°N, respectively, it is possible that the short duration events more likely occur at the grid points located closer to the cold front while the long duration events more likely occur at the grid points located further to the northeast, in the warm sector of cyclones. This relationship between the relative position of the large-scale circulation and the duration of the warming events can only be revealed when warming events are defined at the grid-point scale. When warming events are defined using the second method described above, all the grid points within a contiguous region with T2m greater than 0 °C are treated as one coherent event, the spatial heterogeneity of the event duration cannot be characterized and would thus have to be neglected.

19) Line 228: "a high anomaly to their southeast and a low anomaly to their west." I guess you mean anomalies in SLP, please write this explicitly. And I would write positive/negative (rather than high/low) SLP anomalies (also in the rest of the manuscript).

Thank you for the suggestion. We have changed the high/low anomaly to positive/negative SLP anomaly throughout the manuscript.

20) Fig. 7: It is difficult to distinguish the IVT anomaly vectors and the SLP anomaly contours. Maybe you can change the colour of either of them? And it appears that there should be more SLP anomaly contours above +21 hPa, but the plotting stops at this threshold, which looks a bit strange.

Thank you for the suggestion. We have updated this figure along with Figs. 6 and 8 of the original manuscript.

21) Line 240-242: "… the high anomaly southeast of the event regions already starts to develop for the long duration events (Fig 8)." Please specify which time step you are referring to.

We have specified that we are referring to the time step six days before the onset of the extreme warming events.

22) Line 317-318: "… for a large fraction of the regions where warming events can occur, ARs are the only weather system capable of triggering the occurrence of the warming events." Fig. 12a simply tells you that warm events typically co-occur with ARs, but I don't agree with your statement that they are the only weather systems that can trigger warm events. In contrast, in most cases it is probably the interplay between various weather systems that is important for triggering the event (like, for instance, the surface cyclones to the west and the anticyclones to the east, which channel the poleward heat and moisture transport, as well as geopotential height anomalies at upper levels, etc.). The AR can only reach the Arctic because of this interplay between various weather systems.

We agree with the reviewer that the interaction between Arctic cyclones to the west and the anticyclones to the east is what causes the formation of Arctic ARs and ultimately drives the occurrence of the warming events. However, when these warming events occur, they are directly under the influence of ARs. To account for the subtlety, we have rewritten the sentence as follows:

"*The results here thus suggest that, for a large fraction of the regions where extreme warming events can occur, the presence of ARs and their impact on and interaction with the local environment (Papritz et al., 2023) likely exert a strong control on the occurrence of these events.*"

23) Fig. 13c is not mentioned in the text.

Lines 333-334 in the original manuscript refer to the results shown in Fig. 13c of the original manuscript. We have now made this sentence explicitly referring to Fig. 13c (Fig. 14c in the revised manuscript).

*"During deep intrusion days, the daily IVT averaged over regions poleward of 85°N and between 15°W-60°E increases substantially from the climatological daily mean of ~25 to ~78 kg m⁻¹ s⁻¹ (Fig. 14c)."*

24) Fig. 14: Is mean T2m averaged over the entire Arctic poleward of 80°N or only the Atlantic sector that you consider in the rest of your study?

It is averaged over the entire Arctic poleward of 80°N. We have now pointed this out explicitly in the caption.

25) Line 342-343 and Fig. 14b: "The event occurrence frequency has been increasing at a rate of 2150 events per season per decade." Here it would again be helpful if you could specify that you mean the number of grid points with T2M > 0°C. This number of course strongly depends on the grid spacing. I would find it more meaningful if you showed a timeline of the number of days per season (or the number of hours) where the temperature exceeded 0°C in some region of the domain, or a timeline of the number of your so-called "concurrent warming events".

Thank you for the suggestion. We have now explicitly pointed out that this increasing trend is for the events defined at the grid-point scale.

*"The event occurrence frequency has been increasing at a rate of 2150 events per season per decade for the extreme warming events defined at the grid-point scale (Fig. 15b)."*

It is true that the exact number for the trend in the occurrence frequency depends on the grid spacing. However, the exact number for the trend is not very important, what matters more is the significant positive trend, as it informs us that the total area experienced extreme warming events has been increasing.
As suggested by the reviewer, we also plotted the time series of the number of days per season (Fig. R8a) and number of hours per season (Fig. R8b) with at least one warming event (defined at the grid-point scale) occurring poleward of 80°N. Both trends are significantly positive at the 0.05 level. Results based on Fig. R8 have now been included in Section 3.5 of the revised manuscript and Fig. R8 has been included as Fig. S4 in the supplementary.

*"Consistent with the increasing trend in the occurrence frequency, both the number of days and the number of hours with at least one warming event found over the high Arctic exhibit significant upward trends with magnitude of 6.8 days per season per decade and 114 hours per season per decade (Fig. S4)."*

The time series of the number of "concurrent warming events" per season is shown in Fig. R4c and also included as Fig. 15c in the revised manuscript.

[Figure]

**Figure R8**. Time series of (a) the number of days per season and (b) the number of hours per season with at least one warming event defined at the grid point level occurring poleward of 80°N.

**Technical corrections:**

26) Line 14: "over the high Arctic (poleward of 80°N) occurred during 1980-2021" Typo: "that occurred"

Corrected.

27) Line 16: "regions" – maybe better: "region"

Corrected.

28) Lines 41-44: "ranging from … and heatwaves" should be "ranging from … to heatwaves"

Corrected.

29) Line 59: "phenomenon" should be "phenomena"

Corrected.

30) Line 68: "an episode of extreme warming event" should either be rephrased to "an episode of extreme warming" or "an extreme warming event"

We have changed it to "an episode of extreme warming".

31) Fig. 3: "The red vertical line in (c) …" You probably mean (a). And I think you can remove the "6.32" in the top left corner of Fig. 3a (or move it to the right to the position of the red line).

Yes, it should be "The red vertical line in (a)..." We have now removed the number at the top of the Fig. 3a (Fig. 4a in the revised manuscript).

32) Throughout the manuscript: I would place the references in chronological order.

The references are now in chronological order throughout the manuscript.

33) Line 301: "likely plays roles" – maybe better: "likely plays an important role"

We have made the suggested change.

34) Line 339 and caption Fig. 14: "for those above 0°C" should be "for grid points above 0°C"

Corrected.

35) Line 370: "experiencing warming event" should be "experiencing a warming event"

Corrected.

36) Line 371: "to its west" should be "to their west"

Corrected.

37) Line 373: "located at southwest of the grid point with warming event" should be "located southwest of the grid point of the warming event"

Corrected.

38) Line 379: I would write "anticyclone dominance type" and "cyclone dominance type" instead of "high/low dominance type"

We have made the suggested change.

39) Line 398: "are thus can be expected": delete either "are thus" or "can be"

We have deleted "can be".

40) Line 403: "AR detection algorithm" should be "the AR detection algorithm"

Corrected.

**Reviewer #2**

**General comments**

This paper presents characteristics of extreme wintertime Arctic extreme events for 1979 – 2021 utilizing hourly ERA5 data. Extreme warm events are defined per grid-point when the hourly near-surface temperature (T2m) reaches or exceeds a threshold of 0 °C in the high Arctic (north of 80°N). The strict spatial and threshold criteria restrict the events only to be found in the Atlantic sector. The authors find that these events are rare and short-lived, usually lasting less than one day. The highest frequency and longest duration of these events are found to be confined to a region close to 80°N and within 0-30°E. These events are further associated with positive anomalies in sensible heat fluxes (largest contribution closest to the lower latitude boundary) and positive anomalies in both IWV and DLW, with largest contributions especially further towards the Arctic interior. Large-scale circulation anomalies with a well-known dipole pattern in the SLP anomalies are found to be favourable for warm and moist air advection into the affected grid-points. A 100% match is found between warming events at northerly latitudes and the co-occurrence of atmospheric rivers (ARs). The grid-point wise defined warming events are then elaborated to regions of several grid-points with temperatures at or above 0 °C, i.e., "concurrent warming events", a method taking into account the spatial extent of such events. Three different large-scale circulation patterns are found to be associated with these regionally defined warming events: a strong dipole in SLP, a blocking-like surface anticyclone or a strong negative SLP anomaly over Greenland. The authors finalize the paper by a trend analysis of grid-point wise defined warming events, where a positive trend in both duration, magnitude and frequency of the events are found. This paper has well-done figures and gives a nice overview of extreme warming events (per grid-point) in the high Arctic, and discusses their drivers with respect to the changes in e.g., SEB anomalies and their relation to large-scale circulation and AR frequency. Most of the results presented here agree and follow results from previous studies. I still have a few concerns regarding the event definition and interpretation of the results presented here. Thus, I suggest minor revision before any possible acceptation of the paper. See comments below for clarifications.

We thank the reviewer for carefully reviewing our manuscript and providing the encouraging, positive comments. The constructive and detailed comments have greatly improved the quality and clarity of this manuscript. Our responses to the specific comments are shown below, with their original comments in blue.

**Specific comments**

- The authors aim to make a climatological record of warm extreme events and associate these to large-scale circulation patterns and ARs. But this is something that has already been done by previous studies, however, with different dataset (ERA-Interim) and event definitions. For example, Graham et al. (2017) (https://doi.org/10.1002/2017GL073395), a study that is also cited in this paper, do not only use in-situ observations, but also look at the historical record winter warming events in the Arctic using different temperature thresholds. They also find a positive trend in duration and occurrence of warming events. Also, your Fig. 1b is similar to their Fig. 2a,

showing maximum 2m temperatures, however they use 6-hourly ERA-Interim data compared to yours hourly ERA5 data. Another study that is completely left out of this paper is Messori et al. (2018) (https://doi.org/10.1175/JCLI-D-17-0386.1), where the drivers behind warm extreme events in the high Arctic are also examined. Please refer to these two studies in your paper for a comparison to similar studies. What is novel in your study compared to theirs? What is the reason for an event definition with a fixed threshold in your study?

Our study is inspired by Moore (2016) and Graham et al. (2017), both of which used buoy observations at different locations (grid points) to examine the evolution of the late December 2015 warming event with near surface temperature (T2m) above 0°C over the high Arctic. Therefore, as noted in our manuscript (line 90 in the original manuscript), defining the warming event at a grid point with T2m >= °C can facilitate a more direct comparison with the two studies mentioned above.

Based on Fig. 2 in Moore (2016) and Fig. 1 in Graham et al. (2017), the duration for the T2m of the warming event staying at or above 0°C is very short and less than a few hours. Based on their two figures, a dataset with high temporal resolution is needed to better characterize these events. Indeed, as shown in our study (line 153 in the original manuscript and Fig. 2), about 70% of these extreme warming events with T2m >= 0°C at a grid point lasted less than 12 hours. A dataset with 6-hourly resolution, which was used in both Graham et al. (2017) and Messori et al. (2018), is thus inadequate to characterize these events. Furthermore, defining extreme warming event at the grid-point scale can also provide new insight into processes that determine the duration of the events. For example, we show that the relative position of the sea level pressure (SLP) dipole is important in determining the event duration. Long duration events are driven by anomalous downward sensible heat flux while enhanced downward longwave radiation seems to play a more important role for short duration events. In comparison, factors determining the event duration are not explored in both Graham et al. (2017) and Messori et al. (2018), possibly due partly to the dataset with coarser temporal resolution (6 hourly) used in their studies. In addition, these new insights cannot be obtained if extreme warming events are identified by spatially averaging the T2m over a certain region and then defining those events with the highest spatial average T2m, which is adopted in Messori et al. (2018). We do agree with the reviewer that the findings in Messori et al. (2018) are consistent with and very relevant to our study. Messori et al. (2018) is now properly cited in the revised manuscript.

- Warming event definition:
  • I also find it a bit concerning, as raised by the first reviewer regarding the event definition, that warming events, as the authors name these extremes, are defined based on a fixed temperature threshold (2-m temperature >= 0°C) and for each grid-point separately. What if grid-points close to each other actually belong to the same event that is affected by the same synoptic weather system? Grid-point defined warming events, using a Eulerian approach, don't say that much about the real nature of the synoptic weather event, as this event, let's call it an AR guided by the large-scale circulation, will move and affect grid-points further away in the direction of the AR, leaving the previously affected grid-point to cool. As the authors point out, there can be temperature

fluctuations in hourly data (so that the temperature shortly drops below 0°C), but still be part of the same synoptic event (which has a different impact on the surface temperature depending on the time scale and location). The extent of these intrusions can then also affect adjacent grid-points at the same time step, however, in your study, these two would be counted as two separate events? The authors explain on L98 the method "interval requirement", and state that your results are not affected by the temperature fluctuations. Could you please re-explain how this was done? Could you think of using a longer time span of 4-6 days (representing the synoptic scale) for your requirement? Thus, having less than two days between two warming events (as stated in L168-170) would most likely just be the same event. How about the close-by grids? How often do the authors find that several grid-points experience temperatures above zero degrees at the same time? One way to avoid these issues would be to maybe increase the time from hourly to daily and check for days where the temperature exceeds a threshold in the grid for at least once or up to a certain percentage within the time? The authors could also consider looking at pre-defined regions instead of per grid-point. In Messori et al. (2018), warm extreme events were defined over the polar cap (as in this study) as daily T2m anomalies computed against a transient climatology (long-term trends and seasonality removed) and area-weighted over the study domain. Furthermore, anomalies were smoothed to remove fluctuations and only events at least one week apart were chosen in order to avoid double counting of the same event. Despite this, when considering large-scale drivers, in a study from Murto et al. (2022; also referred to in the current paper), they could associate up to 3 of the warm extreme events from Messori et al. (2018) to one or two consecutive blocking events, suggesting that these warm events were actually one event but affected by a similar large-scale setting. Did the authors of this current paper try to define events using temperature anomalies? The event definition as it is now is for me a bit problematic, and caution must be taken when interpreting the results of your study. Please also point out in the abstract that events are defined per grid point.

We thank the reviewer for raising these concerns which give us the opportunity to clarify. As we have mentioned in the response to the first comment, our study is inspired by previous studies of Moore (2016) and Graham et al. (2017). They use point observations based on meteorological buoys to characterize the late December 2015 high Arctic warming event. This particular warming event garnered a lot of attention because the T2m during the event exceeds 0 °C near the north pole. To facilitate a direct comparison between our study and their studies, we thus define the extreme warming event as a grid point with T2m >= °C and did not consider defining the events using temperature anomalies. We are particularly interested in those events with T2m >= 0°C.

We agree with the reviewer that when many nearby grid points are experiencing T2m >= 0 °C simultaneously, it is likely that they are driven by the same large-scale circulation. It is thus reasonable to treat all of them as one synoptic event. This is exactly the reason why we also define the so-called "concurrent warming events" later in the manuscript. While identifying a warming event as a contiguous region with T2m >= 0 °C can help us better understand the real nature of the synoptic weather system, there are questions that can only be answered when such events are defined at the grid-point scale using a Eulerian approach. For example, when a large-scale circulation pattern drives the T2m to exceed 0 °C over many nearby grid points at the same time, why is the warming duration

longer at some grid points than the others? This question cannot be answered if all the grid points with T2m >= 0 °C are treated as part of one single event. This definition would inevitably ignore the spatial heterogeneity of the event duration within the covered area. However, we can readily answer this kind of question with the Eulerian approach. We believe that defining the events at the grid-point scale and as a contiguous region with T2m exceeding a predefined threshold are two perspectives on studying warming events. Instead of being mutually exclusive, they complement each other. This is very similar to studying extreme precipitation over midlatitudes. Mid-latitude extreme precipitation is often driven by large-scale weather systems, such as atmospheric rivers (ARs). When a small area (or a model grid point) is experiencing extreme precipitation caused by an AR, it is likely that its nearby grid points would also be experiencing extreme precipitation at the same time. Under such a situation, they are driven by the exact same weather system and thus can be treated as part of one single event. However, mid-latitude precipitation extremes are often studied at the grid point level (e.g., Pfahl et al., 2017). Analogously, we believe that it is also a reasonable approach to define warming events at the grid-point scale.

We recognize that when the onset of an event occurs within a few hours of the termination of the previous event over the same grid point, both events are likely associated with the same weather system, such as atmospheric river or cyclone. We thus imposed that if the onset of an event is within 24 hours or 48 hours of the termination of the previous event over the same grid point, the event would have been excluded from any following analyses. We found that our results are not sensitive to whether these constraints are imposed or not. This is because when the gap in between two events is already 1-2 days, it is likely that they are under the influence of different weather systems (e.g., different ARs or cyclones). This happens when an AR family or sequence of cyclones is steered into the Arctic by the atmospheric blocking pattern. Nevertheless, as suggested by the reviewer, we have tested imposing a 5-day time interval requirement. Again, our results are not sensitive to the length of the imposed time interval (Fig. R9).

[Figure]

**Figure R9**. Temporal evolution of the anomalies of T2m, DLW, SHF, LHF, IWV and temperature advection for all the extreme warming events defined as any grid points with T2m >= 0 °C (a, b), short duration events equatorward of 83°N (c, d), short duration events poleward of 85°N (e, f), and long duration events (g, h). Note that long duration events occur only over regions equatorward of 83°N. These curves are constructed by averaging the temporal evolution of various anomaly terms across all extreme warming events within the different defined groups. There are 96697, 7771, 1181, and 4835 events included in the groups of all events, short duration events equatorward of 83°N, short duration events poleward of 85°N and long duration events, respectively. Day 0 corresponds to the start of the extreme warming event. The shading indicates that the anomalies are significant at the 0.05 level based on the Student's t-test. (Same as Fig. 6 in the revised manuscript, but with a 5-day interval imposed between successive extreme warming events over the same grid point.)

We thank the reviewer for their suggestion on the terminologies used to describe the evolution of the extreme warming events. The term "decay" is not used in our original manuscript. The term "onset" is used to refer to the start of the warming event defined at the grid-point scale. In our revised manuscript, we define the start of the event as "*the event starts when the T2m first reaches or exceeds 0 ℃...*". To avoid any misunderstanding, we have now replaced "onset" with "start" or "the start of the event" in describing the time when a grid point's T2m first reaches or exceeds 0 ℃.

It is true that double counting occurs for Figs. 6-8 of the original manuscript (Figs. 7-9 in the revised manuscript). The impact of same large-scale circulation pattern can be counted multiple times within one composite or across composites. However, as stated by the reviewer, "a time step at the onset of one event could be the same time step as the decay of another event". When the same large-scale circulation causes the T2m to exceed 0 ℃ over many grid points at the same time, these grid points are at different stages of the event "lifecycle". The position of the large-scale circulation pattern relative to the grid point at which the composite is centered thus differs across these grid points. Although the same large-scale circulation is counted more than one time when constructing the composites shown in Figs. 6-8 of the original manuscript, the relative position of the same large-scale circulation differs depending on the centered grid points and the stage of the event "lifecycle" experienced at these centered grid points. As the purpose of the composites shown in Figs.6-8 of the original manuscript is to demonstrate that the duration of the warming events is determined by the position of the large-scale

circulation pattern relative to the grid points with extreme warm events occurring, the double counting here would have minimal impact on our conclusion.

To further show that the impact of the double counting on our conclusion is indeed minimal, we reproduced Figs. 6, 7, and 8 by randomly picking one grid-point scale event at each of those hours with at least one event and compositing all randomly picked events. More specifically, at each hour, we first identify all the grid points with T2m first exceeding 0°C. The number of the grid points with T2m first exceeding 0°C gives us the number of events at that hour. We then record the duration and position (latitude) of all the identified events of that hour and classify them into different categories (e.g., long duration events, short duration events poleward of 85°N, and short duration events equatorward of 83°N). For the categories with at least one event, we then randomly pick one event from each of those categories. The above steps are repeated for each hour during the study period. Lastly, the randomly picked events of each category are used to construct the composites. This approach can ensure that only one event is selected, even when neighboring grid points are experiencing warming events simultaneously. The same large-scale circulation pattern is thus counted only once. As shown in Figs. R1, R2, and R3, the results based on the composite approach described above closely resemble those presented in Figs. 6, 7, and 8. The only difference is that Figs. R1, R2, and R3 tend to be nosier, especially for the composites of the short duration events poleward of 85°N, owing to the smaller sample size. These results thus further confirm that double counting have minimal impact on our conclusion.

To point out this caveat, the following discussion has been added to the manuscript:

"*As the same large-scale circulation pattern can cause extreme warming events to occur over more than one grid point, it thus can be counted more than once within the same composite or across different composites. However, as the composites are centred at the grid point where the extreme warming event is identified, the position of the same large-scale circulation relative to the grid point would thus differ among the composites centred at different grid points. As we show below, the relative position of the SLP anomalies is what determines the duration of the extreme warming events. The double counting of the same circulation pattern for the composites shown in Figs. 7, 8 and 9 thus has minimal impact on the conclusion.*"

- Spatial restriction: Why did the authors decide to restrict to the polar cap (north of 80N), when events are defined from absolute temperatures? Studies found that warm extremes are mainly located in the Atlantic sector (Graham et al. 2017) and associated with strong moist-air intrusion from the Atlantic that penetrate into the high Arctic (Messori et al. 2018). With a more relaxed temperature threshold, as done in Graham et al. 2017, Pacific warming events can also occur. I would like to see some discussion to why Pacific warming events are not included in this study. For example, sea ice extends further south at the Pacific side and thus airmasses from southerly latitudes have a longer path to cool before reaching into the polar cap compared to Atlantic pathways. A northerly ice edge at the Atlantic side, on the other side, allows the air to collect moisture and heat for a longer time and distance before losing them while traveling northwards. Storm tracks might also be more active over the Atlantic side. In Graham et al. (2017), a southerly latitude band was also chosen for the Pacific side compared to Atlantic to include both sectors with

warming sources. Studies also show that ARs from the Pacific side, such as in 2007, can have an important impact on the temperatures in the Arctic. Southerly winds promoted ice-export and the warm and moist air transport over the Beaufort Sea enhanced surface temperatures, and led to anomalous SEB fluxes and the onset of sea ice melt (e.g., Graversen et al. 2011 https://doi.org/10.1007/s00382-010-0809-z and Stroeve et al. 2008 https://doi.org/10.1029/2008EO020001). Please add a bit more discussion around the reason for the chosen study area (distance to sea ice, importance of both transport pathways...) in the introduction and around L147.

We thank the reviewer for their suggestions on the warming events over the Pacific sector of the high Arctic. The Pacific sector is certainly another important pathway for the intrusions of moisture and heat into the high Arctic. As we have pointed out in the response to the first comment, our study is inspired by Moore (2016) and Graham et al. (2017). To facilitate a direct comparison between our study and their studies, we thus restrict our focus only to the high Arctic poleward of 80°N and those events with T2m >= 0°C. More explanations on why we restrict our study to the high Arctic and focus only on those events with T2m >= 0 °C over a grid point are provided in Section 2.1 of the revised manuscript. The suggestions of using more relaxed threshold and spatial restriction to enable a robust comparison of warming events between the Pacific and Atlantic sectors would be great for a separate study.

"*Case studies based on in-situ buoy observations have been conducted by previous studies to examine the characteristics and drivers of winter extreme warming events over the high Arctic (Moore, 2016; Graham et al., 2017), which characterized extreme warming events using point observations based on meteorological buoys. Since these extreme warming events are usually driven by large-scale circulation (Woods and Caballero, 2016; Moore, 2016; Kim et al., 2017; Messori et al., 2018), when extreme temperature is detected over a grid point, it is common that such extreme temperature can also be found over its neighbouring grid points. Under such conditions, one approach is to define extreme warming events by using contiguous regions with temperature exceeding a predefined threshold. However, to facilitate a more direct comparison with previous case studies shown in Moore (2016) and Graham et al. (2017), here we choose another approach to define extreme warming events at the grid-point scale. Specifically, in this study, extreme warming events are defined as those events with 2-meter air temperature (T2m) over a grid point reaching or exceeding 0°C over the high Arctic that covers the regions poleward of 80°N.*"

- Concurrent warming events: (Sect. 3.3). For me, this method seems much better, as, to my understanding, these events are defined by finding areas in the Atlantic sector where several grid-points at one timestep satisfy the criterion for warming events. What is the region where these events occur (maybe show on a map figure)? Are these areas coherent, i.e., that the grid points are adjacent to each other? What are the main characteristics for these events (number of points included in one area, duration, trends, seasonal evolution, spatial frequency…)? The lifecycle perspective of these events is again utilized here (onset and decay), which rises similar questions as stated above. How

is the decay defined (when the temperature for all or at least one grid point within the area drops below zero)? Again, as stated in L258, close-by areas can also be affected by the same weather event, and here the authors decide to use a 5-day temporal limit, which is good. How is the temporal criterion applied here (on warming events (grid-points) within a region or for separate regions)? The definition of the peak of these events is somewhat unclear: on L263 it is referred to as the hourly time when the area of the grid-points with T2m>=0°C is the largest, but at L281 the authors refer to the peak time of the T2m anomaly when referring to Fig.10. Does the time at maximum area of grids satisfying the criterion always correspond with maximum T2m anomalies? Please clarify. If this is not the case, maybe an intensity measure where both the spatial extent (area) and the average temperature anomaly (magnitude) could be used to define the peak.

As shown in Fig. 1b, only a relatively small region over the Atlantic sector of the high Arctic ever experienced T2m above °C (outlined by the magenta contour). Since the concurrent warming events occur when the T2m reaches or exceeds 0 °C over many grid points concurrently, this small region defines where these events actually happened.

Since our study region is small, when the T2m reaches or exceeds 0 °C simultaneously over many grid points within such a small region, it can be expected that they are driven by the same large-scale circulation and the grid points with T2m $>= 0$ °C are thus adjacent to each other. There are 96 concurrent warming events identified in our study. We manually examined many of the events and confirmed that these events are spatially coherent (not shown).

Time series of the seasonal occurrence frequency, seasonal mean peak area, seasonal maximum peak area, seasonal mean duration and seasonal maximum duration of the 96 concurrent warming events are shown in the updated Fig. 15 in the revised manuscript and shown above as Fig. R4. All these quantities exhibit significant and positive trends in the past four decades.

For these concurrent warming events, their lifecycle (onset, intensification, and decay) reflects the lifecycle of driving large-scale circulation. We believe that it is reasonable to describe the evolution of these events using the lifecycle perspective. For example, the lifecycle perspective has previously been used to characterize the evolution of synoptic sea ice loss events over the Arctic or a predefined subregion within the Arctic (e.g., Wang et al., 2020; Park et al., 2015).

We did not formally define the decay of concurrent warming events in the manuscript, which refers to the stage when the total area with T2m $>= 0$ °C reaches maximum and starts to decrease. The termination of the concurrent warming events is defined as the time when the total area with T2m $>= 0$ over the high Arctic first drops to zero. The onset of the concurrent warming events refers to the time when the total area with T2m $>= 0$ °C over the high Arctic first rises above zero. If the time interval between the termination of the previous event and the onset of the following event is less than 5 days, we exclude the following event from the analyses. We have updated the manuscript to better explain how this time interval constraint is imposed.

"*We thus impose a constraint that the time interval between the onset of one event and the termination of the subsequent event needs to be longer than five days. Otherwise, the subsequent event is discarded in our analysis of large-scale circulations.*"

The peak of the concurrent warming events is at the time when the total area with T2m >= 0 °C reaches maximum. The definition noted in line 263 of the original manuscript is the corrected one. We have changed the line 281 in the original manuscript from "*compared to the peak time of the T2m anomaly composite of all the events*" to "*compared to the T2m anomaly composite at the peak time of all events*" to avoid confusion.

The timing of the maximum T2m anomaly depends on the region defined over which the spatial average is taken. When the spatial average T2m anomaly is taken over the region between -15°W and 60°E  and poleward of 80°N, which roughly corresponds to the region ever experienced T2m >= 0 °C shown in Fig. 1b, we found that 92 (82) out of the 96 identified concurrent warming events have their peak area occurred within 24 (12) hours of the timing of the maximum T2m anomaly. This suggests that the timing of the peak area of the concurrent warming events corresponds well with the timing of the maximum T2m anomaly over the Atlantic sector of the high Arctic.  This correspondence is also reflected in Fig. 9 of the original manuscript (Fig. 10 in the revised manuscript). As can be clearly seen, both SLP anomaly pattern and IVT anomalies reach maximum at day 0, which is the time with the largest area of T2m >= 0 °C. This suggests that defining the peak of the concurrent warming events based on the total area with T2m >= 0 °C is a reasonable approach. The following discussion has been added to Section 3.3 in the revised manuscript:

"*Further analyses show that the timing of the peak area corresponds well with the timing of the maximum T2m anomaly averaged over the Atlantic sector of the high Arctic (-15°W – 60°E and poleward of 80°N, roughly corresponding to the region ever experienced T2m >= 0°C shown in Fig. 1b). For example, 92 (82) out of the 96 identified concurrent warming events have their peak area occurred within 24 (12) hours of the timing of the maximum T2m anomaly.*"

- How much does the surface type (ocean, sea ice, leads in sea ice) affect the spatial location of the warming events, and the anomalies in e.g., SEB? Did the authors consider to divide the warming events into ocean and sea ice (with different SIC's) – this could improve the quality of the paper and the interpretations of the results. L185 states that the climatological SHF is upward in winter, which is true over warm ocean or openings in the sea ice. But over sea ice, I think that the climatological values of turbulent fluxes are almost negligible, if not slightly positive. The surface type influence can nicely be seen in S2, with a sharp division between negative and near neutral values at the sea ice edge. Thus, L187 "suppression of the upward THF" is true if the surface is relatively warm, but ARs could also enhance downward THF if the surface is relatively cold (e.g., over sea ice). Discussion about the impact of different surface types on the results shown here would be nice to see and adding the sea ice edge on the climatological plots would be helpful.

Although it is not stated explicitly in our manuscript, the reason why we divide the short duration events into two groups, one occurring poleward of 85°N and the other occurring equatorward od 83°N, is because we want to understand how the surface type (fully covered by sea ice versus partially covered by sea ice) would affect the SEB during the warming events. As shown in Fig. 5 of the original manuscript (Fig. 6 in the revised manuscript), downward longwave radiation (DLW) dominates the SEB for the short duration event poleward of 85°N while both DLW and sensible heat flux (SHF) play comparable roles for the short duration events equatorward of 83°N. In line 185 of the original manuscript, when we state that "climatologically, the THF over these regions is upward…", we are actually referring to the regions close to 80°N (line 183 in the original manuscript) that are partially covered by sea ice. We agree with the reviewer that the THF is negligible over the fully sea ice covered regions and enhanced downward THF could occur when warming events occur or ARs are present over these regions. We have now explicitly pointed out that the "*suppression of the upward THF*" occurs "*over these partially sea ice covered regions*" in the revised manuscript. We also provided a more detailed explanation on why we divide the short duration events into the two groups.

"*As shown above, extreme warming events can be divided into the DLW dominated ones that occurred over the fully sea ice covered regions poleward of ~83°N and SHF dominated ones that occurred over the partially sea ice covered regions equatorward of ~83°N (Fig. S2).*"

A climatological 50% sea ice concentration contour has been added to the updated Fig. S2 in the supplementary and shown in Fig. R7.

- Wording and terminology for the warming events: The authors are not always consistent with the wording of their events: warming events or extreme warming events for the grid-point wise defined events, and concurrent warming events or large-scale events (L378) for a larger region.

Please be consistent with the terminology, and always refer to which events (region or grid-point) the analysis shown in a figure or discussed in the text is referring to. For example, add it to the abstract and conclusions, as well as in figure captions (such as in Fig. 5 for the duration categories). In the abstract it is, for example, not clearly stated that the match between ARs and warming events are done on the concurrent events.

Thank you for the suggestion. We have combed through the manuscript to ensure that the terminologies and wordings are used consistently. "Extreme warming event(s)" is now consistently used to describe those events defined at the grid-point scale. And "concurrent warming event(s)" is used to describe those events with more than one grid point experiencing extreme warming events simultaneously. We have also explicitly defined the events in figure captions.

- The word "driving" or "driver" is used several times in the paper. I would rather write "associated with" or "related to" instead of driving. For example, in L300, I would rather write that the SLP pattern are "guiding" the ARs or "making a pathway".

We have made the suggested changes at places where we believe are applicable.

- Figure 2c and Fig. 3a: remove the value shown in the upper left corner but keep it in the captions. Also, I assume the caption for Fig. 3 for the red line should refer to panel (a), not c. I would also suggest to change the color for zero in Fig. 3b to another color to be clearer (blue according to the colorbar means no days or actually more than 0?).

Yes, we are referring to Figs. 3a not 3c for the red line mentioned in the caption. Blue in Fig. 3b means the values are close to zero. Zero value has been masked out. The following sentence has been added to the caption of Fig. 3 (Fig. 4 in the revised manuscript) for clarification.

"*Grid points never experienced an extreme warming event or less than two extreme warming events within a single winter (thus with a mean time interval of zero) have been masked out in (b).*"

We have updated Figs. 2 and 3 as suggested.

- I suggest the authors to add more references, at least when discussing the results of this paper and when comparing directly to other observations or studies (e.g., in L153).

More references have been added to the result section to provide the context for discussion.

- At L158: I assume the authors mean that the duration drops gradually away from the region at 80N, 0-30W to less than 5h north of 85N OR east of 60E (there are no data in the corner north of 85N and east of 60E, right?)

Sorry for the typo. It should be "*Over regions eastward of about 60°E and poleward of about 80°N...*", not 85°N.

- Figure 4: Would be interesting to see how many grid-points satisfy the temperature criterion over all seasons within the study period, to know if the average e.g., DLW is a result of only a few or several events. The average occurrence of these events (S1) gives some idea about this, but maybe a relative frequency plot would better demonstrate this (so that 100% would refer to the max occurrence of events per grid point over all periods). I also find the Figure S1a to be relevant for the main paper, as main characteristics for events include spatial distribution.

Fig. 12b in the original manuscript is the relative frequency plot that shows the fraction of time with T2m >= 0°C over all wintertime hourly snapshots during 1979-2021. The relative frequency is very close to zero over most of the regions that have experienced extreme warming events. Fig. 4 in the original manuscript is obtained by taking the average of all anomaly terms over all hours with T2m >= 0 °C over the given grid point. To better visualize the sample size that Fig. 4 in the original manuscript is based on, Fig. R5 shows the total number of hours with T2m >= 0 °C for the entire study period. As shown in the figure, the number of hours with T2m >= 0°C decreases rapidly as moving away from the region with the maximum occurrence hours. Near the edge of the regions that experienced T2m >= 0 °C, the number of hours with T2m >= 0 °C is usually less than 5. The following discussion based on Fig. R5 has been added to Section 3.2 of the revised manuscript.

"*The total number of hours with T2m >= 0°C over the study period reaches its maximum (more than 1000 hours) over the region near 80°N and between 0°-30°E (Fig. S1). Moving away from this region, the number drops rapidly towards the boundary with regions that have never experienced T2m >= 0°C during winters of the past four decades.*"

Fig. R5 has been added to the supplementary as Fig. S1. We have also moved Fig. S1 to the main text of the revised manuscript as Fig. 3.

- I find it interesting to see that you find a relationship between the duration of the events, their locations and associated SEB anomalies. Have you performed any correlation analysis to make your statement stronger? I assume that we would then find higher correlation between longerlasting events (longer duration) and SHF anomalies, whereas shorter-lived events would be correlated with DLW, especially at higher latitudes. Intuitively, I would think longer lasting events are those that penetrate further into the Arctic, but this confusing comes only because of your different way of defining events and their duration (at a fixed point; your longer-lasting events are closer to the warm and moist air source). Maybe worth reminding the reader again what your definition is to avoid possible confusion.

We haven't done any correlation analysis. We have shown that the duration of the extreme warming events depends on their position relative to the SLP dipole. It would be difficult to conduct a correlation analysis between the duration of the extreme warming events and their location relative to the SLP dipole due to the lack of an appropriate metric for measuring such relative location.
We also show that the SEB of the long duration events is dominated by SHF while the SEB of the short duration events is dominated by DLW poleward of 85°N. However, the duration of the

extreme warming events is not necessarily well correlated with the magnitude of the SHF anomalies or DLW anomalies at the start of the events. The location where the extreme warming events occur would confound this correlation. For example, during the start of the long duration events (Fig. 5g in the original manuscript), DLW anomalies are positive. However, the correlation between the duration of the long duration events and the magnitude of the DLW anomalies is negative (about -0.12, with a negligibly small p value). This negative correlation is probably due to the fact that longer duration events are more likely to occur at lower latitude regions closer to 80°N where the DLW anomalies are relatively weak (Fig. 4c in the original manuscript). This negative correlation makes it more difficult to interpret the contribution of DLW anomalies to the SEB of the long duration events. Furthermore, all the short duration events defined in our study have a duration of one hour. Conducting a correlation analysis between the duration of the short duration events and the SEB anomaly terms is not feasible. Although we do find a positive correlation between the duration of the long duration events and the SHF anomalies averaged over the first four days of the events (~0.46 with a negligible p value), in order to keep the manuscript more concise and avoid possible confusion, we have decided not to present the results of the correlation analysis. Definition for the extreme warming event has been provided in the appropriate figure captions to avoid possible confusion.

- When talking about anomalies in SLP or SEB components, for example, we are mainly interested in the sign of the anomaly. Therefore, I would suggest writing "positive/negative" instead of "high/low" when writing about anomalies.

The terms "high" and "low" have now been replaced by "positive" and "negative", respectively.

- Figure 5: what is the absolute number of events within each sub-category based on duration? A map showing the locations of these grid-points would be nice to see. Further point out in Fig. 5 caption that the durations are defined per grid-point. Could the stronger anomalies shown in Fig. 7 compared to the other ones be a result of less events included in the composite?

The total number is 18586, 1642, and 10097 for the short duration events equatorward of 83°N, short duration events poleward of 85°N and long duration events, respectively. The information about the numbers has now been provided in the figure caption. We have also pointed out that the durations are defined at the grid-point scale in the caption. Since exceeding the 0℃ threshold represents a more extreme condition over the high latitude region, the stronger anomalies shown in Fig. 7 of the original manuscript are thus the result of less, but more extreme events included in the composite. Spatial distributions of the extreme warming events included in each sub-category are shown below (Fig. R10).

[Figure]

**Figure R10**. Spatial distribution of the occurrence frequency of the extreme warming events for the (a) short duration events equatorward of 83°N, (b) short duration events poleward of 85°N, and (c) long duration events.

- Figs 6-8: these figures are nice but also a bit messy with so many black lines. I would suggest changing the color for the SLP anomaly contours to purple (as in Fig. 13a).

We have updated these figures as suggested.

- How do the authors think that the spatial patterns of the events (or occurrence and duration) might directly be affected by cyclones or blocking? Adjacent grid-points could be captured by the same cyclone but affected by either the cold advection or the warm sector (L206). Have you looked at cyclone tracks or used blocking detection algorithms to associate your warming events with them (instead of using ARs and SLP anomalies to represent the large-scale setting)? The authors also write "blocking" (L380) in the conclusions despite not using a blocking detection algorithm (?). I would suggest rewriting to "blocking-like structure" and maybe refer to other studies (e.g., Woods and Caballero 2016, Messori et al. 2018 and Murto et al. 2022).

We think that the interaction between cyclones and blocking is what ultimately causes the occurrence of the wintertime extreme warming events and determines their spatial patterns in the high Arctic. As shown in Figs. 7-9 in the revised manuscript, the relative position of the SLP dipole plays a key role in determining the duration of extreme warming events. Furthermore, as shown in Figs. 10-11 in the revised manuscript, different configurations of the SLP dipole affect both the magnitude and spatial distribution of the T2m anomalies. These results all suggest an important role played by the interaction between cyclones and blocking in shaping the spatial patterns of the concurrent warming events.

In this study, instead of focusing on the role of cyclones and blocking that has been examined by previous studies, such as those pointed out by the reviewer, we choose to focus more on the role of ARs. Ultimately, the way via which cyclones and blocking cause the occurrence of the extreme warming events is through guiding the Arctic ARs and their associated heat and moisture into the Arctic. In this sense, we can view ARs as the direct driver of the extreme warming events. We agree with the reviewer that it is more appropriate to describe the positive SLP anomalies as "blocking-like structure" since we didn't use a detection algorithm to explicitly identify atmospheric blockings. We have replaced "blocking" with "blocking-like structure" or "positive sea level pressure anomalies which resemble blocking" in the revised manuscript.

- In sect. 3.4, why do the authors return back to the grid-point defined warming events to relate them to ARs? Have you looked at how your results would change when using the concurrent warming events?

The link between ARs and the concurrent warming events is shown in Fig. 10 of the original manuscript (Fig. 11 of the revised manuscript). Figs. 10e-h show the AR frequency, which is defined as the fraction of time when a grid point is under AR conditions, during the peak of the concurrent warming events (shaded contours). Compared to the climatological AR frequency that ranges from 0.5-2.5% (solid line contours), the AR frequency during the peak of the concurrent warming events increases substantially, over 30% at some grid points, suggesting that ARs were present over those grid points for more than 30% of the 96 concurrent warming events identified in this study.
As have been pointed out in our response to the reviewer's previous comments, our study is inspired by Moore (2016) and Graham et al. (2017). One of the foci is to understand the connection between wintertime extreme warming events defined at the grid-point scale and Arctic ARs.

- L325: are these nine days of T2m above zero degrees found here in your study? Please mark their locations on a map, e.g., in Fig. 13a. Have you studied the origin of the ARs or utilized the tracking algorithm of the AR shapes, if it was provided with the AR algorithm?
- Figure 10: Are the T2m anomalies shown for all grid-points independent of if the grids are part of a concurrent warming event (or only that the time is at the peak of the event)? Maybe some density lines would be helpful to show where these concurrent warming events are spatially located and how these anomalies extend wrt the originally defined events.

Yes, these nine days with T2m above zero over regions poleward of 85°N are found in our own analyses based on ERA5. As shown in Fig. 1b, the region with T2m >= 0°C poleward of 85°N is very small and confined between 30°W and 60°E. Exact information about where they occur can be found in Fig. R5 shown above.

The AR detection algorithm used in our study has the capability of tracking ARs through space and time. We do think that it is interesting to look at the origin of the ARs affecting the high Arctic. However, since our focus is to investigate the link between ARs and extreme warming events in the high Arctic, rather than the characteristics of ARs, we have decided not to pursue this aspect further in the current study, which should be an interesting topic for a dedicated new study.

T2m anomalies shown in Fig. 10 are the mean T2m anomalies during the peak of the concurrent warming events. They are independent if the grids are part of a concurrent warming event. We have also updated Fig. 10 of the original manuscript (Fig. 11 in the revised manuscript and reproduced below as Fig. R11) to show the fraction of time when a grid point with T2m >= 0 during the peak of the concurrent events (solid line contours in Figs. R11a-d).

[Figure]

**Figure R11**. Spatial distribution of mean T2m anomalies and AR frequency during the peak of the concurrent warming events. The peak of the concurrent warming events is defined as the time when the areas with temperature above 0°C reach maximum. The purple line contours in (a)-(d) depict the fraction of time when the T2m over a grid point reaches or exceeds 0°C during the peak of the concurrent warming events and (e) – (h) the climatology of winter AR frequency.

- One additional concern is the statement at L318: "ARs are the only weather system capable of triggering the occurrence of the warming events". ARs are definitely important! That grid-points are co-occurring in time and space with ARs (even though 100% overlap), does not, however, directly imply that these warm anomalies can only result from an AR. In a recently published paper (Papritz et al. 2023 https://doi.org/10.1175/JCLI-D-22-0883.1), the relative contributions to the warm potential temperature anomalies extending in the whole tropospheric column (associated with extreme positive SEB anomalies over wintertime Arctic sea ice) were investigated. They found (using backward trajectories) that only airmasses ending up in the middle troposphere had an AR-like evolution, whereas airmasses making up the positive anomalies closest to the surface actually had an Arctic origin! These airmasses were either warmed diabatically while crossing over warmer oceans or when airmasses descend from higher altitudes, but all within the Arctic. It was these two airmasses together that could give rise to these anomalous positive vertically extending potential temperature anomalies. Local processes are thus also important, so I would suggest rewriting this strong sentence and add "likely" or "strong impact" instead of stating ARs are the only driver.

Thank you for the suggestion and pointing us to this interesting and relevant paper. We have rewritten the sentence and properly cited the paper mentioned above.

"*The results here thus suggest that, for a large fraction of the regions where extreme warming events can occur, the presence of ARs and their impact on and interaction with the local environment (Papritz et al., 2023) likely exert a strong control on the occurrence of these events.*"

- Are the trends discussed in Sect. 3.5 based on grid-point defined warming events? Referring here to an absolute number of warming events (2150) does not tell the reader so much, as your events (and the number) are dependent on the grid-size, the temporal resolution of the data etc. Maybe more informative would be to show seasonal trends of days with atleast one warming event per grid point? Or rather use the concurrent warming events here and calculate their trends per decade. I also think that the event definition (a fixed temperature threshold) and with a rapidly warming Arctic, warmer temperatures become more common by default (compared to the first decade of your study period).

Yes, the trends discussed in Section 3.5 are based on the grid-point defined extreme warming events. We agree with the reviewer that the absolute number of the trend in event count depends on both the spatial and temporal resolution of the dataset. However, the exact number of this trend is not very important, what matters more is that it shows that the area with T2m >= 0°C has been increasing in the past four decades. We have included a new figure in the supplementary to show the time series of the number of days per season and the number of hours per season with at least one extreme warming event defined at the grid-point scale occurring poleward of 80°N (Fig. R8). The trends in both time series are significantly positive. We have also updated Fig. 14 in the original manuscript (Fig. 15 in the revised manuscript) to include the trends in various characteristics of the concurrent warming events. We found that the frequency, spatial extent and duration of the concurrent warming events all exhibit positive and significant trends in the past four decades.

- Figure 13c is not referred to in the main text

Lines 333-334 in the original manuscript refer to the results shown in Fig. 13c. We have now made this sentence explicitly referring to Fig. 13c.

- I am also lacking a final concluding statement. The authors nicely summarize the findings of the paper in the final section, and list some limitations. What potential of further studies would your study contribute? One way to tie the final section to the rest of the paper is to answer the questions raised in the introduction in the conclusions.

Thank you for the suggestion. The following paragraph has been added to the revised manuscript as the last paragraph to discuss the implication of our study.

"*Given the critical roles played by the SLP dipole and ARs in determining the occurrence and characteristics of the wintertime extreme warming events, it is important to understand their variability at different timescales and identify large-scale climate modes that are responsible for such variability. An improved understanding on the variability of the SLP dipole and ARs would likely lead to a better understanding and prediction of the Arctic climate across timescales. The results in this study also suggest that a correct representation of the SLP dipole and ARs is key to simulating the occurrence of extreme weather events over the high Arctic at synoptic timescale. As we rely on climate models for future Arctic climate projection, further research is needed to evaluate how well climate models can faithfully represent the SLP dipole and ARs that affect the*

*high Arctic. In this study, we focus on the T2m extreme warming events. It is expected that warming events as such would have a considerable impact on the underlying sea ice. As have been shown in this study, these extreme warming events tend to cluster in time. It would be interesting to further investigate their cumulative effects on the longer-term sea ice growth and the subsequent sea ice melt in the following summer. If a link between the occurrence frequency of such warming events and the subsequent summer sea ice minimum can be established, an improved prediction of the SLP dipole and ARs mentioned above would likely further extend the prediction lead time of summer sea ice minimum.*"

- Please add some discussion about ERA5 warm bias to the limitations of this study (relate to the representation of snow and sea ice in ERA5, see e.g., Batrak and Müller 2019 https://doi.org/10.1038/s41467-019-11975-3).

Thank you for pointing us to this interesting and relevant paper. Although the warm bias in ERA5 has been briefly touched upon in Section 2.1 when we introduce the dataset, this paper provides a nice explanation for the causes of the warm bias. Fig. 2a in their paper clearly shows that this warm bias is state-dependent. The warm bias is only pronounced under clear-sky conditions and slightly negative or negligible during anomalously warm periods. This is consistent with what we have discussed in lines 105-107 in the original manuscript: "*However, a recent study points out that the warm bias in ERA5 may be state-dependent, with a positive bias found under radiatively clear condition and a negative bias under opaquely cloudy condition (Herrmannsdörfer et al., 2023).*" This paper has now been cited at the end of the above sentence. In addition, the following discussion has been added to the second-to-last paragraph in the revised manuscript:

"*For example, due to the misrepresentation of sea ice thickness and the absence of snow layer on top of sea ice in the numerical models, reanalysis products, including ERA5, suffer a warm bias over the wintertime ice-cover Arctic under radiatively clear condition (Batrak and Müller, 2019).*"

**Technical corrections**

I agree on the typing errors pointed out by the first reviewer. Below some minor correction suggestions in addition to them:

- Add a reference to panel (f) for the temperature advection in the figure (Fig. 11) caption
Added.

- I would suggest to add the latitude threshold in the caption for the long-lasting events in Fig. 5g, h
We have explicitly pointed out in the caption that long duration events occur only over regions equatorward of 83°N.

- Be consistent with the terminology used in the text and in the figures. In Figs. 5a and 14a, "TS" is used instead of "T2m" (I assume), it is always referred to as T2m in the main text.
Thank you for pointing this out. We have replaced "TS" with "T2m" in these two figures.

[revised manuscript text omitted]

---

## Referee Report (RR1)

Review of "Wintertime Extreme Warming Events in the High Arctic: Characteristics, Drivers, Trends, and the Role of Atmospheric Rivers" by W. Ma et al.

General comments:

The authors have put a lot of effort into the revisions and have addressed many of my previous concerns. The results are interesting and provide insight into the characteristics of extreme Arctic warm events, defined first per grid point and then as contiguous regions with T2m > 0°C (concurrent warm events). I particularly like the analyses of the concurrent warm events. My major concern is still the same as in the first version of the manuscript, namely the definition of Arctic warm events per grid point (see comment 1 below). Many of my remaining comments concern language and terminology and can be addressed with some rewording.

Specific comments:

1)  You have explained in detail why you define Arctic warm events at each grid point separately, but I am still struggling with the fact that adjacent grid points with T2m > 0°C at the same time are referred to as many separate "events". You motivate this with the studies by Moore (2016) and Graham et al. (2017), who used meteorological buoy observations to study the warm event in December 2015. However, even if they looked at different point observations, they did not refer to them as different Arctic warm events – they used the point observations to characterize one single event. I am ok with your method (although I am more convinced by your definition of concurrent warm events, which, however, is not the main focus of the paper), but the problem for me is the word "events" that does not always fit well in the text and can lead to confusion. Maybe you can carefully scan your manuscript and write more often something like "grid points with T2m > 0°C" instead of "Arctic warm events". For instance, you write about the onset of the event (or, in the revised version, the start of the event, which for me is actually the same as the onset) and its termination, but I find the suggestion of the second reviewer much more convincing to use something like "time when a grid point's T2m exceeds / falls below 0°C", as the warm temperatures are most likely simply being advected from one grid point to the next. And instead of short-duration and long-duration events, maybe you could use something like "grid points experiencing the warming for a short / long duration". I understand that this also makes the text more cumbersome, and you can't replace all your mentions of "event" with a formulation like this, but wherever possible this would increase the accuracy of the text and help to minimize confusion for the reader.

2)  I find your new Fig. S4 very interesting, but I am quite surprised by the large number of days per season with at least one grid point with T2m > 0°C. According to this figure, these events are actually not that rare, with most winters having at least one such warm event and many winters even having one in more than 20 out of the 90 days. This is different to the findings of previous studies. For instance, according to Moore 2016, Arctic warm events occur once or twice each decade, and Binder et al. 2017 found 12 out of 36 winters with Arctic warm events. Of course, these studies are based on different methods, different datasets, different study regions, etc., but I am still surprised by the quite different findings. Maybe you could briefly discuss your results and explain why you have a much higher number of events than found in previous studies.
    Also, your sentence in line 16 in the abstract ("They occur rarely, with a total absence during some winters over most of the region.") and the text in Section 3.1 might be revised a bit. And in my opinion, it would be more informative if you could write in how many of the 42 winters there has been at least one Arctic warm event somewhere in the studied region (your new Fig. S4).

3)  Section 2.1: Even though you rewrote large parts of the section, it remains quite difficult to read and to quickly get the relevant information, as you mix in information that rather belongs to the introduction or even the conclusion, and you write a lot about how others defined the events and how the events could be defined, before actually writing how you define them in a different way. My suggestion would be to write a first subsection with the title "Data", where you describe the

ERA5 data, then a second subsection "Definition of extreme Arctic warming events", both in a more streamlined form, and a final subsection "Calculation of anomaly fields". For instance, when describing ERA5 I would leave away the first sentence "Results based on previous case studies suggest that extreme warming events with near surface air temperature above 0 °C tend to be short-lived, …", as you already mentioned this in the introduction and here it rather disturbs the flow. And also the last sentence of the paragraph "The results being presented here thus bear significance to our understanding of …" does not really fit into the data and method section but is rather part of the motivation or even the conclusion of the study. And in the subsection where you define the events, I would start with writing something like "In this study, we define Arctic warm events in two ways: first, as grid points with T2M > 0°C, which is the main focus of the study, and second, as contiguous regions with T2m > 0°C … " (and please also add more details about how you define the contiguous events). Afterwards, it is nice if you write about the motivation of defining the events in these ways, and their advantages and disadvantages.

4) I am not always convinced by the words "drivers", "direct drivers" and "directly driving", which are all used very often in the paper (e.g., abstract line 23, Section 3.2 and Section 3.4). In general, there is not just one driver, but it is the interplay between various processes and weather systems that contributes to an Arctic warm event. For instance, in line 23 in the abstract, you write that "ARs are the direct driver for 100% of these event", but I would rather use "associated with" or "related to" (and similarly in many other parts of the text). You already changed the words in some parts of the paper in the revised version, but they still appear in many places where I don't think that they are suitable.

5) "Subsequent examination reveals the short-lived nature of this event, with the duration of staying above 0°C for less than an hour locally …"
This sentence is not so nice – I would write something like "with buoys close to the North Pole recording temperatures above 0°C for less than an hour …"

---

## Referee Report (RR2)

**Review notes for the revised manuscript to ACP: "Wintertime Extreme Warming Events in the High Arctic: Characteristics, Drivers, Trends, and the Role of Atmospheric Rivers" by Ma et al.**

I highly appreciate the effort of the authors to address both mine and the other reviewer's concerns and feedback, especially regarding the warm event definition and clearly stating which events are discussed in each section of the manuscript. I am pleased to see that some of our small suggestions were also adapted immediately in the revised manuscript. Thus, I find that the manuscript has now improved a lot. After addressing a few questions and suggestions presented below in terms of a minor review, I am happy to accept this manuscript for publication to the journal. The line references are wrt the reviewed manuscript.

**Specific Comments**

- Despite my concerns on the event definition as a grid-point defined event, the authors have succeeded to explain the pros and cons on their selected method. It is true, that grid-point defined warming events favour that one can understand more about what processes affect locally a specific grid point. Thereby, the spatial heterogeneity of characteristics, such as event duration, will be revealed, which might not be the case when looking at events over larger areas. This is also true when investigating e.g., why some grid-points experience earlier melt or freeze-up onset compared to other grid-points, something that could be missing if enlarging the study area. And I agree when the authors write that the large-scale setting differs relative to the relative location of the grid-point that experience warming. The relation to large-scale patterns can still be obtained if using area-based event definitions, but might not indicate as clearly the importance of the relative location. Anyways, thanks for all additional discussions and clarifications regarding the event definition, this will be helpful for the future readers. However, I have a short comment on the discussion authors provide on the temporal requirement between events (which is not obtained in the current analysis), and my concerns of double-counting of similar events. Figure 6 shows the temporal evolution in lag composites for your grid-point events where no interval requirement between consecutive events is applied. In the response documents, the authors provided a similar figure (R9) with a 5-day interval requirement and indicated that the composites do look alike. I don't disagree with the authors, but would like to draw attention to the long-lived events (R9g) with notable differences between the original figure wrt the turbulent heat fluxes (negative values instead of around zero) and IWV around lags -5 to -2. This would indeed suggest on somewhat double counting of events in Fig 6: are these differences only a result of less events included in R9g, or is this really a result of that the previous long-lasting event was actually part of the begin of the next long-lasting event? Maybe one additional notation of this possibility would be good in the manuscript, where Fig 6 is discussed and/or where the authors motivate reasons why not imposing the temporal constraint (around L122).
- It is nice to see the additional motivations in dividing your events into poleward and equatorward of ~85N based on the surface types. I was wondering whether the trends in any characteristics shown in Fig. 15 (thanks for the additional subplots for the concurrent events!) take into account this division (into "surface type" based on spatial restrictions)? Would the trends look different in the sub-categories?
- Abstract: I would suggest adding "2-meter" temperature (L12) as the temperature variable.
- New final paragraph: Thanks for adding this at the end to summarize and indicate possible further research topics. However, there are indeed some previous studies that show e.g., the relationship between persistent atmospheric circulation in March and the minimum sea ice extent in September (see e.g., Kapsch et al. 2019: https://doi.org/10.1007/s00382-018-4279-z), and the preconditioning of warm winter/delayed freeze-up in autumn for thinner ice next spring (e.g., Stroeve et al. 2018: https://doi.org/10.5194/tc-12-1791-2018). In

comparison to these papers, a good addition would be to study these links in the light of your identified warming events, as the authors nicely point out.

- The ECMWF is written out in two sections close to each other (at the end of Introduction and in the beginning of Methods). I would include it when it is mentioned the first time, and just use the acronym when it is mentioned the second time.

- L284: The location of the negative SLP anomalies to the southeast of the events in long-lasting events could also hint for a NAO+, a circulation pattern that would favour moisture and heat transport across the Atlantic towards the Scandinavia. The blocking-like persistent positive SLP anomalies over Scandinavia/Urals would then deflect that airflow northwards, towards the warm events. Studies also show that the decay of a NAO+ pattern could lead to an enhancement of the blocking to the east of the NAO pattern, and a warm anomaly in the Arctic. As your Fig. 9 is not exactly a geographic map (as you have centred the plots around the event location), these patterns might not be exactly over Iceland (negative SLP) and Eurasia (positive SLP), but would cover about the correct regions given that the possible locations for the extreme warm events are spatially constrained in the Atlantic sector. Have the authors thought about these possible connections wrt the long-lasting events? Some references, e.g., Luo et al. 2016 (https://doi.org/10.1175/JCLI-D-15-0612.1), Luo et al. 2017 (https://doi.org/10.1088/1748-9326/aa69d0), and Murto et al. 2022 (https://doi.org/10.5194/wcd-3-21-2022).

- L379: deep penetration of ARs associated with an SLP center located more polewards. Could this be related to locally formed Arctic cyclones, with cyclogenesis in the Arctic north of Greenland? There are studies, such as Messori et al. 2018 and Murto et al. 2022, that also find these local cyclones (associated with negative SLP anomalies polewards) to occur around the time of the warm events. Maybe worth to discuss this possibility also in the current manuscript?

**Technical corrections**
- Fig1 caption: I would add "in (b)" between "The purple line" and "denotes…".
- Fig1a: to help the readers to look at Fig1a and following the first sentence in the results (at L163), I would add a thin (maybe dashed?) contour to highlight the -20℃ isotherm.
- Fig10: It would help the reader to look at the figure if columns had their own titles, e.g., "all concurrent events", "Cluster 1: strong SLP anomaly dipole", "Cluster 2: blocking-like surface anticyclone" and "Cluster 3: strong Greenland SLP anomaly". Similarly for Fig. 11. The number of events in each of the 3 clusters would also be good to know, maybe add in the figure caption for Fig. 10?
- Fig 15: some lines cover the text in the legends, which makes it hard to read the legend.
- All map figures: As the latitude band of 83N is of importance in this study, I would suggest to explicitly mark that latitude in all maps.
- S2: Thank you for adding the climatological sea ice edge contour. Why did the authors choose a SIC of 50 %? As far as I know, the sea ice edge is usually marked as SIC of 15 %. If the figure stays similar with a SIC of 15 %, I would suggest changing.
- L72: "This event *was* driven by an …" instead of "is".
- L91: I would add "local" before "extreme warming events" to emphasise the grid-point based defined events, as the authors are here referring to the studies where local buy observations are utilized.
- L170: Add "mainly" after "Therefore, we focus", because the authors do return to the high Arctic definition later on in the paper, as in the trends.
- L292: I would add the temperature unit after the "zero" (mentioned twice on this line)
- L 400: did the authors forget to include "do" prior to "… changes in AR frequency …"?
- L432: shortly remind the readers here what is meant by "concurrent warming events"

---

## Author Response (AR2)

**egusphere-2023-2018: Response to Second Review Comments**

We thank both referees for carefully reading our revised manuscript and providing constructive comments that further helped improve both the quality and clarity of this work. Below please see our point-by-point responses to all review comments and revisions made to the manuscript. The original comments are in blue.

**Referee #1**

**General comments:**

The authors have put a lot of effort into the revisions and have addressed many of my previous concerns. The results are interesting and provide insight into the characteristics of extreme Arctic warm events, defined first per grid point and then as contiguous regions with T2m > 0°C (concurrent warm events). I particularly like the analyses of the concurrent warm events. My major concern is still the same as in the first version of the manuscript, namely the definition of Arctic warm events per grid point (see comment 1 below). Many of my remaining comments concern language and terminology and can be addressed with some rewording.

We thank the referee again for carefully reviewing our manuscript and providing suggestions which greatly improve the clarity of this manuscript. Our responses to the specific comments are shown below, with their original comments in blue.

**Specific comments:**

1) You have explained in detail why you define Arctic warm events at each grid point separately, but I am still struggling with the fact that adjacent grid points with T2m > 0°C at the same time are referred to as many separate "events". You motivate this with the studies by Moore (2016) and Graham et al. (2017), who used meteorological buoy observations to study the warm event in December 2015. However, even if they looked at different point observations, they did not refer to them as different Arctic warm events – they used the point observations to characterize one single event. I am ok with your method (although I am more convinced by your definition of concurrent warm events, which, however, is not the main focus of the paper), but the problem for me is the word "events" that does not always fit well in the text and can lead to confusion. Maybe you can carefully scan your manuscript and write more often something like "grid points with T2m > 0°C" instead of "Arctic warm events". For instance, you write about the onset of the event (or, in the revised version, the start of the event, which for me is actually the same as the onset) and its termination, but I find the suggestion of the second reviewer much more convincing to use something like "time when a grid point's T2m exceeds / falls below 0°C", as the warm temperatures are most likely simply being advected from one grid point to the next. And instead of short-duration and long-duration events, maybe you could use something like "grid points experiencing the warming for a short / long duration". I understand that this also makes the text more cumbersome, and you can't replace all your mentions of "event" with a formulation like this, but wherever possible this would increase the accuracy of the text and help to minimize confusion for the reader.

Thank you for the suggestions on using different wordings to describe the extreme warming events defined at the grid-point scale, which help to improve the accuracy of the descriptions and avoid potential confusion. Following the referee's suggestions, we explicitly defined extreme warming events using descriptions, such as "defined at the grid-point scale" and "with T2m >= 0°C" wherever appropriate. In addition, when we discussed the start and termination of the extreme warming events, the description "when a grid point's T2m first exceeds/ falls below 0°C" was used to properly define them. Lastly, we also explicitly defined short/long duration events whenever appropriate using something like "grid points experiencing the warming for a short/long duration", as suggested by the referee.

2) I find your new Fig. S4 very interesting, but I am quite surprised by the large number of days per season with at least one grid point with T2m > 0°C. According to this figure, these events are actually not that rare, with most winters having at least one such warm event and many winters even having one in more than 20 out of the 90 days. This is different to the findings of previous studies. For instance, according to Moore 2016, Arctic warm events occur once or twice each decade, and Binder et al. 2017 found 12 out of 36 winters with Arctic warm events. Of course, these studies are based on different methods, different datasets, different study regions, etc., but I am still surprised by the quite different findings. Maybe you could briefly discuss your results and explain why you have a much higher number of events than found in previous studies. Also, your sentence in line 16 in the abstract ("They occur rarely, with a total absence during some winters over most of the region.") and the text in Section 3.1 might be revised a bit. And in my opinion, it would be more informative if you could write in how many of the 42 winters there has been at least one Arctic warm event somewhere in the studied region (your new Fig. S4).

As pointed out by the referee, the occurrence frequency of the Arctic warm events in Moore (2016) and Binder et al. (2017) are based on different methods, different datasets, different study regions, etc. In particular, Moore (2016) defined Arctic warm events as those events with surface air temperature (or T2m) >= 0°C occurring poleward of 85°N during **December** in JRA55. They identified three Arctic warm events in the study period from 1958 to 2015. As shown in our manuscript, if we restrict our focus to regions poleward of 85°N, there are only 9 days in total when Arctic warm events with T2m >= 0°C were found during **December-February** from 1979 to 2021. In this case, the difference in the occurrence frequency of the Arctic warm events between our study and Moore (2016) is small. The slightly more events identified in our studies could be attributed to the difference in the definition of the winter season (December versus December-February), the difference in the studied period (1958-2015 versus 1979-2021), and difference in the datasets used (JRA55 versus ERA5). In particular, the period from 2015 to 2021, which included in our study, contains 3 days with T2m >= 0°C poleward of 85°N. In addition, ERA5 has a higher spatiotemporal resolution compared to JRA55 (0.25° x 0.25° versus 1.25° x 1.25° and hourly versus 6-hourly). This higher spatiotemporal resolution in ERA5 makes the detection of Arctic warm events more likely.

Binder et al. (2017) defined Arctic warm events as those events with T2m >= 0°C occurring poleward of **82°N** during December-February in ERA-Interim. They found 12 out of 36 winters with Arctic warm events from 1979 to 2014. If we also restrict our focus to regions poleward of 82°N, as shown in Fig. R1, we found 16 out of 36 winters with Arctic warm events from 1979 to 2014 in ERA5, which brings our result much closer to the result in Binder et al. (2017). Again,

the slightly more winters with Arctic warm events found in our study could be attributed to the different dataset (ERA5) with higher spatiotemporal resolution used in our study.

We agree with the referee that, Arctic-Wide, based on Fig. S4, these extreme warming events defined at the grid-point scale are not rare. However, based on Fig. 3, they do occur rarely over many of the grid points where T2m ever reaches or exceeds 0°C during the study period. We thus revise line 16 in the abstract from "*They occur rarely, with a total absence during some winters over most of the region.*" to "*They occur rarely over many grid points, with a total absence during some winters.*" In addition, we also revised line 180 in Section 3.1 from "*In addition to being short-lived, these events also occurred very rarely over the winter high Arctic (Fig. 3).*" to "*In addition to being short-lived, these events also occurred very rarely over many of the grid points of the winter high Arctic (Fig. 3).*"

The following discussion has also been added to Section 3.1:

"*Note that while the occurrence frequency of these extreme warming events can be considered rare over many of the grid points, over the entire high Arctic, they occur quite frequently. There was at least one extreme warming event somewhere in the studied region in 40 out of the 42 studied winters (Fig. S1). This appears to disagree with the findings in Moore (2016) and Binder et al. (2017). Specifically, Moore (2016) found that wintertime Arctic extreme warming events with T2m >= 0°C occur only once or twice each decade. Three events were identified from 1958 to 2015 in their study. Binder et al. (2017) found 12 out of the 36 winters with at least one extreme warming event over the wintertime high Arctic from 1979 to 2014. These seeming discrepancies between the results in our studies and those found in Moore (2016) and Binder et al. (2017) stem mostly from the differences in the datasets used, the definition of the high Arctic and the definition of the winter season. In particular, Moore (2016) defines Arctic extreme warming events as those events occurring poleward of 85°N during December in Japanese 55-year Reanalysis (JRA-55) (Kobayashi et al., 2015). As shown later, there are only 9 days with extreme warming events identified during the studied period if we change the analysis regions to poleward of 85°N, which brings our result much closer to that in Moore (2016). The slightly more events identified in our study can be attributed to the differences in the definition of winter season (i.e., December-February versus December) and datasets used (i.e., ERA5 versus JRA55). Following Binder et al. (2017), when the high Arctic is defined as regions poleward of 82°N, we find 16 out of 36 winters with extreme warming events from 1979 to 2014 (not shown), which also makes our quantitative result more comparable to that in Binder et al. (2017). The remaining difference is likely due to the different spatiotemporal resolution of datasets used between studies (i.e., ERA5 versus ERA-Interim).*"

[Figure]

Figure R1. The same as Fig. S4, but with the high Arctic defined as regions poleward of 82°N. Time series in (a) the number of days per season and (b) the number of hours per season with at least one warming event defined at the grid point level occurring poleward of 82°N.

3) Section 2.1: Even though you rewrote large parts of the section, it remains quite difficult to read and to quickly get the relevant information, as you mix in information that rather belongs to the introduction or even the conclusion, and you write a lot about how others defined the events and how the events could be defined, before actually writing how you define them in a different way. My suggestion would be to write a first subsection with the title "Data", where you describe the ERA5 data, then a second subsection "Definition of extreme Arctic warming events", both in a more streamlined form, and a final subsection "Calculation of anomaly fields". For instance, when describing ERA5 I would leave away the first sentence "Results based on previous case studies suggest that extreme warming events with near surface air temperature above 0 °C tend to be short-lived, …", as you already mentioned this in the introduction and here it rather disturbs the flow. And also the last sentence of the paragraph "The results being presented here thus bear significance to our understanding of …" does not really fit into the data and method section but is rather part of the motivation or even the conclusion of the study. And in the subsection where you define the events, I would start with writing something like "In this study, we define Arctic warm events in two ways: first, as grid points with T2M > 0°C, which is the main focus of the study, and second, as contiguous regions with T2m > 0°C … " (and please also add more details about how you define the contiguous events). Afterwards, it is nice if you write about the motivation of defining the events in these ways, and their advantages and disadvantages.

Thank you for the great suggestions on restructuring Section 2.1. Section 2.1 is now broken into three subsections and subsequently revised following the referee's suggestions.

4) I am not always convinced by the words "drivers", "direct drivers" and "directly driving", which are all used very often in the paper (e.g., abstract line 23, Section 3.2 and Section 3.4). In general, there is not just one driver, but it is the interplay between various processes and weather systems that contributes to an Arctic warm event. For instance, in line 23 in the abstract, you write that "ARs are the direct driver for 100% of these event", but I would rather use "associated with" or "related to" (and similarly in many other parts of the text). You already changed the words in some parts of the paper in the revised version, but they still appear in many places where I don't think that they are suitable.

We have used more appropriate wordings in places where "drivers", "direct drivers" and "directly driving" were originally used.

5) "Subsequent examination reveals the short-lived nature of this event, with the duration of staying above 0°C for less than an hour locally …"
This sentence is not so nice – I would write something like "with buoys close to the North Pole recording temperatures above 0°C for less than an hour …"

We have revised the sentence following the referee's suggestion.

**Referee #2**

I highly appreciate the effort of the authors to address both mine and the other reviewer's concerns and feedback, especially regarding the warm event definition and clearly stating which events are discussed in each section of the manuscript. I am pleased to see that some of our small suggestions were also adapted immediately in the revised manuscript. Thus, I find that the manuscript has now improved a lot. After addressing a few questions and suggestions presented below in terms of a minor review, I am happy to accept this manuscript for publication to the journal. The line references are wrt the reviewed manuscript.

We thank the referee for carefully reviewing our manuscript. The detailed comments provided by the referee help improve the clarity and quality of both the text and figures of the manuscript. Our responses to the specific comments are shown below, with their original comments in blue.

**Specific Comments:**

- Despite my concerns on the event definition as a grid-point defined event, the authors have succeeded to explain the pros and cons on their selected method. It is true, that grid-point defined warming events favour that one can understand more about what processes affect locally a specific grid point. Thereby, the spatial heterogeneity of characteristics, such as event duration, will be revealed, which might not be the case when looking at events over larger areas. This is also true when investigating e.g., why some grid-points experience earlier melt or freeze-up onset compared to other grid-points, something that could be missing if enlarging the study area. And I agree when the authors write that the large-scale setting differs relative to the relative location of the grid-point that experience warming. The relation to large-scale patterns can still be obtained if using area-based event definitions, but might not indicate as clearly the importance of the relative location. Anyways, thanks for all additional discussions and clarifications regarding the event definition, this will be helpful for the future readers. However, I have a short comment on the discussion authors provide on the temporal requirement between events (which is not obtained in the current analysis), and my concerns of double-counting of similar events. Figure 6 shows the temporal evolution in lag composites for your grid-point events where no interval requirement between consecutive events is applied. In the response

documents, the authors provided a similar figure (R9) with a 5-day interval requirement and indicated that the composites do look alike. I don't disagree with the authors, but would like to draw attention to the long-lived events (R9g) with notable differences between the original figure wrt the turbulent heat fluxes (negative values instead of around zero) and IWV around lags -5 to -2. This would indeed suggest on somewhat double counting of events in Fig 6: are these differences only a result of less events included in R9g, or is this really a result of that the previous long-lasting event was actually part of the begin of the next long-lasting event? Maybe one additional notation of this possibility would be good in the manuscript, where Fig 6 is discussed and/or where the authors motivate reasons why not imposing the temporal constraint (around L122).

Thank you for pointing out the difference between Fig. R9g and Fig. 6g. Indeed, after imposing a 5-day interval requirement, the anomalies of turbulent heat fluxes, T2m and IWV become negative. As also nicely suggested by the referee, this is likely caused by the previous long-lasting events being double counted as part of the next long-lasting events prior to their onset. Again, as shown in Fig. R2, imposing the 5-day interval requirement would not change the conclusion regarding the relative position of the SLP dipole for long duration events. To point out the sensitivity of our results to the imposed time interval requirement, the following discussion has been added to Section 2.2 in the revised manuscript:

"*We found that only the surface energy budget of the long duration events (defined in Section 3.2) prior to the onset is sensitive to whether a time interval requirement is imposed. More specifically, imposing a 120-hour interval requirement makes the anomalies of turbulent fluxes, T2m and integrated water vapor decrease from around zero (Fig. 6g) to a negative value from around 5-day lag to 2-day lag (not shown). This suggests that previous long-lasting events are probably double counted as part of the next long-lasting events prior to their onset when no time interval is imposed. Other than that, all the results presented in this study are very similar regardless of whether this additional constraint is imposed or not (not shown). For simplicity, the results being presented in this study are obtained without imposing this constraint.*"

[Figure]

Figure R2. Same as Fig. 9 in the main text, but with a 5-day interval requirement imposed. Composites centred at the event grid point for the temporal evolution of integrated water vapor transport (IVT) anomalies (vectors), sea level pressure (SLP) anomalies (lines) and temperature advection anomalies (shading) before, during and after the start of the long duration extreme warming events (i.e., grid points with T2m >= 0°C at least 40 hours). The green star in each panel marks the grid point where the extreme warming events took place. Regions 5° poleward, 20° equatorward, 100° westward/eastward of the event grid point are included in the composites.

- It is nice to see the additional motivations in dividing your events into poleward and equatorward of ~85N based on the surface types. I was wondering whether the trends in any characteristics shown in Fig. 15 (thanks for the additional subplots for the concurrent events!) take into account this division (into "surface type" based on spatial restrictions)? Would the trends look different in the sub-categories?

This is a great question. However, as has been discussed in Section 3.4, there are only 9 days with T2m >= 0°C poleward of 85°N during the studied period. These nine days occur over six different years. Given such a small sample size of the events over regions poleward of 85°N, it would be difficult to calculate their trends. The trends shown in Fig. 15 thus mostly reflect the trends for the events occurring equatorward of 85°N.

- Abstract: I would suggest adding "2-meter" temperature (L12) as the temperature variable.

  "2-meter" has now been added in the abstract to describe temperature.

- New final paragraph: Thanks for adding this at the end to summarize and indicate possible further research topics. However, there are indeed some previous studies that show e.g., the relationship between persistent atmospheric circulation in March and the minimum sea ice extent in September (see e.g., Kapsch et al. 2019: https://doi.org/10.1007/s00382-018- 4279-z), and the preconditioning of warm winter/delayed freeze-up in autumn for thinner ice next spring (e.g., Stroeve et al. 2018: https://doi.org/10.5194/tc-12-1791-2018). In comparison to these papers, a good addition would be to study these links in the light of your identified warming events, as the authors nicely point out.

  Thank you for pointing us to these two relevant papers. They have been now properly cited in the last paragraph.

  "*Links between March persistent atmospheric circulation and September minimum sea ice extent (Kapsch et al., 2019) and between warm winter and subsequent thinner spring sea ice over the Arctic (Stroeve et al., 2018) have been established by previous studies.*"

- The ECMWF is written out in two sections close to each other (at the end of Introduction and in the beginning of Methods). I would include it when it is mentioned the first time, and just use the acronym when it is mentioned the second time.

  Thank you for spotting this redundancy. Now only the acronym is used when it is mentioned the second time.

- L284: The location of the negative SLP anomalies to the southeast of the events in longlasting events could also hint for a NAO+, a circulation pattern that would favour moisture and heat transport across the Atlantic towards the Scandinavia. The blocking-like persistent positive SLP anomalies over Scandinavia/Urals would then deflect that airflow northwards, towards the warm events. Studies also show that the decay of a NAO+ pattern could lead to an enhancement of the blocking to the east of the NAO pattern, and a warm anomaly in the Arctic. As your Fig. 9 is not exactly a geographic map (as you have centred the plots around the event location), these patterns might not be exactly over Iceland (negative SLP) and Eurasia (positive SLP), but would cover about the correct regions given that the possible locations for the extreme warm events are spatially constrained in the Atlantic sector. Have the authors thought about these possible connections wrt the long-lasting events? Some references, e.g., Luo et al. 2016 (https://doi.org/10.1175/JCLI-D-15-0612.1), Luo et al. 2017

(https://doi.org/10.1088/1748-9326/aa69d0), and Murto et al. 2022
(https://doi.org/10.5194/wcd-3-21-2022).

This is a great question. As has been shown in Luo et al. 2017, the combination of positive NAO and Ural blocking provides an optimal circulation pattern which favors moisture and heat intrusions into the Arctic through the Atlantic pathway. We have shown in our study that extreme warming events defined at the grid-point scale are more likely to occur under AR conditions. We thus believe that positive NAO likely contributes to more occurrences of the extreme warming events, including long-lasting events. To see what sea level pressure (SLP) pattern is associated with increased long-lasting event occurrence frequency, we regress the winter SLP anomaly time series onto the detrended long-lasting event occurrence frequency time series. As shown in Fig. R3, the SLP regression pattern features a blocking-like structure over Northern Eurasia. This seems to suggest that the interannual variability of the long-lasting event frequency is largely controlled by the blocking-like structure, and NAO may play a secondary role. However, a more detailed and quantitative analysis is needed to better establish the relationship between NAO and the occurrence frequency of the long-lasting events.

[Figure]

Figure R3. Spatial pattern of the regression of the winter sea level pressure (SLP) anomaly time series onto the detrended long-lasting event occurrence frequency time series. Stippled areas indicate the regression is significant at the 0.05 level based on the Student's t-test.

- L379: deep penetration of ARs associated with an SLP center located more polewards. Could this be related to locally formed Arctic cyclones, with cyclogenesis in the Arctic north of Greenland? There are studies, such as Messori et al. 2018 and Murto et al. 2022, that also find these local cyclones (associated with negative SLP anomalies polewards) to occur around the time of the warm events. Maybe worth to discuss this possibility also in the current manuscript?

It is totally possible that these deep penetration of ARs are associated with locally formed Arctic cyclones. To account for this possibility, the following discussion has been added to Section 3.4 of this manuscript:

"*As has been shown in previous studies (Messori et al., 2018; Murto et al., 2022), the more poleward located negative SLP center could be associated with the locally generated Arctic cyclones.*"

**Technical corrections:**

- Fig1 caption: I would add "in (b)" between "The purple line" and "denotes…".

  Added.

- Fig1a: to help the readers to look at Fig1a and following the first sentence in the results (at L163), I would add a thin (maybe dashed?) contour to highlight the -20°C isotherm.

  A dashed contour has been added to Fig. 1a to highlight the -20°C isotherm.

- Fig10: It would help the reader to look at the figure if columns had their own titles, e.g., "all concurrent events", "Cluster 1: strong SLP anomaly dipole", "Cluster 2: blocking-like surface anticyclone" and "Cluster 3: strong Greenland SLP anomaly". Similarly for Fig. 11. The number of events in each of the 3 clusters would also be good to know, maybe add in the figure caption for Fig. 10?

  A short title has been added to each column in Figs. 10 and 11. In addition, the number of events in each cluster has also been specified in the caption of Fig. 10.

- Fig 15: some lines cover the text in the legends, which makes it hard to read the legend.

  We have adjusted the figure so that the legends are now not covered by lines.

- All map figures: As the latitude band of 83N is of importance in this study, I would suggest to explicitly mark that latitude in all maps.

We tried marking the 83°N latitude in some of the map figures. After adding the 83°N latitude, these figures appear slightly more distracting to us. To keep the figures clean, we decided not to mark the 83°N latitude in the map figures.

- S2: Thank you for adding the climatological sea ice edge contour. Why did the authors choose a SIC of 50 %? As far as I know, the sea ice edge is usually marked as SIC of 15 %. If the figure stays similar with a SIC of 15 %, I would suggest changing.

Winter sea ice concentration is always higher than 15% poleward of 80°N during the studied period. We thus decide to draw the climatological 50% sea ice concentration contour.

- L72: "This event was driven by an …" instead of "is".

Corrected.

- L91: I would add "local" before "extreme warming events" to emphasise the grid-point based defined events, as the authors are here referring to the studies where local buy observations are utilized.

As suggested by referee #1, this sentence has been removed from the main text.

- L170: Add "mainly" after "Therefore, we focus", because the authors do return to the high Arctic definition later on in the paper, as in the trends.

Added.

- L292: I would add the temperature unit after the "zero" (mentioned twice on this line)

These two zeros are referring to the area with T2m >= 0°C. To clarify, the original sentence "*The onset of these concurrent warming events is then defined as the time when the area first exceeds zero (one grid point), and the event ends when the area first falls back to zero.*" has been revised as "*The onset of these concurrent warming events is then defined as the time when the area with T2m >= 0°C first exceeds zero (one grid point), and the event ends when the area with T2m >= 0°C first falls back to zero.*"

- L 400: did the authors forget to include "do" prior to "… changes in AR frequency …"?

We have now added "whether" before "changes in AR frequency".

- L432: shortly remind the readers here what is meant by "concurrent warming events"

We have added "*defined as a contiguous region with T2m >= 0℃ concurrently*" after "concurrent warming events" to remind readers the meaning of concurrent warming events.